# The Three Stages of Learning Dynamics in High-dimensional Kernel Methods

## Abstract

To understand how deep learning works, it is crucial to understand the training dynamics of neural networks. Several interesting hypotheses about these dynamics have been made based on empirically observed phenomena, but there exists a limited theoretical understanding of when and why such phenomena occur.

In this paper, we consider the training dynamics of gradient flow on kernel least-squares objectives, which is a limiting dynamics of SGD trained neural networks. Using precise high-dimensional asymptotics, we characterize the dynamics of the fitted model in two "worlds": in the *Oracle World* the model is trained on the population distribution and in the *Empirical World* the model is trained on a sampled dataset. We show that under mild conditions on the kernel and $L^2$ target regression function the training dynamics undergo three stages characterized by the behaviors of the models in the two worlds. Our theoretical results also mathematically formalize some interesting deep learning phenomena. Specifically, in our setting we show that SGD progressively learns more complex functions and that there is a "deep bootstrap" phenomenon: during the second stage, the test error of both worlds remain close despite the empirical training error being much smaller. Finally, we give a concrete example comparing the dynamics of two different kernels which shows that faster training is not necessary for better generalization.

## 1 Introduction

In order to fundamentally understand how and why deep learning works, there has been much effort to understand the dynamics of neural networks trained by gradient descent based algorithms. This effort has led to the discovery of many intriguing empirical phenomena (e.g. Frankle et al. (2020); Fort et al. (2020); Nakkiran et al. (2019a;b; 2020)) that help shape our conceptual framework for understanding the learning process in neural networks. Nakkiran et al. (2019b) provides evidence that SGD starts by first learning a linear classifier and over time learns increasingly complex functions. Nakkiran et al. (2020) introduces the "deep bootstrap" phenomenon: for some deep learning tasks the empirical world test error remains close to the oracle world error[1] for many SGD iterations, even if the empirical training and test errors display a large gap. To better understand such phenomena, it is useful to study training dynamics in related but mathematically tractable settings.

One approach for theoretical investigation is to study kernel methods, which were recently shown to have a tight connection with over-parameterized neural networks (Jacot et al., 2018; Du et al., 2018). Indeed, consider a sequence of neural networks $(f_N(\boldsymbol{x}; \boldsymbol{\theta}))_{N \in \mathbb{N}}$ with the widths of the layers going to infinity as $N \to \infty$. Assuming proper parametrization and initialization, for large $N$ the SGD dynamics on $f_N$ is known to be well approximated by the corresponding dynamics on the first-order Taylor expansion of $f_N$ around its initialization $\boldsymbol{\theta}^0$,

$$f_{N,\text{lin}}(\boldsymbol{x}; \boldsymbol{\theta}) = f_N(\boldsymbol{x}; \boldsymbol{\theta}^0) + \langle \boldsymbol{\nabla}_{\boldsymbol{\theta}} f_N(\boldsymbol{x}; \boldsymbol{\theta}^0), \boldsymbol{\theta} - \boldsymbol{\theta}^0 \rangle.$$

Thus, in the large width limit it suffices to study the dynamics on the linearization $f_{N,\text{lin}}$. When using the squared loss, these dynamics correspond to optimizing a kernel least-squares objective with the neural tangent kernel $K_N(\boldsymbol{x}, \boldsymbol{x}') = \langle \nabla_{\boldsymbol{\theta}} f_N(\boldsymbol{x}; \boldsymbol{\theta}^0), \nabla_{\boldsymbol{\theta}} f_N(\boldsymbol{x}'; \boldsymbol{\theta}^0) \rangle$.

---

[1] Their paper uses "Ideal World" for "Oracle World" and "Real World" for "Empirical World".

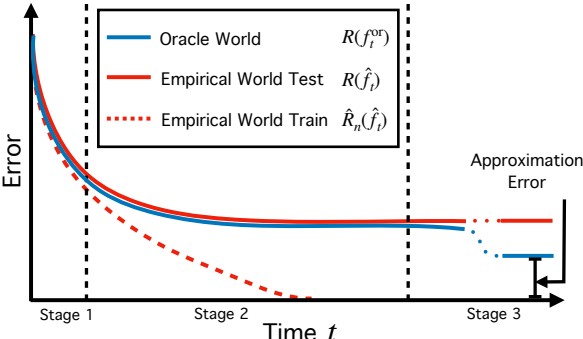

Figure 1: A conceptual drawing of empirical and oracle world learning curves. **Stage 1:** all curves are together. **Stage 2:** training error goes to zero while test and oracle error stay together. **Stage 3:** test error remains constant while oracle error decays to the RKHS approximation error. See Section 1.1 for a more detailed discussion. (Dotted lines in stage 3 indicate compressed time interval.)

Over the past few years, researchers have used kernel machines as a tractable model to investigate many neural network phenomena including benign overfitting, i.e., generalization despite the interpolation of noisy data (Bartlett et al., 2020; Liang & Rakhlin, 2020) and double-descent, i.e, risk curves that are not classically U-shaped (Belkin et al., 2020; Liu et al., 2021). Kernels have also been studied to better understand certain aspects of neural network architectures such as invariance and stability (Bietti & Mairal, 2017; Mei et al., 2021b). Although kernel methods cannot be used to explain some phenomena such as feature learning, they can still be conceptually useful for understanding other neural networks properties.

## 1.1 THREE STAGES OF KERNEL DYNAMICS

Despite much classical work in the study of gradient descent training of kernel machines (e.g. Yao et al. (2007); Raskutti et al. (2014)) there has been limited work understanding the high-dimensional setting, which is the setting of interest in this paper. Although solving the linear dynamics of gradient flow is simple, the statistical analysis of the fitted model requires involved random matrix theory arguments. In our analysis we study the dynamics of the *Oracle World*, where training is done on the (usually inaccessible) population risk, and the *Empirical World*, where training is done on the empirical risk (as is done in practice). Associated with the oracle world model $f_t^{\mathrm{or}}$ and the empirical world model $\hat{f}_t$ are the following quantities of interest: the **empirical training error** $\widehat{R}_n(\hat{f}_t)$, the **empirical test error** $R(\hat{f}_t)$, and the **oracle error** $R(f_t^{\mathrm{or}})$ defined in Eqs. (1), (2), (3) for which we derive expressions that are accurate in high dimensions.

Informally, our main results show that under reasonable conditions on the regression function and the kernel the training dynamics undergo the following three stages:

- **Stage one**: the empirical training error, the empirical test error, and the oracle error are all close.
- **Stage two**: the empirical training error decays to zero, but the empirical test error and the oracle error stay close and keep approximately constant.
- **Stage three**: the empirical training error is still zero, the empirical test error stays approximately constant, but the oracle test error decays to the approximation error.

We conceptually illustrate the error curves of the oracle and empirical world in Fig. 1 and provide intuition for the evolution of the learned models in Fig. 2. The existence of the first and third stages are not unexpected: at the beginning of training the model has not fit the dataset enough to distinguish the oracle and empirical world and at the end of training an expressive enough model with infinite samples will outperform one with finitely many. The most interesting stage is the second one where the empirical model begins to "overfit" the training set while still remaining close to the non-interpolating oracle model in the $L^2$ sense (see Fig. 2).

In Section 2 we discuss some related work. In Section 3 we elaborate our description of the three stages and give a mathematical characterization for two particular settings in Theorems 1 and 2.

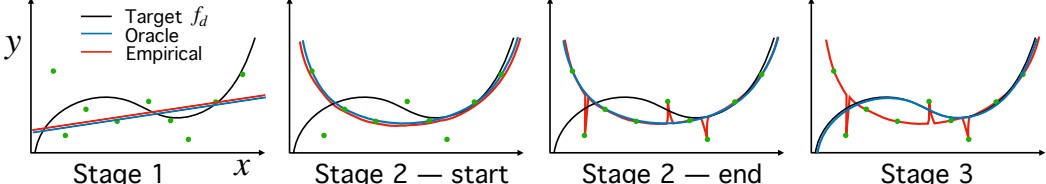

Figure 2: A conceptual drawing of the evolution of the empirical and oracle models $\hat{f}_t$ and $f_t^{\mathrm{or}}$. In stage 1, $\hat{f}_t$ and $f_t^{\mathrm{or}}$ learn the best linear approximation of $f_d$. At the start of stage 2, $\hat{f}_t$ and $f_t^{\mathrm{or}}$ learn the best quadratic approximation. At the end of stage 2, $\hat{f}_t$ interpolates the training set but is close to $f_t^{\mathrm{or}}$ in the $L^2$ sense. Lastly in stage 3, $f_t^{\mathrm{or}}$ learns $f_d$ while $\hat{f}_t$ stays the same as the end of stage 2.

Although the three stages arise fairly generally, we remark that certain stages will vanish if the problem parameters are chosen in a special way (c.f. Remark 1). We connect our theoretical results to related empirical deep learning phenomena in Remark 3 and discuss the relation to deep learning in practice in Remark 4. In Section 4 we provide numerical simulations to illustrate the theory more concretely and in Section 5 we end with a summary and discussion of the results.

## 2    RELATED LITERATURE

The generalization error of the kernel ridge regression (KRR) solution has been well-studied in both the fixed dimension regime (Wainwright, 2019, Chap. 13), (Caponnetto & De Vito, 2007) and the high-dimensional regime (El Karoui, 2010; Liang & Rakhlin, 2020; Liu et al., 2021; Ghorbani et al., 2020; 2021; Mei et al., 2021a;b). Most closely related to our results is the setting of (Ghorbani et al., 2021; Mei et al., 2021a;b). Analysis of the entire KRR training trajectory has also been done (Yao et al., 2007; Raskutti et al., 2014; Cao et al., 2019) but only for the fixed dimensional setting. Classical non-parametric rates are often obtained by specifying a strong regularity assumption on the target function (e.g. the source condition in Fischer & Steinwart (2020)), whereas in our work the assumption on the target function is mild.

Another line of work directly studies the dynamics of learning in linear neural networks (Saxe et al., 2013; Li et al., 2018; Arora et al., 2019; Vaskevicius et al., 2019). Similar to us, these works show that some notion of complexity (typically effective rank or sparsity) increases in the linear network over the course of optimization.

The relationship between the speed of iterative optimization and gap between population and empirical quantities has been studied before in the context of algorithmic stability (Bousquet & Elisseeff, 2002; Hardt et al., 2016; Chen et al., 2018). These analyses certify good empirical generalization by using stability in the first few iterations to upper bound the gap between train and test error. In contrast, our analysis directly computes the errors at an arbitrary time $t$ (c.f. Remark 2). The relationship between oracle and empirical training dynamics has been considered before in Bottou & LeCun (2004) and Pillaud-Vivien et al. (2018).

## 3    RESULTS

In this section we introduce the problem and present a specialization of our results to two concrete settings: dot product and group invariant kernels on the sphere (Theorems 1 and 2 respectively). The more general version of our results is described in Appendix A.3.

### 3.1    PROBLEM SETUP

We consider the supervised learning problem where we are given i.i.d. data $(\boldsymbol{x}_i, y_i)_{i \leq n}$. The covariate vectors $(\boldsymbol{x}_i)_{i \leq n} \sim_{iid} \mathrm{Unif}(\mathbb{S}^{d-1}(\sqrt{d}))$ and the real-valued noisy responses $y_i = f_d(\boldsymbol{x}_i) + \varepsilon_i$ for some unknown target function $f_d \in L^2(\mathbb{S}^{d-1}(\sqrt{d}))$ and $(\varepsilon_i)_{i \leq n} \sim_{iid} \mathcal{N}(0, \sigma_\varepsilon^2)$. Given a function

$f \in L^2(\mathbb{S}^{d-1}(\sqrt{d}))$, we define its test error $R(f)$ and its training error $\widehat{R}_n(f)$ as

$$R(f) \equiv \mathbb{E}_{(\boldsymbol{x}_{\text{new}}, y_{\text{new}})}\{(y_{\text{new}} - f(\boldsymbol{x}_{\text{new}}))^2\}, \qquad \widehat{R}_n(f) \equiv \frac{1}{n} \sum_{i=1}^n (y_i - f(\boldsymbol{x}_i))^2 \qquad (1)$$

where $(\boldsymbol{x}_{\text{new}}, y_{\text{new}})$ is i.i.d. with $(\boldsymbol{x}_i, y_i)_{i \leq n}$. The test error $R(f)$ measures the fit of $f$ on the population distribution and the training error $\widehat{R}_n(f)$ measures the fit of $f$ to the training set.

For a kernel function $H_d : \mathbb{S}^{d-1}(\sqrt{d}) \times \mathbb{S}^{d-1}(\sqrt{d}) \to \mathbb{R}$, we analyse the dynamics of the following two fitted models indexed by time $t$: the *oracle model* $f_t^{\text{or}}$ and the *empirical model* $\hat{f}_t$, which are given by the gradient flow on $R$ and $\widehat{R}_n$ over the associated RKHS $\mathcal{H}_d$ respectively

$$\frac{\mathrm{d}}{\mathrm{d}t} f_t^{\text{or}}(\boldsymbol{x}) = -\boldsymbol{\nabla} R(f_t^{\text{or}}(\boldsymbol{x})) = \mathbb{E}[H_d(\boldsymbol{x}, \boldsymbol{z})(f_d(\boldsymbol{z}) - f_t^{\text{or}}(\boldsymbol{z}))], \qquad (2)$$

$$\frac{\mathrm{d}}{\mathrm{d}t} \hat{f}_t(\boldsymbol{x}) = -\boldsymbol{\nabla} \widehat{R}_n(\hat{f}_t(\boldsymbol{x})) = \frac{1}{n} \sum_{i=1}^n H_d(\boldsymbol{x}, \boldsymbol{x}_i)(y_i - \hat{f}_t(\boldsymbol{x}_i)), \qquad (3)$$

with zero initialization $f_0^{\text{or}} \equiv \hat{f}_0 \equiv 0$. These dynamics are motivated from the neural tangent kernel perspective of over-parameterized neural networks (Jacot et al., 2018; Du et al., 2018). A precise mathematical definition and derivation of these two dynamics are provided in Appendix E.1.

For our results we make some assumptions on the spectral properties of the kernels $H_d$ similar to those in Mei et al. (2021a) that are discussed in detail in Section A.2. At a high-level we require that the diagonal elements of the kernel concentrate, that the kernel eigenvalues obey certain spectral gap conditions, and that the top eigenfunctions obey a hyperconctractivity condition which says they are "delocalized". For the specific settings of Theorems 1 and 2 we give more specific conditions on the kernels that are more easily verified and imply the required spectral properties.

### 3.2 DOT PRODUCT KERNELS

In our first example, we consider dot product kernels $H_d$ of the form

$$H_d(\boldsymbol{x}_1, \boldsymbol{x}_2) = h_d(\langle \boldsymbol{x}_1, \boldsymbol{x}_2 \rangle / d), \qquad \forall \boldsymbol{x}_1, \boldsymbol{x}_2 \in \mathbb{S}^{d-1}(\sqrt{d}), \qquad (4)$$

for some function $h_d : [-1, 1] \to \mathbb{R}$. Our results apply to general dot product kernels under weak conditions on $h_d$ given in Appendix C.2. In particular they apply to the random feature and neural tangent kernels associated to certain fully connected neural networks (Jacot et al., 2018).

Before presenting our results for this setting we introduce some notation. Denote by $\overline{\mathsf{P}}_{\leq \ell}$ the orthogonal projection onto the subspace of $L^2(\mathbb{S}^{d-1}(\sqrt{d}))$ spanned by polynomials of degree less than or equal to $\ell$. The projectors $\overline{\mathsf{P}}_\ell$ and $\overline{\mathsf{P}}_{>\ell}$ are defined analogously (see Appendix G for details). We use $o_d(\cdot)$ for standard little-o relations, where the subscript $d$ emphasizes the asymptotic variable. The statement $f(d) = \omega_d(g(d))$ is equivalent to $g(d) = o_d(f(d))$. We use $o_{d,\mathbb{P}}(\cdot)$ in probability relations. Namely for two sequences of random variables $Z_1(d)$ and $Z_2(d)$, $Z_1(d) = o_{d,\mathbb{P}}(Z_2(d))$ if for any $\varepsilon, C_\varepsilon > 0$ there exists $d_\varepsilon \in \mathbb{Z}_{>0}$, such that $\mathbb{P}(|Z_1(d)/Z_2(d)| < C_\varepsilon) \leq \varepsilon$ for all $d \geq d_\varepsilon$. The asymptotic notations $O_d, O_{d,\mathbb{P}}$ etc. are defined analogously.

**Theorem 1** (Dot Product Kernels). *Let $\{f_d \in L^2(\mathbb{S}^{d-1}(\sqrt{d}))\}_{d \geq 1}$ be a sequence of functions such that for some $\eta > 0$, $\|f_d\|_{L^{2+\eta}} = O_d(1)$, and let $\{H_d\}_{d \geq 1}$ be a sequence of dot product kernels satisfying Assumption 4. Assume that for some fixed integers $\mathsf{j}, \mathsf{s} \geq 0$ and some $\delta > 0$ that*

$$d^{\mathsf{j}+\delta} \leq t \leq d^{\mathsf{j}+1-\delta}, \qquad d^{\mathsf{s}+\delta} \leq n \leq d^{\mathsf{s}+1-\delta}.$$

*Then we have the following characterizations,*

(a) *(Oracle World) The oracle model learns every degree component of $f_d$ as time progresses*

$$R(f_t^{\text{or}}) = \left\| \overline{\mathsf{P}}_{>\mathsf{j}} f_d \right\|_{L^2}^2 + \sigma_\varepsilon^2 + o_d(1), \qquad \left\| f_t^{\text{or}} - \overline{\mathsf{P}}_{\leq \mathsf{j}} f_d \right\|_{L^2}^2 = o_d(1).$$

(b) *(Empirical World – Train) Empirical training error follows oracle error then goes to zero*

$$\widehat{R}_n(\hat{f}_t) = \left\| \overline{\mathsf{P}}_{>\mathsf{j}} f_d \right\|_{L^2}^2 + \sigma_\varepsilon^2 + o_{d,\mathbb{P}}(1) \qquad \text{if } t/n = o_d(1),$$

$$\widehat{R}_n(\hat{f}_t) = o_{d,\mathbb{P}}(1) \qquad \qquad \qquad \text{if } t/n = \omega_d(1).$$

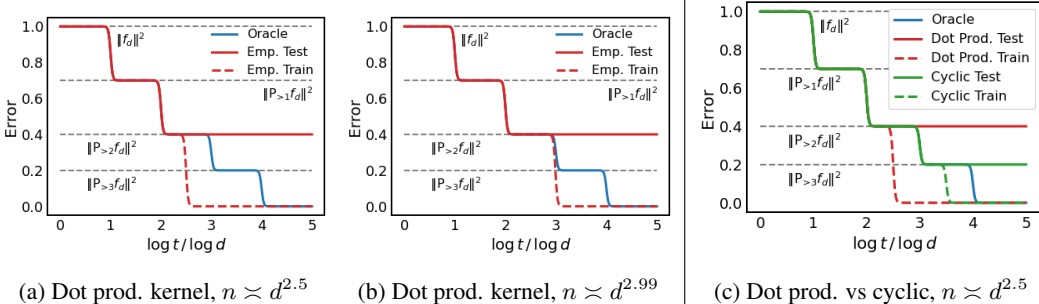

(a) Dot prod. kernel, $n \asymp d^{2.5}$     (b) Dot prod. kernel, $n \asymp d^{2.99}$     (c) Dot prod. vs cyclic, $n \asymp d^{2.5}$

Figure 3: Schematic drawings of the conclusions of Theorems 1 and 2 for three different noiseless settings. Panels (3a) and (3b) illustrate the performance of a dot product kernel $H_d$ for two different scalings $n(d)$. The full three stages appear in (3a), but the second stages disappears in (3b) since $\log n / \log d$ is nearly an integer (see Remark 1). Panel (3c) compares the performance of a dot product kernel $H_d$ (red) and corresponding cyclic kernel $H_{d,\mathrm{inv}}$ (green) for a cyclic target function $f_d$. The cyclic kernel in (3c) generalizes better but optimizes more slowly (see Remark 2).

*(c)* *(Empirical World – Test) Empirical test error follows oracle error until the empirical model learns the degree-*s *component of* $f_d$

$$R(\hat{f}_t) = \left\|\overline{\mathsf{P}}_{>\min\{\mathsf{j},\mathsf{s}\}} f_d\right\|_{L^2}^2 + \sigma_\varepsilon^2 + o_{d,\mathbb{P}}(1), \qquad \left\|\hat{f}_t - \overline{\mathsf{P}}_{\leq\min\{\mathsf{j},\mathsf{s}\}} f_d\right\|_{L^2}^2 = o_{d,\mathbb{P}}(1).$$

The results are conceptually illustrated in an example in Fig. 3a which shows the stair-case phenomenon in high-dimensions and the three learning stages. We see that in both the oracle world and the empirical world, the prediction model increases in complexity over time. More precisely, the model learns the best polynomial fit to the target function (in an $L^2$ sense) of increasingly higher degree. In the empirical world the maximum complexity is determined by the sample size $n$, which is in contrast to the oracle world where there are effectively infinite samples.

The results imply that generally (but not always c.f. Remark 1) there will be three stages of learning. In the first stage the oracle and empirical world models are close in $L^2$ and fit a polynomial with degree determined by $t$. The first stage lasts from $t = 0$ to $t = nd^{-\varepsilon} \ll n$ for some small $\varepsilon > 0$. As $t$ approaches $n$, there is a phase transition and the empirical world training error goes to zero at $t = nd^{\varepsilon} \gg n$. From time $nd^{-\varepsilon}$ till at least $d^{s+1}$ is the second stage where the empirical and oracle models remain close in $L^2$ but the gap between test and train error can be large. If the sample size $n$ is not large enough for $\hat{f}_t$ to learn the target function, then at some large enough $t$ we will enter a third stage where $f_t^{\mathrm{or}}$ improves in performance, outperforming $\hat{f}_t$ which remains the same. On synthetic data in finite dimensions we can see a resemblance of the staircase shape which becomes sharper with increasing $d$ (c.f. Appendix F).

**Remark 1** (Degenerate stages of Learning)**.** *For special problem parameters we will not observe the second and/or third stages. The second stage will disappear if* $n \asymp d^q$ *for some* $q \in \mathbb{N}$ *(see Fig. 3b), or if* $f_d$ *is a degree-*s *polynomial. The third stage will not occur if* $\overline{\mathsf{P}}_{>\mathsf{s}} f_d$ *lies in the orthogonal complement of the RKHS* $\mathcal{H}_d$.

### 3.3 GROUP INVARIANT KERNELS

We now consider our second setting which concerns the invariant function estimation problem introduced in Mei et al. (2021b). As before, we are given i.i.d. data $(\boldsymbol{x}_i, y_i)_{i \leq n}$ where the feature vectors $(\boldsymbol{x}_i)_{i \leq n} \sim_{iid} \mathrm{Unif}(\mathbb{S}^{d-1}(\sqrt{d}))$ and noisy responses $y_i = f_d(\boldsymbol{x}_i) + \varepsilon_i$. We now assume that the target function $f_d$ satisfies an invariant property. We consider a general type of invariance, defined by a group $\mathcal{G}_d$ that is represented as a subgroup of the orthogonal group in $d$ dimensions. The group element $g \in \mathcal{G}_d$ acts on a vector $\boldsymbol{x} \in \mathbb{R}^d$ via $\boldsymbol{x} \mapsto g \cdot \boldsymbol{x}$. We say that $f_\star$ is $\mathcal{G}_d$-invariant if $f_\star(\boldsymbol{x}) = f_\star(g \cdot \boldsymbol{x})$ for all $g \in \mathcal{G}_d$. We denote the space of square integrable $\mathcal{G}_d$-invariant functions by $L^2(\mathbb{S}^{d-1}(\sqrt{d}), \mathcal{G}_d)$. We focus on groups $\mathcal{G}_d$ that are *groups of degeneracy* $\alpha$ as defined below. As an example, we consider the cyclic group

$\text{Cyc}_d = \{g_0, g_1, \ldots, g_{d-1}\}$ where for any $\boldsymbol{x} = (x_1, \ldots, x_d)^\mathsf{T} \in \mathbb{S}^{d-1}(\sqrt{d})$, the group action is defined by $g_i \cdot \boldsymbol{x} = (x_{i+1}, x_{i+2}, \ldots, x_d, x_1, x_2, \ldots, x_i)^\mathsf{T}$. The cyclic group has degeneracy 1.

**Definition 1** (Groups of degeneracy $\alpha$). *Let $V_{d,k}$ be the subspace of degree-$k$ polynomials that are orthogonal to polynomials of degree at most $(k-1)$ in $L^2(\mathbb{S}^{d-1}(\sqrt{d}))$, and denote by $V_{d,k}(\mathcal{G}_d)$ the subspace of $V_{d,k}$ formed by polynomials that are $\mathcal{G}_d$-invariant. We say that $\mathcal{G}_d$ has degeneracy $\alpha$ if for any integer $k \geq \alpha$ we have $\dim(V_{d,k}/V_{d,k}(\mathcal{G}_d)) \asymp d^\alpha$ (i.e., there exists $0 < c_k \leq C_k < +\infty$ such that $c_k \leq \dim(V_{d,k}/V_{d,k}(\mathcal{G}_d)) \leq C_k$ for any $d \geq 2$).*

To encode invariance in our kernel we consider $\mathcal{G}_d$-invariant kernels $H_d$ of the form

$$H_{d,\text{inv}}(\boldsymbol{x}_1, \boldsymbol{x}_2) = \int_{\mathcal{G}_d} h(\langle \boldsymbol{x}_1, g \cdot \boldsymbol{x}_2 \rangle / d) \pi_d(\mathrm{d}g) \tag{5}$$

where $\pi_d$ is the Haar measuare on $\mathcal{G}_d$. Such kernels satisfy the following invariance property: for all $g, g' \in \mathcal{G}_d$ and for $H_d(\boldsymbol{x}_1, \boldsymbol{x}_2) = H_d(g \cdot \boldsymbol{x}_1, g' \cdot \boldsymbol{x}_2)$ for every $\boldsymbol{x}_1, \boldsymbol{x}_2$. For the cyclic group, $\pi_d$ is the uniform measure. We now present our results for the group invariant setting.

**Theorem 2** (Group Invariant Kernels). *Let $\mathcal{G}_d$ be a group of degeneracy $\alpha \leq 1$ according to Definition 1. Let $\{f_d \in L^2(\mathbb{S}^{d-1}(\sqrt{d}), \mathcal{G}_d)\}_{d \geq 1}$ a sequence of $\mathcal{G}_d$-invariant functions such that for some $\eta > 0$, $\|f_d\|_{L^{2+\eta}} = O_d(1)$, and let $\{H_d\}_{d \geq 1}$ be a sequence of $\mathcal{G}_d$-invariant kernels satisfying Assumption 5. Assume that for some fixed integers $\mathsf{j} \geq 0, \mathsf{s} \geq 1$ and some $\delta > 0$ that*

$$d^{\mathsf{j}+\delta} \leq t \leq d^{\mathsf{j}+1-\delta}, \qquad d^{\mathsf{s}-\alpha+\delta} \leq n \leq d^{\mathsf{s}+1-\alpha-\delta}.$$

*Then we have the following characterizations,*

*(a) (Oracle World) The oracle model learns every degree component of $f_d$ as time progresses*

$$R(f_t^{\text{or}}) = \left\| \overline{\mathsf{P}}_{>\mathsf{j}} f_d \right\|_{L^2}^2 + \sigma_\varepsilon^2 + o_d(1), \qquad \left\| f_t^{\text{or}} - \overline{\mathsf{P}}_{\leq \mathsf{j}} f_d \right\|_{L^2}^2 = o_d(1).$$

*(b) (Empirical World – Train) Empirical training error follows oracle error then goes to zero*

$$\begin{aligned}
\widehat{R}_n(\hat{f}_t) &= \left\| \overline{\mathsf{P}}_{>\mathsf{j}} f_d \right\|_{L^2}^2 + \sigma_\varepsilon^2 + o_{d,\mathbb{P}}(1) & \text{if } t/n = o_d(d^\alpha), \\
\widehat{R}_n(\hat{f}_t) &= o_{d,\mathbb{P}}(1) & \text{if } t/n = \omega_d(d^\alpha).
\end{aligned}$$

*(c) (Empirical World – Test) Empirical test error follows oracle error until the empirical model learns the degree-$\mathsf{s}$ component of $f_d$*

$$R(\hat{f}_t) = \left\| \overline{\mathsf{P}}_{>\min\{\mathsf{j},\mathsf{s}\}} f_d \right\|_{L^2}^2 + \sigma_\varepsilon^2 + o_{d,\mathbb{P}}(1), \qquad \left\| \hat{f}_t - \overline{\mathsf{P}}_{\leq \min\{\mathsf{j},\mathsf{s}\}} f_d \right\|_{L^2}^2 = o_{d,\mathbb{P}}(1).$$

With respect to the dot product kernel setting (c.f. Theorem 1), in this setting the behavior of the oracle world is unchanged, but the empirical world behaves as if it has $d^\alpha$ times as many samples. This is illustrated graphically in Fig. 3c. It can be shown that using an invariant kernel is equivalent to using a dot product kernel and augmenting the dataset to $\{(g \cdot \boldsymbol{x}_i, y_i) : g \in \mathcal{G}_d, i \in [n]\}$ (c.f. Appendix E.2). Hence for the cyclic group which has size $d^\alpha = d$, we arrive at the following intriguing conclusion: if the target function is invariant, then using a dot product kernel and augmenting $n$ i.i.d samples to $nd$ many samples is asymptotically equivalent to using $nd$ i.i.d samples.

**Remark 2** (Optimization Speed versus Generalization). *Interestingly, training with an invariant kernel is slower than with a dot product kernel and takes longer to interpolate the dataset despite eventually generalizing better on invariant function estimation tasks (c.f. Fig. 3c). This conclusion is not an artifact of the continuous time analysis (c.f. Appendix E.3) and is observed empirically in Section 4.2 for discrete-time SGD. This example highlights the limitation of stability based analyses (e.g. Hardt et al. (2016)) which argue that faster SGD training leads to better generalization. While a faster rate leads to better generalization in the first stage when stability can control the gap between train and test error, our analysis shows that the duration length of the first stage also impacts the final generalization error.*

**Remark 3** (Connection with Deep Phenomena). *The dynamics of high-dimensional kernel regression display behaviors that parallel some empirically observed phenomena in deep learning. For*

*kernel regression, we have shown that the complexity of the empirical model, measured as the number of learned eigenfunctions, depends on the time optimized when $t \ll n$ and the sample size when $t \gg n$. At a high-level, we also expect a similar story for neural networks but for some other notion of complexity. It is believed that neural networks first learn simple functions and then progressively more complex ones, until the complexity saturates after interpolating at some time proportional to $n$ (Nakkiran et al., 2019b). We have also shown that in kernel regression there is a non-trivial "deep boostrap" phenomenon (Nakkiran et al., 2020) during the second learning stage: the gap between the oracle world and empirical world test errors is negligible whereas the train and test errors exhibit a substantial gap. The gradient flow results for kernel regression can also provide insight into the deep bootstrap for random feature networks SGD as these results can approximately predict their behavior (see Section 4.2).*

**Remark 4** (Connection with Deep Learning Practice). *Although we believe our results conceptually shed light on some of the interesting behaviors observed in the training dynamics of deep learning, due to our stylized setting we may not exactly see the predicted phenomena in practice. Accurately observing the three stages of kernel regression requires sufficiently high-dimensional data in order for the kernel eigenvalues to obey a staircase-like decay and for training to be sufficiently long as the time axis should be in log-scale. Our results hold for regression whereas for classification the empirical model may continue improving after classifying the train set correctly. Despite these caveats, certain conclusions can be observed in some realistic settings (c.f. Appendix E.4).*

## 4 NUMERICAL SIMULATIONS

As mentioned previously, Fig. 3 is a "cartoon" of the conclusions stated in Theorems 1 and 2. In this section, we verify the qualitative predictions of our theorems using synthetic data. Concretely, throughout this section we take $d = 400$ and $n = d^{1.5} = 8000$, and following our theoretical setup (c.f. Section 3.1) generate covariates $(\boldsymbol{x}_i)_{i \leq n} \sim_{iid} \mathrm{Unif}(\mathbb{S}^{d-1}(\sqrt{d}))$ and responses $y_i = f_\star(\boldsymbol{x}_i) + \varepsilon_i$ with $(\varepsilon_i)_{i \leq n} \sim_{iid} \mathcal{N}(0, \sigma_\varepsilon^2)$, for different choices of target function $f_\star$. All simulations in this section are for the noiseless case $\sigma_\varepsilon^2 = 0$ but a noisy example is given in Appendix F.

In Section 4.1, we simulate the gradient flows of kernel least-squares with dot product kernels and cyclic kernels (Fig. 4) to reproduce the three stages as shown in Fig. 3. In Section 4.2 we show that SGD of (dot product and cyclic) random-feature models (Fig. 5) exhibit similar three stages phenomena, in which the second stage behaviors are consistent with the deep bootstrap phenomena observed in deep learning experiments (Nakkiran et al., 2020). Empirical quantities are averaged over 10 trials and the shaded regions indicate one standard deviation from the mean.

### 4.1 GRADIENT FLOW OF KERNEL LEAST-SQUARES

Under the synthetic data set-up mentioned earlier, we first simulate the oracle world and empirical world errors curves of gradient flow dynamics using dot product kernels of the form

$$H(\boldsymbol{x}_1, \boldsymbol{x}_2) = \mathbb{E}_{\boldsymbol{w} \sim \mathbb{S}^{d-1}}[\sigma(\langle \boldsymbol{w}, \boldsymbol{x}_1 \rangle)\sigma(\langle \boldsymbol{w}, \boldsymbol{x}_2 \rangle)] \tag{6}$$

for some activation function $\sigma$. We will examine a few different choices of $f_\star$ and kernel $H$, which are specified in the descriptions of each figure.

The oracle world error is computed analytically but the empirical world errors require sampling train and test datasets. We compute empirical world errors by averaging over 10 trials. The results are visualized both in log-scale and linear-scale on the time axis. The log-scale plots allow for direct comparison with the cartoons in Fig. 3. The linear-scale plots are zoomed into the region $0 < t \leq nd^{0.4}$ since: 1) plotting the full interval squeeze all curves to the left boundary which is uninformative 2) in practice one would not optimize for very long after interpolation.

In Figs. 4a and 4b we take the target function to be a polynomial of the form

$$f_\star(\boldsymbol{x}) = a_0 \, \mathrm{He}_0(x_1) + \ldots + a_k \, \mathrm{He}_k(x_1), \quad \left\| \overline{\mathsf{P}}_j f_\star \right\|_{L^2}^2 \approx a_j^2 j!, \tag{7}$$

where $\mathrm{He}_i(t)$ is the $i$th Hermite polynomial (c.f. Appendix G.4) and the approximate equality in Eq. (7) holds in high-dimensions. In panel (4a) we consider use the ReLU activation function $\sigma(t) = \max(t, 0)$, and take $f_\star$ to be a quadratic polynomial with $(a_0, a_1, a_2) = (1/2, 1/\sqrt{2}, 1/\sqrt{8})$. With

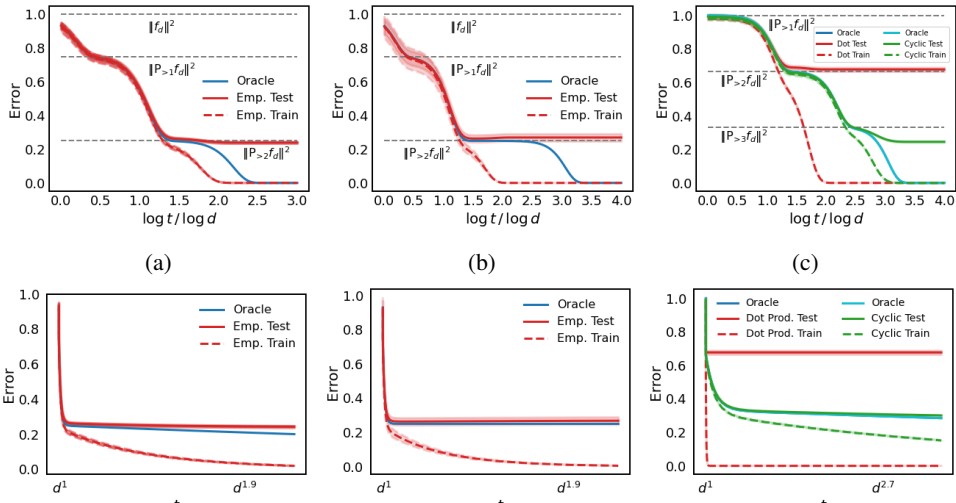

Figure 4: **Top row:** Log-scale plot of errors versus training time for dot product kernel (4a, 4b) and {dot product, cyclic} kernels in (4c). In (4a) $\sigma = \text{ReLU}$ and $(a_0, a_1, a_2) = (1/2, 1/\sqrt{2}, 1/\sqrt{8})$. In (4b), $\sigma = \text{ReLU} + 0.1\,\text{He}_3$ and $(a_0, a_1, a_2, a_3) = (1/2, 1/\sqrt{2}, 0, 1/\sqrt{24})$. In (4c), $f_\star$ is Eq. (8) and $\sigma = \text{ReLU} + 0.1\,\text{He}_3$. **Bottom row:** Same as the top row but with linear-scale time and zoomed in.

such a choice of parameters, we can see the three stages phenomenon. In panel (4b) we choose $f_\star$ to be a cubic polynomial with $(a_0, a_1, a_2, a_3) = (1/2, 1/\sqrt{2}, 0, 1/\sqrt{24})$ and $\sigma(t) = \max(t, 0) + 0.1\,\text{He}_3(t)$ (we need the third Hermite coefficient of $\sigma$ to be non-zero for stage 3 to occur). This choice of coefficients for $f_\star$ is such that $\|\overline{\mathsf{P}}_{>1} f_\star\|_{L^2}^2 \approx \|\overline{\mathsf{P}}_{>2} f_\star\|_{L^2}^2$, so that the second stage in (4b) is longer compared to (4a).

In Fig. 4c, we take the target function to be a cubic cyclic polynomial

$$f_\star(\boldsymbol{x}) = \frac{1}{\sqrt{3d}}\left(\sum_{i=1}^{d} x_i + \sum_{i=1}^{d} x_i x_{i+1} + \sum_{i=1}^{d} x_i x_{i+1} x_{i+2}\right) \tag{8}$$

where the subindex addition in $x_{i+k}$ is understood to be taken modulo $d$. We compare the performance of the dot product kernel $H$ and its invariant version $H_{\text{inv}}$ (c.f. Eq. (5)) with activation function $\sigma(t) = \max(t, 0) + 0.1\,\text{He}_3(t)$. The kernel $H_{\text{inv}}$ with $n$ samples performs equivalently to $H$ with $nd$ samples (c.f. Remark 2), but is more computationally efficient since the size of the kernel matrix is still $n \times n$. Using $H_{\text{inv}}$ elongates the first stage by a factor $d$, delaying the later stages and ensuring that the empirical world model improves longer.

Although in the simulations, the dimension $d$ is not yet high enough to see a totally sharp staircase phenomenon as in the illustrations of Fig. 3, even for this $d$ we are still able to clearly see the three predicted learning stages and deep bootstrap phenomenon across a range of settings. To better understand the effect of dimension we show similar plots with varying $d$ in Appendix F.

## 4.2 SGD FOR TWO-LAYER RANDOM-FEATURE MODELS

To more closely relate with deep learning practice and the deep bootstrap phenomenon (Nakkiran et al., 2020), we simulate the error curves of SGD training on random-feature (RF) models (i.e. two-layer networks with random first-layer weights and trainable second-layer weights), in the same synthetic data setup as before. In particular, we look at dot product RF models

$$\hat{f}_{\text{dot}}(\boldsymbol{x}; \boldsymbol{a}) = \frac{1}{\sqrt{N}}\sum_{i=1}^{N} a_i \sigma(\langle \boldsymbol{w}_i, \boldsymbol{x}\rangle), \qquad \boldsymbol{w}_i \sim_{iid} \text{Unif}(\mathbb{S}^{d-1}),$$

and cyclic invariant RF models

$$\hat{f}_{\text{cyc}}(\boldsymbol{x}; \boldsymbol{a}) = \frac{1}{\sqrt{N}}\sum_{i=1}^{N} a_i \int_{\mathcal{G}_d} \sigma(\langle \boldsymbol{w}_i, g \cdot \boldsymbol{x}\rangle)\pi_d(\mathrm{d}g), \qquad \boldsymbol{w}_i \sim_{iid} \text{Unif}(\mathbb{S}^{d-1}).$$

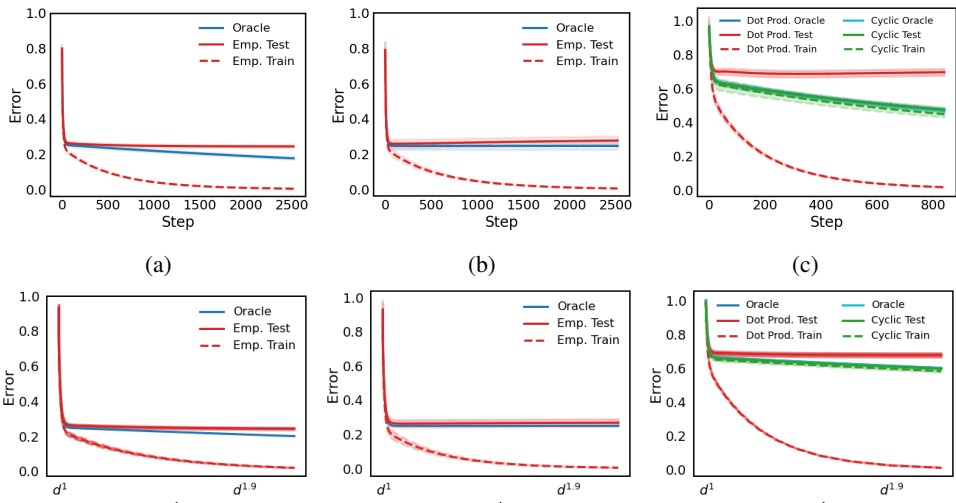

Figure 5: **Top Row:** SGD dynamics of random-feature models as described in Section 4.2. The target function and data distribution of (5a), (5b), (5c) are that of (4a), (4b), (4c) respectively. **Bottom Row:** The corresponding linear-scale errors of kernel least-squares gradient flow for comparison.

For all following experiments we take the activation $\sigma$ to be ReLU and $N = 4 \times 10^5 \approx n^{1.4}$.

For a given data distribution and RF model we train two fitted functions, one on a finite dataset (empirical world) and the other on the data distribution (oracle world). More specifically, the training of the empirical model is done using multi-pass SGD on a finite training set of size $n$ with learning rate $\eta = 0.1$ and batch size $b = 50$. The training of the oracle model is done using one-pass SGD with the same learning rate $\eta$ and batch size $b$, but at each iteration a fresh batch is sampled from the population distribution. Both models $\hat{f}_t, f_t^{\text{or}}$ are initialized with $a_i = 0$ for $i \in [N]$. To speed up and stabilize optimization we use momentum $\beta = 0.9$. Note that if we took $N \to \infty$, $\eta \to 0$, and $\beta = 0$ we would be exactly in the dot product kernel gradient flow setting.

In Fig. 5, the data generating distributions of panels (5a), (5b), (5c) are respectively the same as that of panels (4a), (4b), and (4c) from Section 4.1. The top row of Fig. 5 shows SGD for {dot product, cyclic} RF models, and the bottom row shows the corresponding gradient flow for {dot product, cyclic} kernel least-squares. We see that the corresponding curves in these two rows exhibit qualitatively the same behaviors. Additionally, the results in panel (5c) show that as predicted, even for discrete SGD dynamics the dot product RF optimizes faster but fails to generalize, whereas the invariant RF optimizes slower but generalizes better.

## 5 SUMMARY AND DISCUSSION

In this paper, we used precise asymptotics to study the oracle world and empirical world dynamics of gradient flow on kernel least-squares objectives for high-dimensional regression problems. Under reasonable conditions on the target function and kernel, we showed that in this setting there are three learning stages based on the behaviors of the empirical and oracle models and also connected our results to some empirical deep learning phenomena.

Although our setting already captures some interesting aspects of deep learning training dynamics, there are some limitations which would be interesting to resolve in future work. We require very high-dimensional data in order for the asymptotics to be accurate, but real data distributions have low-dimensional structure. We work in a limiting regime of neural network training where the dynamics are linear and the step-size is infinitesimal. It is an important direction to extend this analysis to the non-linear feature learning regime and to consider discrete step-size minibatch SGD, as these are considered important aspects of network training. Our results hold for the square-loss in regression problems, but many deep learning problems involve classification using cross-entropy loss. Lastly, our analysis holds specifically for gradient flow, so it would be also interesting to consider other iterative learning algorithms such as boosting.

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

# A    GENERAL SETTING

In this section we present our theory for training dynamics of kernel regression in an abstract setting similar to that of Mei et al. (2021a). We first introduce the setting of interest, then state the relevant assumptions, and finally we provide our theoretical results. We provide proofs of these results in Appendix B.

## A.1    PROBLEM SETUP

Consider a sequence of Polish probability spaces $(\mathcal{X}_d, \nu_d)$, where $\nu_d$ is a probability measure on the configuration space $\mathcal{X}_d$, indexed by an integer $d$. We denote by $L^2(\mathcal{X}_d) = L^2(\mathcal{X}_d, \nu_d)$ the space of square integrable functions on $(\mathcal{X}_d, \nu_d)$. For $p \geq 1$, we denote $\|f\|_{L^p(\mathcal{X}_d)} = \mathbb{E}_{\boldsymbol{x} \sim \nu_d}[|f(\boldsymbol{x})^p|]^{1/p}$ the $L^p$ norm of $f$. Let $\mathcal{D}_d \subseteq L^2(\mathcal{X}_d)$ be a closed linear subspace. In some simple applications we will consider $\mathcal{D}_d = L^2(\mathcal{X}_d)$, but the extra generality will be useful in certain applications.

We are concerned with a supervised learning problem where we are given i.i.d data $(y_i, \boldsymbol{x}_i)_{i \leq n}$. The feature vectors $\boldsymbol{x}_i \sim_{iid} \nu_d$ are in $\mathcal{X}_d$ and the empirical-valued noisy responses $y_i$ are given by

$$y_i = f_d(\boldsymbol{x}_i) + \varepsilon_i$$

for some unknown target function $f_d \in \mathcal{D}_d$ and $\varepsilon_i \sim_{iid} \mathcal{N}(0, \sigma_\varepsilon^2)$.

We consider a general RKHS defined on $(\mathcal{X}_d, \nu_d)$ via the compact self-adjoint positive definite operator $\mathbb{H}_d : \mathcal{D}_d \to \mathcal{D}_d$ which admits the representation

$$\mathbb{H}_d g(\boldsymbol{x}) = \int_{\mathcal{X}_d} H_d(\boldsymbol{x}, \boldsymbol{x}') g(\boldsymbol{x}') \nu_d(\mathrm{d}\boldsymbol{x}'),$$

where $H_d \in L^2(\mathcal{X}_d \times \mathcal{X}_d)$ with the property that $\int_{\mathcal{X}_d} H_d(\boldsymbol{x}, \boldsymbol{x}') g(\boldsymbol{x}') \nu_d(\mathrm{d}\boldsymbol{x}') = 0$ for $g \in \mathcal{D}_d^\perp$.

By the spectral theorem of compact operators, there exists an orthonormal basis $(\psi_j)_{j \geq 1}$, $\mathrm{span}(\psi_j, j \geq 1) = \mathcal{D}_d \subseteq L^2(\mathcal{X}_d)$ and empirical eigenvalues $(\lambda_{d,j})_{j \geq 1}$ with nonincreasing absolute values $|\lambda_{d,1}| \geq |\lambda_{d,2}| \geq \cdots$ and $\sum_{j \geq 1} \lambda_{d,j}^2 < \infty$ such that

$$H_d(\boldsymbol{x}_1, \boldsymbol{x}_2) = \sum_{j=1}^{\infty} \lambda_{d,j}^2 \psi_j(\boldsymbol{x}_1) \psi_j(\boldsymbol{x}_2)$$

where convergence holds in $L^2(\mathcal{X}_d \times \mathcal{X}_d)$.

For $S \subseteq \{1, 2, \ldots\}$ we denote $\mathsf{P}_S$ to be the projection operator from $L^2(\mathcal{X}_d)$ onto $\mathcal{D}_{d,S} := \mathrm{span}(\psi_j, j \in S)$. We denote $\mathbb{H}_{d,S}$ to be the operator

$$\mathbb{H}_{d,S} = \sum_{j=1}^{\infty} \lambda_{d,j}^2 \psi_j \psi_j^\star$$

and $H_{d,S}$ the corresponding kernel.

If $S = \{j \in \mathbb{N} : j \leq \ell\}$ we will write as short-hand $\mathbb{H}_{d, \leq \ell}$ and analogously for $S = \{j \in \mathbb{N} : j > \ell\}$. The trace of this operator is given by

$$\mathrm{Tr}(\mathbb{H}_{d,S}) \equiv \sum_{j \in S} \lambda_{d,j}^2 = \mathbb{E}_{\boldsymbol{x} \sim \nu_d}[H_{d,S}(\boldsymbol{x}, \boldsymbol{x})] < \infty.$$

Define the test error $R : L^2(\mathcal{X}_d) \to \mathbb{R}$ and training error $\widehat{R}_n : L^2(\mathcal{X}_d) \to \mathbb{R}$

$$R(f) \equiv \mathbb{E}_{(\boldsymbol{x}_{\text{new}}, y_{\text{new}})}\{(y_{\text{new}} - f(\boldsymbol{x}_{\text{new}}))^2\}, \qquad \widehat{R}_n(f) \equiv \frac{1}{n} \sum_{i=1}^{n} (y_i - f(\boldsymbol{x}_i))^2 \qquad (9)$$

where $(\boldsymbol{x}_1, y_1), \ldots, (\boldsymbol{x}_n, y_n), (\boldsymbol{x}_{\text{new}}, y_{\text{new}})$ are iid. For a kernel $H_d \in L^2(\mathcal{X}_d \times \mathcal{X}_d)$ we will consider the oracle model $f_t^{\text{or}}$ and the empirical model $\hat{f}_t$ which satisfy the following gradient flows

$$\frac{\mathrm{d}}{\mathrm{d}t} f_t^{\text{or}}(\boldsymbol{x}) = -\boldsymbol{\nabla} R(f_t^{\text{or}}(\boldsymbol{x})) = \mathbb{E}_{\boldsymbol{z} \sim \nu_d}[H_d(\boldsymbol{x}, \boldsymbol{z})(f_d(\boldsymbol{z}) - f_t^{\text{or}}(\boldsymbol{z}))], \qquad (10)$$

$$\frac{\mathrm{d}}{\mathrm{d}t} \hat{f}_t(\boldsymbol{x}) = -\boldsymbol{\nabla} \widehat{R}_n(\hat{f}_t(\boldsymbol{x})) = \frac{1}{n} \sum_{i=1}^{n} H_d(\boldsymbol{x}, \boldsymbol{x}_i)(y_i - \hat{f}_t(\boldsymbol{x}_i)). \qquad (11)$$

## A.2 General Assumptions

We now state our assumptions on the kernel and the sequence of probability spaces $(\mathcal{X}_d, \nu_d)$.

**Assumption 1** ($\{n(d), \mathsf{m}(d)\}_{d \geq 1}$-Kernel Concentration Property). *We say that the sequence of operators $\{\mathbb{H}_d\}_{d \geq 1}$ satisfies the Kernel Concentration Property (KCP) with respect to the sequence $\{n(d), \mathsf{m}(d)\}_{d \geq 1}$ if there exists a sequence of integers $\{r(d)\}_{d \geq 1}$ with $r(d) \geq \mathsf{m}(d)$, such that the following conditions hold.*

(a) *(Hypercontractivity of finite eigenspaces.) For any fixed $q \geq 1$, there exists a constant $C$ such that for any $h \in \mathcal{D}_{d, \leq r(d)} = \mathrm{span}(\psi_s, 1 \leq s \leq r(d))$, we have*

$$\|h\|_{L^{2q}} \leq C \|h\|_{L^2}.$$

(b) *(Properly decaying eigenvalues) There exists fixed $\delta_0 > 0$, such that, for all $d$ large enough,*

$$n(d)^{2+\delta_0} \leq \frac{(\sum_{j=r(d)+1}^{\infty} \lambda_{d,j}^4)^2}{\sum_{j=r(d)+1}^{\infty} \lambda_{d,j}^8},$$

$$n(d)^{2+\delta_0} \leq \frac{(\sum_{j=r(d)+1}^{\infty} \lambda_{d,j}^2)^2}{\sum_{j=r(d)+1}^{\infty} \lambda_{d,j}^4}.$$

(c) *(Concentration of diagonal elements of kernel) For $(\boldsymbol{x}_i)_{i \in [n(d)]} \sim_{iid} \nu_d$, we have:*

$$\max_{i \in [n(d)]} \left| \mathbb{E}_{\boldsymbol{x} \sim \nu_d}[H_{d, > \mathsf{m}(d)}(\boldsymbol{x}_i, \boldsymbol{x})^2] - \mathbb{E}_{\boldsymbol{x}, \boldsymbol{x}' \sim \nu_d}[H_{d, > \mathsf{m}(d)}(\boldsymbol{x}, \boldsymbol{x}')^2] \right| = o_{d, \mathbb{P}}(1) \cdot \mathbb{E}_{\boldsymbol{x}, \boldsymbol{x}' \sim \nu_d}[H_{d, > \mathsf{m}(d)}(\boldsymbol{x}, \boldsymbol{x}')^2],$$

$$\max_{i \in [n(d)]} \left| H_{d, > \mathsf{m}(d)}(\boldsymbol{x}_i, \boldsymbol{x}_i) - \mathbb{E}_{\boldsymbol{x}}[H_{d, > \mathsf{m}(d)}(\boldsymbol{x}, \boldsymbol{x})] \right| = o_{d, \mathbb{P}}(1) \cdot \mathbb{E}_{\boldsymbol{x}}[H_{d, > \mathsf{m}(d)}(\boldsymbol{x}, \boldsymbol{x})].$$

Assumption 1(a) can be interpreted as requiring that the top eigenfunctions of $\mathbb{H}_d$ are delocalized. Assumption 1(b) concerns the tail of eigenvalues of $\mathbb{H}_d$ and is a mild assumption in high-dimensions. Lastly, 1(c) essentially requires that "most points" in $\mathcal{X}_d$ behave similarly in the sense of having similar values of the kernel diagonal $H_d(\boldsymbol{x}, \boldsymbol{x})$.

**Assumption 2** (Eigenvalue condition at level $\{(n(d), \mathsf{m}(d))\}_{d \geq 1}$). *We say that the sequence of kernel operators $\{\mathbb{H}_d\}_{d \geq 1}$ satisfies the Eigenvalue Condition at level $\{(n(d), \mathsf{m}(d))\}_{d \geq 1}$ if the following conditions hold for all $d$ large enough*

(a) *There exists a fixed $\delta_0 > 0$, such that*

$$n(d)^{1+\delta_0} \leq \frac{1}{\lambda_{d, \mathsf{m}(d)+1}^4} \sum_{k=\mathsf{m}(d)+1}^{\infty} \lambda_{d,k}^4, \tag{12}$$

$$n(d)^{1+\delta_0} \leq \frac{1}{\lambda_{d, \mathsf{m}(d)+1}^2} \sum_{k=\mathsf{m}(d)+1}^{\infty} \lambda_{d,k}^2. \tag{13}$$

(b) *There exists a fixed $\delta_0 > 0$, such that*

$$n(d)^{1-\delta_0} \geq \frac{1}{\lambda_{d, \mathsf{m}(d)}^2} \sum_{k=\mathsf{m}(d)+1}^{\infty} \lambda_{d,k}^2.$$

(c) *There exists a fixed $\delta_0 > 0$, such that*

$$\mathsf{m}(d) \leq n(d)^{1-\delta_0}.$$

Assumptions 2(a) and 2(b) can be seen as a spectral gap assumption. This ensures a clear separation between the eigenvalues in the subspace $\mathcal{D}_{d, \leq \mathsf{m}(d)}$ and the subspace $\mathcal{D}_{d, > \mathsf{m}(d)}$. The technical requirement 2(c) is mild.

In the asymptotic setting, we will be interested in understanding the model learned at a time $t = t(d)$ which scales with the dimension. The next set of assumptions give requirements for a valid scaling.

**Assumption 3** (Admissible Time at $\{t(d), \mathsf{u}(d), n(d), \mathsf{m}(d)\}_{d\geq 1}$)**.** *We say that the sequence $\{t(d), \mathsf{u}(d), n(d), \mathsf{m}(d)\}_{d\geq 1}$ is an Admissible Time for the sequence of kernel operators $\{\mathbb{H}_d\}_{d\geq 1}$ if the following conditions hold*

(a) *There exists a fixed $\delta_0 > 0$, such that for $d$ large enough*

$$\frac{1}{\lambda_{d,\mathsf{u}(d)}^2} \leq t(d)^{1-\delta_0} \leq t(d)^{1+\delta_0} \leq \frac{1}{\lambda_{d,\mathsf{u}(d)+1}^2}.$$

(b) *If $\mathsf{u}(d) < \mathsf{m}(d)$ for infinitely many $d$, then*

$$\frac{t(d)}{n(d)} \sum_{k=\mathsf{m}(d)+1}^{\infty} \lambda_{d,k}^2 = o_d(1).$$

(c) *There exists a constant $C$ such that*

$$\sum_{k=1}^{\mathsf{m}(d)} \lambda_{d,k}^2 \leq C \sum_{k=\mathsf{m}(d)+1}^{\infty} \lambda_{d,k}^2.$$

Assumption 3(a) is similar to the spectral gap condition Assumption 2(a), 2(b) for $(t(d), \mathsf{u}(d))_{d\geq 1}$. Assumption 3(b) relates the ordering of the indices $\mathsf{u}(d), \mathsf{m}(d)$ to the relative growth of $t(d), n(d)$. Assumption 3(c) requires that the eigenvalue tail does not decay too abruptly.

## A.3 MAIN RESULTS

In this section we give the main theoretical results. Recall the problem set-up and notation from Appendix A.1. We will characterize the gradient flow dynamics dynamics of the oracle model $f_t^{\mathrm{or}}$ Eq. (10) and the empirical model $\hat{f}_t$ Eq. (11) for a general kernel $H_d$.

**Theorem 3** (Oracle World)**.** *Let $\{f_d \in \mathcal{D}_d\}_{d\geq 1}$ be a sequence of functions and $\{\mathbb{H}_d\}_{d\geq 1}$ be a sequence of kernel operators such that $\{(\mathbb{H}_d, t(d), \mathsf{u}(d))\}_{d\geq 1}$ satisfies Assumption 3(a), then*

$$R(f_t^{\mathrm{or}}) = \left\|\mathsf{P}_{>\mathsf{u}(d)} f_d\right\|_{L^2}^2 + \sigma_\varepsilon^2 + o_d(1) \cdot \|f_d\|_{L^2}^2,$$

$$\left\|f_t^{\mathrm{or}} - \mathsf{P}_{\leq\mathsf{u}(d)} f_d\right\|_{L^2}^2 = o_d(1) \cdot \|f_d\|_{L^2}^2,$$

*where $\mathsf{P}_{\leq\mathsf{u}(d)}$ and $\mathsf{P}_{>\mathsf{u}(d)}$ are the projection operators onto the subspace spanned by the top $\mathsf{u}(d)$ kernel eigenfunctions and the orthogonal complement respectively, as defined in Appendix A.1.*

The error of the oracle model is determined solely by optimization time $t$ through $\mathsf{u}(d)$. Due to the spectral gap assumption 3(a) learning only occurs along the top $\mathsf{u}(d)$ eigenfunctions.

The next results describe the empirical model. First we characterize the training error.

**Theorem 4** (Empirical World - Train)**.** *Let $\{f_d \in \mathcal{D}_d\}_{d\geq 1}$ be a sequence of functions, $(\boldsymbol{x}_i)_{i\in[n(d)]} \sim \nu_d$ independently, and $\{\mathbb{H}_d\}_{d\geq 1}$ be a sequence of kernel operators such that $\{(\mathbb{H}_d, n(d), \mathsf{m}(d), t(d), \mathsf{u}(d))\}_{d\geq 1}$ satisfies $\{(n(d), \mathsf{m}(d))\}_{d\geq 1}$-KPCP (Assumption 1), eigenvalue condition at level $\{(n(d), \mathsf{m}(d))\}_{d\geq 1}$ (Assumption 2), and $\{(n(d), \mathsf{m}(d), t(d), \mathsf{u}(d))\}_{d\geq 1}$ is a valid time (Assumption 3). Define $\kappa_H = \mathrm{Tr}(\mathbb{H}_{d,>\mathsf{m}(d)})$ and $\ell(d) = \min\{\mathsf{u}(d), \mathsf{m}(d)\}$. Then for any $\eta > 0$ we have,*

$$\widehat{R}_n(\hat{f}_t) = \left\|\mathsf{P}_{>\ell(d)} f_d\right\|_{L^2}^2 + \sigma_\varepsilon^2 + o_{d,\mathbb{P}}(1) \cdot (\|f_d\|_{L^{2+\eta}}^2 + \sigma_\varepsilon^2), \qquad \text{if } t = o_d(n/\kappa_H),$$

$$\widehat{R}_n(\hat{f}_t) = o_{d,\mathbb{P}}(1) \cdot (\|f_d\|_{L^{2+\eta}}^2 + \sigma_\varepsilon^2), \qquad \text{if } t = \omega_d(n/\kappa_H).$$

In the early-time regime $t \ll n/\kappa_H$ the training error may be non-zero and matches the oracle world error if also $\mathsf{u}(d) \leq \mathsf{m}(d)$. In the late-time regime $t \gg n/\kappa_H$ the training error is negligible and the model interpolates the training set. The quantity $\kappa_H$ arises since the empirical kernel matrix can be decomposed as $\boldsymbol{H} = \boldsymbol{H}_{\leq\mathsf{m}} + \boldsymbol{H}_{>\mathsf{m}}$ and the second component is approximately a multiple of the identity: $\boldsymbol{H}_{>\mathsf{m}} \approx \mathrm{Tr}(\boldsymbol{H}_{>\mathsf{m}}) \cdot \mathbf{I}_n = \kappa_H \cdot \mathbf{I}_n$. This term acts as a self-induced ridge-regularizer.

Our final result characterizes the test error of the empirical model.

**Theorem 5** (Empirical World - Test). *Let $\{f_d \in \mathcal{D}_d\}_{d \geq 1}$ be a sequence of functions, $(\boldsymbol{x}_i)_{i \in [n(d)]} \sim \nu_d$ independently, and $\{\mathbb{H}_d\}_{d \geq 1}$ be a sequence of kernel operators such that $\{(\mathbb{H}_d, n(d), \mathsf{m}(d), t(d), \mathsf{u}(d))\}_{d \geq 1}$ satisfies $\{(n(d), \mathsf{m}(d))\}_{d \geq 1}$-KPCP (Assumption 1), eigenvalue condition at level $\{(n(d), \mathsf{m}(d))\}_{d \geq 1}$ (Assumption 2), and $\{(n(d), \mathsf{m}(d), t(d), \mathsf{u}(d))\}_{d \geq 1}$ is a valid time (Assumption 3). Define $\ell(d) = \min\{\mathsf{u}(d), \mathsf{m}(d)\}$. Then for any $\eta > 0$ we have,*

$$R(\hat{f}_t) = \left\| \mathsf{P}_{>\ell(d)} f_d \right\|_{L^2}^2 + \sigma_\varepsilon^2 + o_{d,\mathbb{P}}(1) \cdot (\|f_d\|_{L^{2+\eta}}^2 + \sigma_\varepsilon^2),$$

$$\left\| \hat{f}_t - \mathsf{P}_{\leq \ell(d)} f_d \right\|_{L^2}^2 = o_{d,\mathbb{P}}(1) \cdot (\|f_d\|_{L^{2+\eta}}^2 + \sigma_\varepsilon^2).$$

The result shows that the empirical model is essentially the projection of the regression function onto the first $\ell(d)$ eigenfunctions. The quantity $\ell(d)$ controls the complexity of the model which increases with time $t$ up until $\mathsf{u}(d) \geq \mathsf{m}(d)$ after which the complexity is limited by $n$.

## B   PROOF OF GENERAL SETTING

### B.1   ORACLE WORLD - PROOF OF THEOREM 3

The oracle model ODE Eq. (10) with initialization $f_0^{\mathrm{or}} \equiv 0$ can be solved (c.f. Appendix E.1) to yield the solution

$$f_t^{\mathrm{or}} = f_d - e^{-t\mathbb{H}_d} f_d.$$

Therefore the excess risk is given by

$$R(f_t^{\mathrm{or}}) - \sigma_\varepsilon^2 = \mathbb{E}_{\boldsymbol{x} \sim \nu_d}(f_d(\boldsymbol{x}) - f_t^{\mathrm{or}}(\boldsymbol{x}))^2 = \int_{\mathcal{X}_d} f_d(\boldsymbol{x}) e^{-2t\mathbb{H}_d} f_d(\boldsymbol{x}) \nu_d(\mathrm{d}\boldsymbol{x}) = \sum_{k=0}^{\infty} \exp\big(-2t\lambda_{d,k}^2\big) \hat{f}_k^2.$$

where $\lambda_{d,k}^2$ are the kernel eigenvalues and $\hat{f}_k := \langle f_d, \psi_k \rangle_{L^2}$ are the Fourier coefficients of $f_d$ in the kernel eigenbasis (c.f. Appendix A.1). We can control this quantity as follows,

$$|R(f_t^{\mathrm{or}}) - \sigma_\varepsilon^2 - \|\mathsf{P}_{>\mathsf{u}} f_d\|_{L^2}^2 |/\|f_d\|_{L^2}^2 \leq \max\left\{ \max_{k \leq \mathsf{u}} \exp\big(-2t\lambda_{d,k}^2\big), \max_{k \geq \mathsf{u}+1} 1 - \exp\big(-2t\lambda_{d,k}^2\big)\right\}$$

$$\leq \max\left\{ \max_{k \leq \mathsf{u}} \exp\big(-\Omega(t\lambda_{d,k}^2)\big), \max_{k \geq \mathsf{u}+1} O(t\lambda_{d,k}^2)\right\}$$

$$\leq \max\left\{ \exp\big(-\Omega(t^{\delta_0})\big), O(t^{-\delta_0})\right\} = o_d(1)$$

where the last inequality follows from Assumption 3(a). This shows the first theorem statement,

$$\|f_t^{\mathrm{or}} - \mathsf{P}_{\leq \mathsf{u}} f_d\|_{L^2}^2 = o_d(1) \cdot \|f_d\|_{L^2}^2.$$

Now observe that

$$\mathsf{P}_{\leq u} f_d - f_t^{\mathrm{or}} = \sum_{k=0}^{\mathsf{u}} e^{-t\lambda_{d,k}^2} \hat{f}_k \psi_k - \sum_{k \geq \mathsf{u}+1} (1 - e^{-t\lambda_{d,k}^2}) \hat{f}_k \psi_k$$

hence

$$\|f_t^{\mathrm{or}} - \mathsf{P}_{\leq u} f_d\|_{L^2}^2 / \|f_d\|_{L^2}^2 \leq \max\left\{ \max_{k \leq \mathsf{u}} \exp\big(-2t\lambda_{d,k}^2\big), \max_{k \geq \mathsf{u}+1} (1 - e^{-t\lambda_{d,k}^2})^2\right\}$$

$$\leq \max\left\{ \max_{k \leq \mathsf{u}} \exp\big(-2t\lambda_{d,k}^2\big), \max_{k \geq \mathsf{u}+1} 1 - \exp\big(-2t\lambda_{d,k}^2\big)\right\} = o_d(1)$$

where the second inequality follows from the fact that $(1 - e^{-x})^2 \leq 1 - e^{-2x}$ and the final equality is from the proof of the first part of the theorem. Thus,

$$\|f_t^{\mathrm{or}} - \mathsf{P}_{\leq u} f_d\|_{L^2}^2 = o_d(1) \cdot \|f_d\|_{L^2}^2$$

completing the proof.

### B.2   EMPIRICAL WORLD – PRELIMINARIES

We will introduce some useful notations and the objects of analysis for studying the empirical world.

#### B.2.1   TRAINING DYNAMICS

For the training of the empirical world c.f. Eq. (11), if $\boldsymbol{u}(t) = (\hat{f}_t(\boldsymbol{x}_1), \ldots, \hat{f}_t(\boldsymbol{x}_n)) \in \mathbb{R}^n$ then from Eq. (44) if $\boldsymbol{u}(0) = \boldsymbol{0}$ then

$$\boldsymbol{u}(t) = \boldsymbol{y} - e^{-t\boldsymbol{H}/n}\boldsymbol{y} = (\mathbf{I}_n - e^{-t\boldsymbol{H}/n})\boldsymbol{y}, \tag{14}$$

where $\boldsymbol{H} \in \mathbb{R}^{n \times n}$ with the $(i, j)$th element given by $H_{ij} = H_d(\boldsymbol{x}_i, \boldsymbol{x}_j)$ and $\boldsymbol{y} = \boldsymbol{f} + \boldsymbol{\varepsilon} \in \mathbb{R}^n$ with $\boldsymbol{f} = (f_d(\boldsymbol{x}_1), \ldots, f_d(\boldsymbol{x}_n))^{\mathsf{T}} \in \mathbb{R}^n$ and $\boldsymbol{\varepsilon} = (\varepsilon_1, \ldots, \varepsilon_n)^{\mathsf{T}} \in \mathbb{R}^n$.

For $\boldsymbol{x} \in \mathbb{R}^d$, define $\boldsymbol{h}(\boldsymbol{x}) = (H_d(\boldsymbol{x}_1, \boldsymbol{x}), \ldots, H_d(\boldsymbol{x}_n, \boldsymbol{x}))^\mathsf{T} \in \mathbb{R}^n$. The training and test errors as defined in Eq. (1) can be written as

$$\widehat{R}_n(\hat{f}_t) = \frac{1}{n}\|\boldsymbol{u}(t) - \boldsymbol{y}\|_2^2 = \frac{1}{n}\boldsymbol{y}^\mathsf{T} e^{-2t\boldsymbol{H}/n}\boldsymbol{y},$$

$$R(\hat{f}_t) = \mathbb{E}_{\boldsymbol{x}}\big[(f_d(\boldsymbol{x}) - \boldsymbol{u}(t)^\mathsf{T}\boldsymbol{H}^{-1}\boldsymbol{h}(\boldsymbol{x}))^2\big].$$

Expanding $R(\hat{f}_t)$ yields

$$R(\hat{f}_t) = \mathbb{E}_{\boldsymbol{x}}[f_d(\boldsymbol{x})^2] - 2\boldsymbol{u}(t)^\mathsf{T}\boldsymbol{H}^{-1}\boldsymbol{E} + \boldsymbol{u}(t)^\mathsf{T}\boldsymbol{H}^{-1}\boldsymbol{M}\boldsymbol{H}^{-1}\boldsymbol{u}(t) \tag{15}$$

where $\boldsymbol{E} = (E_1, \ldots, E_n)^\mathsf{T} \in \mathbb{R}^n$, $\boldsymbol{M} = (M_{ij})_{i,j\in[n]} \in \mathbb{R}^{n\times n}$, and $\boldsymbol{H} = (H_{ij})_{i,j\in[n]} \in \mathbb{R}^{n\times n}$ with

$$E_i = \mathbb{E}_{\boldsymbol{x}}[f_d(\boldsymbol{x})H_d(\boldsymbol{x}, \boldsymbol{x}_i)],$$
$$M_{ij} = \mathbb{E}_{\boldsymbol{x}}[H_d(\boldsymbol{x}, \boldsymbol{x}_i)H_d(\boldsymbol{x}, \boldsymbol{x}_j)],$$
$$H_{ij} = H_d(\boldsymbol{x}_i, \boldsymbol{x}_j).$$

### B.2.2 DECOMPOSITIONS AND NOTATIONS

In this section we recall some useful decompositions of empirical quantities from Mei et al. (2021a). As mentioned earlier the eigendecomposition of $H_d$ is given by

$$H_d(\boldsymbol{x}, \boldsymbol{y}) = \sum_{k=1}^{\infty} \lambda_{d,k}^2 \psi_k(\boldsymbol{x})\psi_k(\boldsymbol{y}).$$

We write the orthogonal decomposition of $f_d$ in the basis $\{\psi_k\}_{k\geq 1}$ as

$$f_d(\boldsymbol{x}) = \sum_{k=1}^{\infty} \hat{f}_{d,k}\psi_k(\boldsymbol{x}).$$

Define

$$\boldsymbol{\psi}_k = (\psi_k(\boldsymbol{x}_1), \ldots, \psi_k(\boldsymbol{x}_n))^\mathsf{T} \in \mathbb{R}^n,$$
$$\boldsymbol{D}_{\leq \mathsf{m}} = \mathrm{diag}(\lambda_{d,1}, \lambda_{d,2}, \ldots, \lambda_{d,\mathsf{m}}) \in \mathbb{R}^{\mathsf{m}\times\mathsf{m}},$$
$$\boldsymbol{\Psi}_{\leq \mathsf{m}} = (\psi_k(\boldsymbol{x}_i))_{i\in[n], k\in[\mathsf{m}]} \in \mathbb{R}^{n\times\mathsf{m}},$$
$$\widehat{\boldsymbol{f}}_{\leq \mathsf{m}} = (\hat{f}_{d,1}, \hat{f}_{d,2}, \ldots, \hat{f}_{d,\mathsf{m}})^\mathsf{T} \in \mathbb{R}^{\mathsf{m}}.$$

We have the following orthogonal basis decompositions of $\boldsymbol{f}, \boldsymbol{H}, \boldsymbol{E}$ and $\boldsymbol{M}$

$$\boldsymbol{f} = \boldsymbol{f}_{\leq\mathsf{m}} + \boldsymbol{f}_{>\mathsf{m}}, \qquad \boldsymbol{f}_{\leq\mathsf{m}} = \boldsymbol{\Psi}_{\leq\mathsf{m}}\widehat{\boldsymbol{f}}_{\leq\mathsf{m}}, \qquad \boldsymbol{f}_{>\mathsf{m}} = \sum_{k=\mathsf{m}+1}^{\infty} \hat{f}_{d,k}\boldsymbol{\psi}_k,$$

$$\boldsymbol{H} = \boldsymbol{H}_{\leq\mathsf{m}} + \boldsymbol{H}_{>\mathsf{m}}, \quad \boldsymbol{H}_{\leq\mathsf{m}} = \boldsymbol{\Psi}_{\leq\mathsf{m}}\boldsymbol{D}_{\leq\mathsf{m}}^2\boldsymbol{\Psi}_{\leq\mathsf{m}}^\mathsf{T}, \quad \boldsymbol{H}_{>\mathsf{m}} = \sum_{k=\mathsf{m}+1}^{\infty} \lambda_{d,k}^2\boldsymbol{\psi}_k\boldsymbol{\psi}_k^\mathsf{T},$$

$$\boldsymbol{E} = \boldsymbol{E}_{\leq\mathsf{m}} + \boldsymbol{E}_{>\mathsf{m}}, \qquad \boldsymbol{E}_{\leq\mathsf{m}} = \boldsymbol{\Psi}_{\leq\mathsf{m}}\boldsymbol{D}_{\leq\mathsf{m}}^2\widehat{\boldsymbol{f}}_{\leq\mathsf{m}}, \quad \boldsymbol{E}_{>\mathsf{m}} = \sum_{k=\mathsf{m}+1}^{\infty} \lambda_{d,k}^2\hat{f}_{d,k}\boldsymbol{\psi}_k, \tag{16}$$

$$\boldsymbol{M} = \boldsymbol{M}_{\leq\mathsf{m}} + \boldsymbol{M}_{>\mathsf{m}}, \quad \boldsymbol{M}_{\leq\mathsf{m}} = \boldsymbol{\Psi}_{\leq\mathsf{m}}\boldsymbol{D}_{\leq\mathsf{m}}^4\boldsymbol{\Psi}_{\leq\mathsf{m}}^\mathsf{T}, \quad \boldsymbol{M}_{>\mathsf{m}} = \sum_{k=\mathsf{m}+1}^{\infty} \lambda_{d,k}^4\boldsymbol{\psi}_k\boldsymbol{\psi}_k^\mathsf{T}.$$

By Lemma 6 below, under Assumptions 1 and 2(a) the matrices $\boldsymbol{H}$ and $\boldsymbol{M}$ can be written as

$$\boldsymbol{H} = \boldsymbol{\Psi}_{\leq\mathsf{m}}\boldsymbol{D}_{\leq\mathsf{m}}^2\boldsymbol{\Psi}_{\leq\mathsf{m}}^\mathsf{T} + \kappa_H(\mathbf{I}_n + \boldsymbol{\Delta}_H), \tag{17}$$

$$\boldsymbol{M} = \boldsymbol{\Psi}_{\leq\mathsf{m}}\boldsymbol{D}_{\leq\mathsf{m}}^4\boldsymbol{\Psi}_{\leq\mathsf{m}}^\mathsf{T} + \kappa_M(\mathbf{I}_n + \boldsymbol{\Delta}_M), \tag{18}$$

where

$$\kappa_H = \text{Tr}(\mathbb{H}_{d,>\mathsf{m}}) = \sum_{k \geq \mathsf{m}+1}^{\infty} \lambda_{d,k}^2,$$

$$\kappa_M = \text{Tr}(\mathbb{H}_{d,>\mathsf{m}}^2) = \sum_{k \geq \mathsf{m}+1}^{\infty} \lambda_{d,k}^4,$$

and

$$\max\{\|\mathbf{\Delta}_H\|_{\text{op}}, \|\mathbf{\Delta}_M\|_{\text{op}}\} = o_{d,\mathbb{P}}(1).$$

We will use $\alpha$ as shorthand for the scalar valued dimension dependent quantity $e^{-(t/n)\kappa_H}$ and take

$$\boldsymbol{K}_{\leq \mathsf{m}} := \mathbf{I}_n - \alpha e^{-(t/n)\boldsymbol{\Psi}_{\leq \mathsf{m}} \boldsymbol{D}_{\leq \mathsf{m}}^2 \boldsymbol{\Psi}_{\leq \mathsf{m}}^\mathsf{T}}. \tag{19}$$

We also introduce the shrinkage matrix defined as

$$\boldsymbol{S}_{\leq \mathsf{m}} = \left(\mathbf{I}_{\mathsf{m}} + \frac{\kappa_H}{n} \boldsymbol{D}_{\leq \mathsf{m}}^{-2}\right)^{-1} = \text{diag}((s_j)_{j \in [\mathsf{m}]}) \in \mathbb{R}^{\mathsf{m} \times \mathsf{m}}, \quad \text{where } s_j = \frac{\lambda_{d,j}^2}{\lambda_{d,j}^2 + \frac{\kappa_H}{n}}. \tag{20}$$

If unspecified we will typically use $\mathbf{\Delta}, \mathbf{\Delta}'$, etc. to denote matrices with operator norm $o_{d,\mathbb{P}}(1)$. For positive integers $\ell < \mathsf{m}$, define $[\ell, \mathsf{m}] = \{\ell + 1, \ldots, \mathsf{m}\}$. We will use the following notation

$$\boldsymbol{S}_{\ell\mathsf{m}} = \text{diag}((s_j)_{j \in [\ell,\mathsf{m}]}) \qquad \text{for } s_j \text{ defined in Eq. (20)}$$

$$\boldsymbol{\Psi}_{\ell\mathsf{m}} = (\psi_k(\boldsymbol{x}_i))_{i \in [n], k \in [\ell,\mathsf{m}]} \in \mathbb{R}^{n \times (\mathsf{m}-\ell)}$$

$$\boldsymbol{f}_{\ell\mathsf{m}} = \sum_{k \in [\ell,\mathsf{m}]} \hat{f}_{d,k} \psi_k$$

### B.2.3 Auxiliary Lemmas

Here we collect some lemmas which will be of use to us.

**Lemma 1** (Matrix Exponential Perturbation Inequality). *For matrix operator norm $\| \cdot \|$, if $\boldsymbol{A}, \boldsymbol{B} \in \mathbb{R}^{n \times n}$ are symmetric then*

$$\|e^{\boldsymbol{A}} - e^{\boldsymbol{B}}\| \leq \|\boldsymbol{A} - \boldsymbol{B}\| \max\{\|e^{\boldsymbol{A}}\|, \|e^{\boldsymbol{B}}\|\}.$$

*For general square matrices $\boldsymbol{A}, \boldsymbol{B} \in \mathbb{R}^{n \times n}$ we have*

$$\|e^{\boldsymbol{A}} - e^{\boldsymbol{B}}\| \leq \|\boldsymbol{A} - \boldsymbol{B}\| e^{\|\boldsymbol{A}\|} e^{\|\boldsymbol{B}\|}.$$

**Lemma 2.** *Let $\boldsymbol{Z} \in \mathbb{R}^{m \times n}, \boldsymbol{\Psi} \in \mathbb{R}^{m \times p}, \boldsymbol{D} \in \mathbb{R}^{p \times p}$ and $t \in \mathbb{R}$. Denote $\boldsymbol{A} = \boldsymbol{Z}^\mathsf{T} \boldsymbol{\Psi} \in \mathbb{R}^{n \times p}$ and $\boldsymbol{B} = \boldsymbol{\Psi}^\mathsf{T} \boldsymbol{\Psi} \in \mathbb{R}^{p \times p}$. Then,*

$$\boldsymbol{Z}^\mathsf{T} e^{t\boldsymbol{\Psi} \boldsymbol{D} \boldsymbol{\Psi}^\mathsf{T}} \boldsymbol{\Psi} = \boldsymbol{A} e^{t\boldsymbol{D}\boldsymbol{B}}.$$

The notations in Lemmas 3-8 all follow the notations given in Section B.2.2.

**Lemma 3** (Lemma 12 from Mei et al. (2021a) with $\lambda = 0$). *Let Assumptions 1 and 2 hold. Then,*

$$\left\|n\boldsymbol{H}^{-1}\boldsymbol{M}\boldsymbol{H}^{-1} - \boldsymbol{\Psi}_{\leq \mathsf{m}} \boldsymbol{S}_{\leq \mathsf{m}}^2 \boldsymbol{\Psi}_{\leq \mathsf{m}}^\mathsf{T}/n\right\|_{\text{op}} = o_{d,\mathbb{P}}(1).$$

**Lemma 4** (Theorem 6(b) from Mei et al. (2021a)). *Let Assumptions 1(a), 2(c) hold. Then,*

$$\left\|\boldsymbol{\Psi}_{\leq \mathsf{m}}^\mathsf{T} \boldsymbol{\Psi}_{\leq \mathsf{m}}/n - \mathbf{I}_\mathsf{m}\right\|_{\text{op}} = o_{d,\mathbb{P}}(1).$$

**Lemma 5** (Lemma 13 from Mei et al. (2021a) with $\lambda = 0$). *Let Assumptions 1 and 2 hold. Then,*

$$\left\|\boldsymbol{\Psi}_{\leq \mathsf{m}}^\mathsf{T} \boldsymbol{H}^{-1} \boldsymbol{\Psi}_{\leq \mathsf{m}} \boldsymbol{D}_{\leq \mathsf{m}}^2 - \boldsymbol{S}_{\leq \mathsf{m}}\right\|_{\text{op}} = o_{d,\mathbb{P}}(1).$$

**Lemma 6** (Theorem 6 from Mei et al. (2021a)). *Let Assumptions 1 and 2(a) hold. Then we can decompose the kernel matrices as follows*

$$\boldsymbol{H} = \boldsymbol{\Psi}_{\leq \mathsf{m}} \boldsymbol{D}_{\leq \mathsf{m}}^2 \boldsymbol{\Psi}_{\leq \mathsf{m}}^\mathsf{T} + \kappa_H(\mathbf{I}_n + \mathbf{\Delta}_H),$$

$$\boldsymbol{M} = \boldsymbol{\Psi}_{\leq \mathsf{m}} \boldsymbol{D}_{\leq \mathsf{m}}^4 \boldsymbol{\Psi}_{\leq \mathsf{m}}^\mathsf{T} + \kappa_M(\mathbf{I}_n + \mathbf{\Delta}_M),$$

*where*

$$\kappa_H = \text{Tr}(\mathbb{H}_{d,>\mathsf{m}}) = \sum_{k \geq \mathsf{m}+1}^{\infty} \lambda_{d,k}^2,$$

$$\kappa_M = \text{Tr}(\mathbb{H}_{d,>\mathsf{m}}^2) = \sum_{k \geq \mathsf{m}+1}^{\infty} \lambda_{d,k}^4,$$

*and*

$$\max\{\|\boldsymbol{\Delta}_H\|_{\text{op}}, \|\boldsymbol{\Delta}_M\|_{\text{op}}\} = o_{d,\mathbb{P}}(1).$$

**Lemma 7.** *Let Assumption 1(a) hold. Let $S, T$ be disjoint subsets of $\mathbb{N}$. Let $\boldsymbol{D} \in \mathbb{R}^{|S| \times |S|}$ be a diagonal matrix. Then we have that for any $\eta > 0$, there exists $C(\eta)$ (independent of $d$), such that*

$$\mathbb{E}[\widehat{\boldsymbol{f}}_T^\mathsf{T} \boldsymbol{\Psi}_T^\mathsf{T} \boldsymbol{\Psi}_S \boldsymbol{D} \boldsymbol{\Psi}_S^\mathsf{T} \boldsymbol{\Psi}_T \widehat{\boldsymbol{f}}_T]/n^2 \leq C(\eta) \|\mathsf{P}_T f_d\|_{L^{2+\eta}}^2 \text{Tr}(\boldsymbol{D})/n,$$

*where the expectation is with respect to the randomness in $\boldsymbol{\Psi}_S, \boldsymbol{\Psi}_T$.*

*Proof.* Let $\iota : S \to [|S|]$ be the bijection such that $\boldsymbol{\Psi}_S = (\psi_{\iota^{-1}(k)}(\boldsymbol{x}_i))_{i \in [n], k \in [|S|]}$ then we have

$$\mathbb{E}[\widehat{\boldsymbol{f}}_T^\mathsf{T} \boldsymbol{\Psi}_T^\mathsf{T} \boldsymbol{\Psi}_S \boldsymbol{D} \boldsymbol{\Psi}_S^\mathsf{T} \boldsymbol{\Psi}_T \widehat{\boldsymbol{f}}_T]/n^2 = \sum_{u,v \in T} \sum_{s \in S} \sum_{i,j \in [n]} \boldsymbol{D}_{\iota(s)\iota(s)} \mathbb{E}[\psi_u(\boldsymbol{x}_i)\psi_s(\boldsymbol{x}_i)\psi_s(\boldsymbol{x}_j)\psi_v(\boldsymbol{x}_j)] \hat{f}_u \hat{f}_v/n^2$$

$$= \sum_{u,v \in T} \sum_{s \in S} \sum_{i \in [n]} \boldsymbol{D}_{\iota(s)\iota(s)} \mathbb{E}[\psi_u(\boldsymbol{x}_i)\psi_s(\boldsymbol{x}_i)\psi_s(\boldsymbol{x}_i)\psi_v(\boldsymbol{x}_i)] \hat{f}_u \hat{f}_v/n^2$$

$$= \frac{1}{n} \sum_{s \in S} \boldsymbol{D}_{\iota(s)\iota(s)} \mathbb{E}_{\boldsymbol{x}}[(\mathsf{P}_T f_d(\boldsymbol{x}))^2 \psi_s(\boldsymbol{x})^2]$$

$$\leq \frac{1}{n} \sum_{s \in S} \boldsymbol{D}_{\iota(s)\iota(s)} \|\mathsf{P}_T f_d\|_{L^{2+\eta}}^2 \|\psi_s\|_{L^{(4+2\eta)/\eta}}^2$$

$$\leq C(\eta) \|\mathsf{P}_T f_d\|_{L^{2+\eta}}^2 \sum_{i=1}^n \boldsymbol{D}_{ii}/n,$$

where the second to last inequality is by Holder's inequality and the last inequality used the hyper-contractivity assumption as in Assumption 1(a). $\qquad\square$

**Lemma 8** (Exponential Kernel Decomposition). *Assume the conditions of Lemma 6 hold and assume there exists $\delta_0 > 0$, such that $(t(d), \mathsf{u}(d))$ satisfies the condition*

$$t(d)^{1+\delta_0} \leq \frac{1}{\lambda_{d,\mathsf{u}(d)+1}^2}. \tag{21}$$

*Let $j(d)$ satisfy $\min\{\mathsf{u}(d), \mathsf{m}(d)\} \leq j(d) \leq \mathsf{m}(d)$ then*

$$\left\| e^{-t\boldsymbol{H}_d/n} - e^{-(t/n)(\boldsymbol{\Psi}_{\leq j}\boldsymbol{D}_{\leq j}^2\boldsymbol{\Psi}_{\leq j}^\mathsf{T} + \kappa_H \mathbf{I}_n)} \right\|_{\text{op}} = o_{d,\mathbb{P}}(1).$$

*Proof.* First consider the regime $t = O_d(n/\kappa_H)$. Recall the decomposition,

$$\boldsymbol{H} = \boldsymbol{\Psi}_{\leq \mathsf{m}} \boldsymbol{D}_{\leq \mathsf{m}}^2 \boldsymbol{\Psi}_{\leq \mathsf{m}}^\mathsf{T} + \kappa_H (\mathbf{I}_n + \boldsymbol{\Delta}_H)$$

where $\|\boldsymbol{\Delta}_H\|_{\text{op}} = o_{d,\mathbb{P}}(1)$. By Lemma 1 and Lemma 4,

$$\left\| e^{-t\boldsymbol{H}/n} - e^{-(t/n)(\boldsymbol{\Psi}_{\leq j}\boldsymbol{D}_{\leq j}^2\boldsymbol{\Psi}_{\leq j}^\mathsf{T} + \kappa_H \mathbf{I})} \right\|_{\text{op}} \leq \left\| \sum_{k \geq j+1}^{\mathsf{m}} t\lambda_{d,k}^2 \psi_k \psi_k^\mathsf{T}/n \right\|_{\text{op}} + (t/n)\kappa_H \|\boldsymbol{\Delta}_h\|_{\text{op}}$$

$$\leq (t \cdot \max_{k \geq j+1} \cdot \lambda_{d,k}^2) \left\| \boldsymbol{\Psi}_{\leq \mathsf{m}}^\mathsf{T} \boldsymbol{\Psi}_{\leq \mathsf{m}}/n \right\|_{\text{op}} + o_{d,\mathbb{P}}(1)$$

$$\leq O_{d,\mathbb{P}}(t \cdot \max_{k \geq j+1} \lambda_{d,k}^2) + o_{d,\mathbb{P}}(1)$$

$$\overset{(a)}{=} O_{d,\mathbb{P}}(t^{-\delta_0}) + o_{d,\mathbb{P}}(1) = o_{d,\mathbb{P}}(1),$$

where equality $(a)$ holds by assumption Eq. (21). Now if $t = \omega_d(n/\kappa_H)$, then it is easy to see that

$$\max\left\{\left\|e^{-t\boldsymbol{H}/n}\right\|_{\text{op}}, \left\|e^{-(t/n)(\boldsymbol{\Psi}_{\leq j}\boldsymbol{D}_{\leq j}^2\boldsymbol{\Psi}_{\leq j}^\mathsf{T}+\kappa_H\mathbf{I})}\right\|_{\text{op}}\right\} \leq e^{-\omega_{d,\mathbb{P}}(1)} = o_{d,\mathbb{P}}(1),$$

and therefore

$$\left\|e^{-t\boldsymbol{H}/n} - e^{-(t/n)(\boldsymbol{\Psi}_{\leq j}\boldsymbol{D}_{\leq j}^2\boldsymbol{\Psi}_{\leq j}^\mathsf{T}+\kappa_H\mathbf{I})}\right\|_{\text{op}}$$
$$\leq 2\max\left\{\left\|e^{-t\boldsymbol{H}/n}\right\|_{\text{op}}, \left\|e^{-(t/n)(\boldsymbol{\Psi}_{\leq j}\boldsymbol{D}_{\leq j}^2\boldsymbol{\Psi}_{\leq j}^\mathsf{T}+\kappa_H\mathbf{I})}\right\|_{\text{op}}\right\} = o_{d,\mathbb{P}}(1).$$

$\square$

### B.3 EMPIRICAL WORLD - TRAIN

If $t = \omega_d(n/\kappa_H)$, then by Eq. (17) it is easy to see that $\left\|e^{-2(t/n)\boldsymbol{H}}\right\|_{\text{op}} = o_{d,\mathbb{P}}(1)$, hence

$$\widehat{R}_n(\hat{f}_t) = o_{d,\mathbb{P}}(1) \cdot \|f_d\|_{L^2}^2.$$

From now on we focus on the case that $t = o_d(n/\kappa_H)$. Let us decompose the training error as follows

$$\widehat{R}_n(\hat{f}_t) = R_1 + R_2 + R_3$$

where

$$R_1 = \frac{1}{n}\boldsymbol{f}^\mathsf{T}e^{-2(t/n)\boldsymbol{H}}\boldsymbol{f},$$
$$R_2 = \frac{2}{n}\boldsymbol{\varepsilon}^\mathsf{T}e^{-2(t/n)\boldsymbol{H}}\boldsymbol{f},$$
$$R_3 = \frac{1}{n}\boldsymbol{\varepsilon}^\mathsf{T}e^{-2(t/n)\boldsymbol{H}}\boldsymbol{\varepsilon}.$$

#### B.3.1 TERM $R_1$

Let us start by analysing $R_1$. We can write

$$R_1 = \frac{1}{n}(\boldsymbol{f}_{\leq\ell} + \boldsymbol{f}_{>\ell})^\mathsf{T}e^{-2(t/n)\boldsymbol{H}}(\boldsymbol{f}_{\leq\ell} + \boldsymbol{f}_{>\ell})$$
$$= T_1 + 2T_2 + T_3$$

where

$$T_1 = \frac{1}{n}\boldsymbol{f}_{\leq\ell}^\mathsf{T}e^{-2(t/n)\boldsymbol{H}}\boldsymbol{f}_{\leq\ell},$$
$$T_2 = \frac{1}{n}\boldsymbol{f}_{>\ell}^\mathsf{T}e^{-2(t/n)\boldsymbol{H}}\boldsymbol{f}_{\leq\ell},$$
$$T_3 = \frac{1}{n}\boldsymbol{f}_{>\ell}^\mathsf{T}e^{-2(t/n)\boldsymbol{H}}\boldsymbol{f}_{>\ell}.$$

Let us analyse the term $T_1$,

$$T_1 \leq \frac{1}{n}\boldsymbol{f}_{\leq\ell}^\mathsf{T}e^{-2(t/n)\boldsymbol{\Psi}_{\leq\ell}\boldsymbol{D}_{\leq\ell}^2\boldsymbol{\Psi}_{\leq\ell}^\mathsf{T}}\boldsymbol{f}_{\leq\ell}.$$

By Lemma 2, denoting $\boldsymbol{B} = \boldsymbol{\Psi}_{\leq\ell}^\mathsf{T}\boldsymbol{\Psi}_{\leq\ell}/n$,

$$\frac{1}{n}\boldsymbol{\Psi}_{\leq\ell}^\mathsf{T}e^{-2(t/n)\boldsymbol{\Psi}_{\leq\ell}\boldsymbol{D}_{\leq\ell}^2\boldsymbol{\Psi}_{\leq\ell}^\mathsf{T}}\boldsymbol{\Psi}_{\leq\ell} = \boldsymbol{B}e^{-2t\boldsymbol{D}_{\leq\ell}^2}\boldsymbol{B}.$$

We will show that

$$\left\|\boldsymbol{B}e^{-2t\boldsymbol{D}_{\leq\ell}^2}\boldsymbol{B}\right\|_{\text{op}} = o_{d,\mathbb{P}}(1). \tag{22}$$

By Lemma 4, since $\|\boldsymbol{B}\|_{\mathrm{op}} = \Theta_{d,\mathbb{P}}(1)$, for $d$ large $\boldsymbol{B}$ has a positive-definite square root $\boldsymbol{B}^{1/2}$ w.h.p. Therefore we can write

$$\boldsymbol{B}e^{-2t\boldsymbol{D}_{\le\ell}^2\boldsymbol{B}} = \boldsymbol{B}^{1/2}e^{-2t\boldsymbol{B}^{1/2}\boldsymbol{D}_{\le\ell}^2\boldsymbol{B}^{1/2}}\boldsymbol{B}^{1/2}$$

and bound the operator norm

$$\left\|\boldsymbol{B}e^{-2t\boldsymbol{D}_{\le\ell}^2\boldsymbol{B}}\right\|_{\mathrm{op}} \le \|\boldsymbol{B}\|_{\mathrm{op}}\left\|e^{-2t\boldsymbol{B}^{1/2}\boldsymbol{D}_{\le\ell}^2\boldsymbol{B}^{1/2}}\right\|_{\mathrm{op}}$$

$$= \|\boldsymbol{B}\|_{\mathrm{op}}\, e^{-2t\lambda_{\min}(\boldsymbol{B}^{1/2}\boldsymbol{D}_{\le\ell}^2\boldsymbol{B}^{1/2})}$$

$$\le \|\boldsymbol{B}\|_{\mathrm{op}}\, e^{-2t\lambda_{\max}(\boldsymbol{B})\lambda_{\min}(\boldsymbol{D}_{\le\ell}^2)} = o_{d,\mathbb{P}}(1).$$

since by Assumption 3(a), $t\lambda_{\min}(\boldsymbol{D}_{\le\ell}^2) = \Omega(t^{\delta_0})$. Therefore $T_1 = o_{d,\mathbb{P}}(1) \cdot \|\mathsf{P}_{\le\ell}f_d\|_{L^2}^2$.

Now we analyse the term $T_3$. Observe that by the inequality $1 - x \le e^{-x} \le 1$,

$$\frac{1}{n}\|\boldsymbol{f}_{>\ell}\|_2^2 - (2t/n)\frac{1}{n}\boldsymbol{f}_{>\ell}^\mathsf{T}(\boldsymbol{\Psi}_{\le\ell}\boldsymbol{D}_{\le\ell}^2\boldsymbol{\Psi}_{\le\ell}^\mathsf{T} + \kappa_H(\mathbf{I}_n + \boldsymbol{\Delta}_H))\boldsymbol{f}_{>\ell} \le T_3 \le \frac{1}{n}\|\boldsymbol{f}_{>\ell}\|_2^2.$$

We have by Lemma 7,

$$t\mathbb{E}[\widehat{\boldsymbol{f}}_{>\ell}^\mathsf{T}\boldsymbol{\Psi}_{>\ell}^\mathsf{T}\boldsymbol{\Psi}_{\le\ell}\boldsymbol{D}_{\le\ell}^2\boldsymbol{\Psi}_{\le\ell}^\mathsf{T}\boldsymbol{\Psi}_{>\ell}\widehat{\boldsymbol{f}}_{>\ell}]/n^2 \le C(\eta)\,\|\mathsf{P}_{>\ell}f_d\|_{L^{2+\eta}}^2\left(\frac{t}{n}\sum_{s=1}^\ell \lambda_{d,s}^2\right)$$

where the last quantity is $o_{d,\mathbb{P}}(1) \cdot \|\mathsf{P}_{>\ell}f_d\|_{L^{2+\eta}}^2$ by Assumption 3(c). Thus we see that

$$T_3 = \|\mathsf{P}_{>\ell}f_d\|_{L^2}^2 + o_{d,\mathbb{P}}(1) \cdot \|\mathsf{P}_{>\ell}f_d\|_{L^{2+\eta}}^2.$$

Observe that by the Cauchy-Schwarz inequality,

$$T_2 \le (T_1T_3)^{1/2} \le o_{d,\mathbb{P}}(1)\,\|\mathsf{P}_{\le\ell}f_d\|_{L^2}\,\|\mathsf{P}_{>\ell}f_d\|_{L^{2+\eta}}.$$

Putting everything together we see that,

$$R_1 = \|\mathsf{P}_{>\ell}f_d\|_{L^2}^2 + o_{d,\mathbb{P}}(1) \cdot (\|f_d\|_{L^2}^2 + \|\mathsf{P}_{>\ell}f_d\|_{L^{2+\eta}}^2).$$

### B.3.2    TERM $R_2$

Turning to $R_2$, we take the second-moment with respect to $\boldsymbol{\varepsilon}$

$$\frac{1}{\sigma_\varepsilon^2}\mathbb{E}_{\boldsymbol{\varepsilon}}[R_2^2] = \frac{4}{n^2}\mathbb{E}_{\boldsymbol{\varepsilon}}[\boldsymbol{\varepsilon}^\mathsf{T}e^{-2(t/n)\boldsymbol{H}}\boldsymbol{f}\boldsymbol{f}^\mathsf{T}e^{-2(t/n)\boldsymbol{H}}\boldsymbol{\varepsilon}]/\sigma_\varepsilon^2$$

$$= \frac{4}{n^2}\boldsymbol{f}^\mathsf{T}e^{-4(t/n)\boldsymbol{H}}\boldsymbol{f} \le \frac{4}{n}(\|\boldsymbol{f}\|_2^2/n)$$

$$= o_{d,\mathbb{P}}(1) \cdot \|f_d\|_{L^2}^2.$$

By Markov's inequality $R_2 = o_{d,\mathbb{P}}(1) \cdot \|f_d\|_{L^2}^2\,\sigma_\varepsilon^2$.

### B.3.3    TERM $R_3$

Now let us analyse $R_3$. Recalling the definition $\boldsymbol{B} = \boldsymbol{\Psi}_{\le\ell}^\mathsf{T}\boldsymbol{\Psi}_{\le\ell}/n$, we can compute the expectation

$$\frac{1}{\sigma_\varepsilon^2}(1 - \mathbb{E}_{\boldsymbol{\varepsilon}}[R_3]) = \frac{1}{n}\,\mathrm{Tr}\Big(\mathbf{I}_n - e^{-2(t/n)\boldsymbol{H}}\Big)$$

$$\stackrel{(a)}{=} \frac{1}{n}e^{-\kappa_H t/n}\,\mathrm{Tr}\Big(\mathbf{I}_n - e^{-2(t/n)(\boldsymbol{\Psi}_{\le\ell}\boldsymbol{D}_{\le\ell}^2\boldsymbol{\Psi}_{\le\ell}^\mathsf{T})}\Big) + o_{d,\mathbb{P}}(1)$$

$$\stackrel{(b)}{\le} \frac{1}{n}e^{-\kappa_H t/n}\,\mathrm{Tr}\big(2(t/n)(\boldsymbol{\Psi}_{\le\ell}\boldsymbol{D}_{\le\ell}^2\boldsymbol{\Psi}_{\le\ell}^\mathsf{T})\big) + o_{d,\mathbb{P}}(1)$$

$$\stackrel{(c)}{=} \frac{1}{n}e^{-\kappa_H t/n}\,\mathrm{Tr}\big(2t\boldsymbol{B}\boldsymbol{D}_{\le\ell}^2\big) + o_{d,\mathbb{P}}(1)$$

$$\le \frac{2t}{n}e^{-\kappa_H t/n}\,\|\boldsymbol{B}\|_{\mathrm{op}}\,\mathrm{Tr}\big(\boldsymbol{D}_{\le\ell}^2\big) + o_{d,\mathbb{P}}(1) = o_{d,\mathbb{P}}(1).$$

Equality $(a)$ follows from Lemma 8. For $(b)$ we used the inequality $1 - e^{-x} \leq x$ and that trace is the sum of eigenvalues. Equality $(c)$ uses the cyclic property of trace. In the last equality we used that by Lemma 4, $\|\boldsymbol{B}\|_{\mathrm{op}} = O_{d,\mathbb{P}}(1)$ and Assumption 3(c) implies that $(t/n) \operatorname{Tr}(\boldsymbol{D}_{\leq \ell}^2) = O_d((t/n)\kappa_H) = o_d(1)$ since by assumption we consider $t = o_d(n/\kappa_H)$. Turning to the variance

$$
\begin{aligned}
\operatorname{Var}_{\boldsymbol{\varepsilon}}[R_3] &= \mathbb{E}_{\boldsymbol{\varepsilon}}[R_3^2] - \mathbb{E}_{\boldsymbol{\varepsilon}}[R_3]^2 \\
&= \frac{1}{n^2} \mathbb{E}_{\boldsymbol{\varepsilon}}[(\boldsymbol{\varepsilon}^{\mathsf{T}} e^{-2(t/n)\boldsymbol{H}} \boldsymbol{\varepsilon})^2] - \mathbb{E}_{\boldsymbol{\varepsilon}}[R_3]^2 \\
&\leq \frac{1}{n^2} \mathbb{E}_{\boldsymbol{\varepsilon}}[(\boldsymbol{\varepsilon}^{\mathsf{T}} \boldsymbol{\varepsilon})^2] - \sigma_{\varepsilon}^4 (1 + o_{d,\mathbb{P}}(1)) \\
&\overset{(a)}{=} O(1/n) \cdot \sigma_{\varepsilon}^4 + \sigma_{\varepsilon}^4 - \sigma_{\varepsilon}^4 (1 + o_{d,\mathbb{P}}(1)) \\
&= o_{d,\mathbb{P}}(1) \cdot \sigma_{\varepsilon}^4,
\end{aligned}
$$

where the first inequality uses that $\left\|e^{-2(t/n)\boldsymbol{H}}\right\|_{\mathrm{op}} \leq 1$ and $(a)$ uses that for $\varepsilon_i \sim \mathcal{N}(0, \sigma_{\varepsilon}^2)$, $\mathbb{E}[\varepsilon_i^4] = 3\sigma_{\varepsilon}^4$. Therefore by Chebyshev's inequality, $R_3 = \sigma_{\varepsilon}^2 (1 + o_{d,\mathbb{P}}(1))$.

Putting everything together yields

$$
\widehat{R}_n(\hat{f}_t) = R_1 + R_2 + R_3 = \|\mathsf{P}_{>\ell}\|_{L^2}^2 + o_{d,\mathbb{P}}(1) \cdot (\|f_d\|_{L^{2+\eta}}^2 + \sigma_{\varepsilon}^2).
$$

## B.4    EMPIRICAL WORLD - TEST

Recalling $\boldsymbol{u}(t)$ from Eq. (14), let

$$
\boldsymbol{u}(t) = \boldsymbol{v}(t) + \boldsymbol{\varepsilon}(t)
$$

where

$$
\begin{aligned}
\boldsymbol{v}(t) &= (\mathbf{I}_n - e^{-t\boldsymbol{H}/n})\boldsymbol{f}, & (23) \\
\boldsymbol{\varepsilon}(t) &= (\mathbf{I}_n - e^{-t\boldsymbol{H}/n})\boldsymbol{\varepsilon}. & (24)
\end{aligned}
$$

Recall the expansion of the test error from Eq. (15),

$$
\begin{aligned}
R(\hat{f}_t) &= \mathbb{E}_{\boldsymbol{x}}[f_d(\boldsymbol{x})^2] - 2\boldsymbol{u}(t)^{\mathsf{T}} \boldsymbol{H}^{-1} \boldsymbol{E} + \boldsymbol{u}(t)^{\mathsf{T}} \boldsymbol{H}^{-1} \boldsymbol{M} \boldsymbol{H}^{-1} \boldsymbol{u}(t) \\
&= \|f_d\|_{L^2}^2 - 2T_1 + T_2 + T_3 - 2T_4 + 2T_5,
\end{aligned}
$$

where

$$
\begin{aligned}
T_1 &= \boldsymbol{v}(t)^{\mathsf{T}} \boldsymbol{H}^{-1} \boldsymbol{E}, \\
T_2 &= \boldsymbol{v}(t)^{\mathsf{T}} \boldsymbol{H}^{-1} \boldsymbol{M} \boldsymbol{H}^{-1} \boldsymbol{v}(t), \\
T_3 &= \boldsymbol{\varepsilon}(t)^{\mathsf{T}} \boldsymbol{H}^{-1} \boldsymbol{M} \boldsymbol{H}^{-1} \boldsymbol{\varepsilon}(t), \\
T_4 &= \boldsymbol{\varepsilon}(t)^{\mathsf{T}} \boldsymbol{H}^{-1} \boldsymbol{E}, \\
T_5 &= \boldsymbol{\varepsilon}(t)^{\mathsf{T}} \boldsymbol{H}^{-1} \boldsymbol{M} \boldsymbol{H}^{-1} \boldsymbol{v}(t).
\end{aligned}
$$

The proof for the test error is the most involved, but will follow a similar strategy of analysing each term in the expansion. We first begin by analysing $T_2$ in Section B.4.1, then $T_1$ in Section B.4.2, and finally terms $T_3, T_4, T_5$, which are all simpler than the first two, in Section B.4.3. At the end of this Appendix section we present the proof of Theorem 5.

### B.4.1    TERM $T_2$

As before, we will analyze each term separately. We begin with term $T_2$.

**Proposition 1** (Term $T_2$).

$$
T_2 = \boldsymbol{v}(t)^{\mathsf{T}} \boldsymbol{H}^{-1} \boldsymbol{M} \boldsymbol{H}^{-1} \boldsymbol{v}(t) = \left\| \boldsymbol{S}_{\leq \ell} \widehat{\boldsymbol{f}}_{\leq \ell} \right\|_2^2 + o_{d,\mathbb{P}}(1) \cdot \|f_d\|_{L^{2+\eta}}^2
$$

where we recall the shrinkage matrix $\boldsymbol{S}_{\leq \mathsf{m}}$ defined in Eq. (20) and $\boldsymbol{v}(t)$ in Eq. (23).

*Proof of Proposition 1.* To analyze $T_2$ we further decompose it into the following terms

$$T_2 = (\boldsymbol{f}_{\leq \ell} + \boldsymbol{f}_{> \ell})^{\mathsf{T}}(\mathbf{I}_n - e^{-t\boldsymbol{H}/n})\boldsymbol{H}^{-1}\boldsymbol{M}\boldsymbol{H}^{-1}(\mathbf{I}_n - e^{-t\boldsymbol{H}/n})(\boldsymbol{f}_{\leq \ell} + \boldsymbol{f}_{> \ell})$$
$$= T_{21} + T_{22} + T_{23},$$

where

$$T_{21} = \boldsymbol{f}_{\leq \ell}^{\mathsf{T}}(\mathbf{I}_n - e^{-t\boldsymbol{H}/n})\boldsymbol{H}^{-1}\boldsymbol{M}\boldsymbol{H}^{-1}(\mathbf{I}_n - e^{-t\boldsymbol{H}/n})\boldsymbol{f}_{\leq \ell}, \tag{25}$$

$$T_{22} = 2\boldsymbol{f}_{\leq \ell}^{\mathsf{T}}(\mathbf{I}_n - e^{-t\boldsymbol{H}/n})\boldsymbol{H}^{-1}\boldsymbol{M}\boldsymbol{H}^{-1}(\mathbf{I}_n - e^{-t\boldsymbol{H}/n})\boldsymbol{f}_{> \ell}, \tag{26}$$

$$T_{23} = \boldsymbol{f}_{> \ell}^{\mathsf{T}}(\mathbf{I}_n - e^{-t\boldsymbol{H}/n})\boldsymbol{H}^{-1}\boldsymbol{M}\boldsymbol{H}^{-1}(\mathbf{I}_n - e^{-t\boldsymbol{H}/n})\boldsymbol{f}_{> \ell}. \tag{27}$$

Using Lemma 9 and 10 proven below, by the Cauchy-Schwarz inequality,

$$T_{22} \leq (2T_{21}T_{23})^{1/2} = o_{d,\mathbb{P}}(1) \cdot \|\mathsf{P}_{\leq \ell}f_d\|_{L^2} \|f_d\|_{L^{2+\eta}}$$

and hence

$$T_2 = T_{21} + T_{22} + T_{23} = \left\|\boldsymbol{S}_{\leq \ell}\widehat{\boldsymbol{f}}_{\leq \ell}\right\|_2^2 + o_{d,\mathbb{P}}(1) \cdot \|f_d\|_{L^{2+\eta}}^2.$$

$\square$

**Lemma 9** (Term $T_{21}$ Eq. (25))**.**

$$T_{21} = \left\|\boldsymbol{S}_{\leq \ell}\widehat{\boldsymbol{f}}_{\leq \ell}\right\|_2^2 + o_{d,\mathbb{P}}(1) \cdot \|\mathsf{P}_{\leq \ell}f_d\|_{L^2}^2$$

*Proof of Lemma 9.* Recall the notation $\alpha$ and $\boldsymbol{K}_{\leq \ell}$ from Eq. (19). Define $\boldsymbol{\Delta}$ such that

$$\boldsymbol{\Delta} := (\mathbf{I}_n - e^{-t\boldsymbol{H}/n}) - \boldsymbol{K}_{\leq \ell}$$

which by Lemma 8 satisfies $\|\boldsymbol{\Delta}\|_{\mathrm{op}} = o_{d,\mathbb{P}}(1)$. Then we can split $T_{21}$ into

$$T_{21} = T_{211} + T_{212} + T_{213} + T_{214}$$

where

$T_{211} = \boldsymbol{f}_{\leq \ell}^{\mathsf{T}}\boldsymbol{K}_{\leq \ell}\boldsymbol{\Psi}_{\leq \ell}\boldsymbol{S}_{\leq \ell}^2\boldsymbol{\Psi}_{\leq \ell}^{\mathsf{T}}\boldsymbol{K}_{\leq \ell}\boldsymbol{f}_{\leq \ell}/n^2,$

$T_{212} = \boldsymbol{f}_{\leq \ell}^{\mathsf{T}}\boldsymbol{K}_{\leq \ell}\boldsymbol{\Psi}_{\ell\mathsf{m}}\boldsymbol{S}_{\ell\mathsf{m}}^2\boldsymbol{\Psi}_{\ell\mathsf{m}}^{\mathsf{T}}\boldsymbol{K}_{\leq \ell}\boldsymbol{f}_{\leq \ell}/n^2,$

$T_{213} = \boldsymbol{f}_{\leq \ell}^{\mathsf{T}}\boldsymbol{K}_{\leq \ell}(\boldsymbol{H}^{-1}\boldsymbol{M}\boldsymbol{H}^{-1} - \boldsymbol{\Psi}_{\leq \mathsf{m}}\boldsymbol{S}_{\leq \mathsf{m}}^2\boldsymbol{\Psi}_{\leq \mathsf{m}}^{\mathsf{T}}/n^2)\boldsymbol{K}_{\leq \ell}\boldsymbol{f}_{\leq \ell},$

$T_{214} = \boldsymbol{f}_{\leq \ell}^{\mathsf{T}}\boldsymbol{\Delta}\boldsymbol{H}^{-1}\boldsymbol{M}\boldsymbol{H}^{-1}\boldsymbol{K}_{\leq \ell}\boldsymbol{f}_{\leq \ell} + \boldsymbol{f}_{\leq \ell}^{\mathsf{T}}\boldsymbol{K}_{\leq \ell}\boldsymbol{H}^{-1}\boldsymbol{M}\boldsymbol{H}^{-1}\boldsymbol{\Delta}\boldsymbol{f}_{\leq \ell} + \boldsymbol{f}_{\leq \ell}^{\mathsf{T}}\boldsymbol{\Delta}\boldsymbol{H}^{-1}\boldsymbol{M}\boldsymbol{H}^{-1}\boldsymbol{\Delta}\boldsymbol{f}_{\leq \ell}.$

We will show that the dominant term is $T_{211}$ and the others are of lower order. By Lemma 3,

$$\left\|n\boldsymbol{H}^{-1}\boldsymbol{M}\boldsymbol{H}^{-1} - \boldsymbol{\Psi}_{\leq \mathsf{m}}\boldsymbol{S}_{\leq \mathsf{m}}^2\boldsymbol{\Psi}_{\leq \mathsf{m}}^{\mathsf{T}}/n\right\|_{\mathrm{op}} = o_{d,\mathbb{P}}(1),$$

and since $\boldsymbol{K}_{\leq \ell} \preceq \mathbf{I}_n$, $T_{213} = o_{d,\mathbb{P}}(1) \cdot \|\mathsf{P}_{\leq \ell}f_d\|_{L^2}^2$. By Lemma 3, Lemma 4 and since $\boldsymbol{S}_{\leq \mathsf{m}} \preceq \mathbf{I}_\mathsf{m}$,

$$\left\|n\boldsymbol{H}^{-1}\boldsymbol{M}\boldsymbol{H}^{-1}\right\|_{\mathrm{op}} \leq \left\|\boldsymbol{\Psi}_{\leq \mathsf{m}}\boldsymbol{S}_{\leq \mathsf{m}}^2\boldsymbol{\Psi}_{\leq \mathsf{m}}^{\mathsf{T}}/n\right\|_{\mathrm{op}} + o_{d,\mathbb{P}}(1) \leq 1 + o_{d,\mathbb{P}}(1)$$

hence it is easy to see that $T_{214} = o_{d,\mathbb{P}}(1) \cdot \|\mathsf{P}_{\leq \ell}f_d\|_{L^2}^2$. Turning to $T_{212}$ we have

$$T_{212} \leq \frac{1}{n^2}\boldsymbol{f}_{\leq \ell}^{\mathsf{T}}(\mathbf{I}_n - \alpha e^{-(t/n)\boldsymbol{\Psi}_{\leq \ell}\boldsymbol{D}_{\leq \ell}^2\boldsymbol{\Psi}_{\leq \ell}^{\mathsf{T}}})\boldsymbol{\Psi}_{\ell\mathsf{m}}\boldsymbol{\Psi}_{\ell\mathsf{m}}^{\mathsf{T}}(\mathbf{I}_n - \alpha e^{-(t/n)\boldsymbol{\Psi}_{\leq \ell}\boldsymbol{D}_{\leq \ell}^2\boldsymbol{\Psi}_{\leq \ell}^{\mathsf{T}}})\boldsymbol{f}_{\leq \ell}$$

$$\overset{(a)}{=} \frac{1}{n^2}\widehat{\boldsymbol{f}}_{\leq \ell}^{\mathsf{T}}(\mathbf{I} - \alpha e^{-(t/n)\boldsymbol{B}\boldsymbol{D}_{\leq \ell}^2})\boldsymbol{\Psi}_{\leq \ell}^{\mathsf{T}}\boldsymbol{\Psi}_{\ell\mathsf{m}}\boldsymbol{\Psi}_{\ell\mathsf{m}}^{\mathsf{T}}\boldsymbol{\Psi}_{\leq \ell}(\mathbf{I} - \alpha e^{-(t/n)\boldsymbol{B}\boldsymbol{D}_{\leq \ell}^2})\widehat{\boldsymbol{f}}_{\leq \ell}^{\mathsf{T}}$$

$$\leq \left\|\widehat{\boldsymbol{f}}_{\leq \ell}\right\|_2^2 \left\|\boldsymbol{\Psi}_{\leq \ell}^{\mathsf{T}}\boldsymbol{\Psi}_{\ell\mathsf{m}}/n\right\|_{\mathrm{op}}^2 = o_{d,\mathbb{P}}(1) \cdot \|\mathsf{P}_{\leq \ell}f_d\|_{L^2}^2,$$

where the first inequality used $\boldsymbol{S}_{\ell\mathsf{m}}^2 \preceq \mathbf{I}_{\mathsf{m}-\ell}$, we used Lemma 2 in $(a)$, and the last equality used Lemma 4 (note that $\boldsymbol{\Psi}_{\leq \ell}^{\mathsf{T}}\boldsymbol{\Psi}_{\ell\mathsf{m}}/n$ is an off-diagonal block of $\boldsymbol{\Psi}_{\leq \mathsf{m}}^{\mathsf{T}}\boldsymbol{\Psi}_{\leq \mathsf{m}}/n$ which corresponds to an all zeroes submatrix of $\mathbf{I}_\mathsf{m}$). Finally we look at the main term $T_{211}$. Defining the matrices

$$\boldsymbol{A} := \frac{1}{n}\boldsymbol{\Psi}_{\leq \ell}^{\mathsf{T}}e^{-(t/n)\boldsymbol{\Psi}_{\leq \ell}\boldsymbol{D}_{\leq \ell}^2\boldsymbol{\Psi}_{\leq \ell}^{\mathsf{T}}}\boldsymbol{\Psi}_{\leq \ell},$$

$$\boldsymbol{B} := \frac{1}{n}\boldsymbol{\Psi}_{\leq \ell}^{\mathsf{T}}\boldsymbol{\Psi}_{\leq \ell},$$

we can write

$$T_{211} = \frac{1}{n^2} \widehat{\boldsymbol{f}}_{\leq \ell}^{\mathsf{T}} \boldsymbol{\Psi}_{\leq \ell}^{\mathsf{T}} (\mathbf{I}_n - \alpha e^{-(t/n)\boldsymbol{\Psi}_{\leq \ell} \boldsymbol{D}_{\leq \ell}^2 \boldsymbol{\Psi}_{\leq \ell}^{\mathsf{T}}}) \boldsymbol{\Psi}_{\leq \ell} \boldsymbol{S}_{\leq \ell}^2 \boldsymbol{\Psi}_{\leq \ell}^{\mathsf{T}} (\mathbf{I}_n - \alpha e^{-(t/n)\boldsymbol{\Psi}_{\leq \ell} \boldsymbol{D}_{\leq \ell}^2 \boldsymbol{\Psi}_{\leq \ell}^{\mathsf{T}}}) \boldsymbol{\Psi}_{\leq \ell} \widehat{\boldsymbol{f}}_{\leq \ell}$$
$$= \widehat{\boldsymbol{f}}_{\leq \ell}^{\mathsf{T}} (\boldsymbol{B} \boldsymbol{S}_{\leq \ell}^2 \boldsymbol{B} - 2\alpha \boldsymbol{A} \boldsymbol{S}_{\leq \ell}^2 \boldsymbol{B} + \alpha^2 \boldsymbol{A} \boldsymbol{S}_{\leq \ell}^2 \boldsymbol{A}) \widehat{\boldsymbol{f}}_{\leq \ell}.$$

Now observe that by Lemma 2,

$$\boldsymbol{A} = \frac{1}{n} \boldsymbol{\Psi}_{\leq \ell}^{\mathsf{T}} e^{-(t/n)\boldsymbol{\Psi}_{\leq \ell} \boldsymbol{D}_{\leq \ell}^2 \boldsymbol{\Psi}_{\leq \ell}^{\mathsf{T}}} \boldsymbol{\Psi}_{\leq \ell} = \boldsymbol{B} e^{-t \boldsymbol{D}_{\leq \ell}^2 \boldsymbol{B}}$$

and by the same argument used to show Eq. (22), $\|\boldsymbol{A}\|_{\mathrm{op}} = o_{d,\mathbb{P}}(1)$. Since $\boldsymbol{B} = \mathbf{I}_n + \boldsymbol{\Delta}'$ and $\boldsymbol{S}_{\leq \ell} \preceq \mathbf{I}_\ell$, we have

$$T_{211} = \left\| \boldsymbol{S}_{\leq \ell} \widehat{\boldsymbol{f}}_{\leq \ell} \right\|_2^2 + o_{d,\mathbb{P}}(1) \cdot \|\mathsf{P}_{\leq \ell} f_d\|_{L^2}^2 ,$$

hence combining terms

$$T_{21} = T_{211} + T_{212} + T_{213} + T_{214} = \left\| \boldsymbol{S}_{\leq \ell} \widehat{\boldsymbol{f}}_{\leq \ell} \right\|_2^2 + o_{d,\mathbb{P}}(1) \cdot \|\mathsf{P}_{\leq \ell} f_d\|_{L^2}^2 .$$

$\square$

**Lemma 10** (Term $T_{23}$ Eq. (27))**.**

$$T_{23} = \boldsymbol{f}_{>\ell}^{\mathsf{T}} (\mathbf{I}_n - e^{-t\boldsymbol{H}/n}) \boldsymbol{H}^{-1} \boldsymbol{M} \boldsymbol{H}^{-1} (\mathbf{I}_n - e^{-t\boldsymbol{H}/n}) \boldsymbol{f}_{>\ell} = o_{d,\mathbb{P}}(1) \cdot \|f_d\|_{L^{2+\eta}}^2 .$$

*Proof of Lemma 10.* Let us define the matrix

$$\boldsymbol{G} = (\mathbf{I}_n - e^{-t\boldsymbol{H}/n}) \boldsymbol{H}^{-1} \boldsymbol{M} \boldsymbol{H}^{-1} (\mathbf{I}_n - e^{-t\boldsymbol{H}/n}).$$

Using similar reasoning from Lemma 9 we can write

$$\boldsymbol{G} = \frac{1}{n^2} \boldsymbol{K}_{\leq \mathsf{m}} \boldsymbol{\Psi}_{\leq \mathsf{m}} \boldsymbol{S}_{\leq \mathsf{m}}^2 \boldsymbol{\Psi}_{\leq \mathsf{m}}^{\mathsf{T}} \boldsymbol{K}_{\leq \mathsf{m}} + \boldsymbol{\Delta}$$
$$= \frac{1}{n^2} \boldsymbol{K}_{\leq \ell} \boldsymbol{\Psi}_{\leq \mathsf{m}} \boldsymbol{S}_{\leq \mathsf{m}}^2 \boldsymbol{\Psi}_{\leq \mathsf{m}}^{\mathsf{T}} \boldsymbol{K}_{\leq \ell} + \boldsymbol{\Delta}'.$$

for matrices $\boldsymbol{\Delta}, \boldsymbol{\Delta}'$ satisfying $\max\{\|\boldsymbol{\Delta}\|_{\mathrm{op}}, \|\boldsymbol{\Delta}'\|_{\mathrm{op}}\} = o_{d,\mathbb{P}}(1)$. We can split $T_{23}$ into

$$T_{23} = \frac{1}{n} \boldsymbol{f}_{>\ell}^{\mathsf{T}} \boldsymbol{G} \boldsymbol{f}_{>\ell}$$
$$= \frac{1}{n} (\boldsymbol{f}_{>\mathsf{m}} + \boldsymbol{f}_{\ell\mathsf{m}})^{\mathsf{T}} \boldsymbol{G} (\boldsymbol{f}_{>\mathsf{m}} + \boldsymbol{f}_{\ell\mathsf{m}})$$
$$= \frac{1}{n} \boldsymbol{f}_{>\mathsf{m}}^{\mathsf{T}} \boldsymbol{G} \boldsymbol{f}_{>\mathsf{m}} + \frac{2}{n} \boldsymbol{f}_{\ell\mathsf{m}}^{\mathsf{T}} \boldsymbol{G} \boldsymbol{f}_{>\mathsf{m}} + \frac{1}{n} \boldsymbol{f}_{\ell\mathsf{m}}^{\mathsf{T}} \boldsymbol{G} \boldsymbol{f}_{\ell\mathsf{m}}$$
$$= T_{231} + T_{232} + T_{233} + T_{234}$$

where

$$T_{231} = \frac{1}{n^2} \boldsymbol{f}_{>\mathsf{m}}^{\mathsf{T}} \boldsymbol{K}_{\leq \mathsf{m}} \boldsymbol{\Psi}_{\leq \mathsf{m}} \boldsymbol{S}_{\leq \mathsf{m}}^2 \boldsymbol{\Psi}_{\leq \mathsf{m}}^{\mathsf{T}} \boldsymbol{K}_{\leq \mathsf{m}} \boldsymbol{f}_{>\mathsf{m}},$$
$$T_{232} = \frac{2}{n^2} \boldsymbol{f}_{\ell\mathsf{m}}^{\mathsf{T}} \boldsymbol{G} \boldsymbol{f}_{>\mathsf{m}},$$
$$T_{233} = \frac{1}{n^2} \boldsymbol{f}_{\ell\mathsf{m}}^{\mathsf{T}} \boldsymbol{K}_{\leq \ell} \boldsymbol{\Psi}_{\leq \mathsf{m}} \boldsymbol{S}_{\leq \mathsf{m}}^2 \boldsymbol{\Psi}_{\leq \mathsf{m}}^{\mathsf{T}} \boldsymbol{K}_{\leq \ell} \boldsymbol{f}_{\ell\mathsf{m}},$$
$$T_{234} = o_{d,\mathbb{P}}(1) \cdot \|\mathsf{P}_{>\ell} f_d\|_{L^2}^2 .$$

Let us first start with analysing $T_{231}$, defining $\boldsymbol{B} = \boldsymbol{\Psi}_{\leq \mathsf{m}}^{\mathsf{T}} \boldsymbol{\Psi}_{\leq \mathsf{m}}/n$ we have

$$T_{231} = \frac{1}{n^2} \boldsymbol{f}_{>\mathsf{m}}^{\mathsf{T}} \boldsymbol{\Psi}_{\leq \mathsf{m}} (\mathbf{I}_{\mathsf{m}} - \alpha e^{-(t/n)\boldsymbol{D}_{\leq \mathsf{m}}^2 \boldsymbol{B}}) \boldsymbol{S}_{\leq \mathsf{m}}^2 (\mathbf{I}_{\mathsf{m}} - \alpha e^{-(t/n)\boldsymbol{B} \boldsymbol{D}_{\leq \mathsf{m}}^2}) \boldsymbol{\Psi}_{\leq \mathsf{m}}^{\mathsf{T}} \boldsymbol{f}_{>\mathsf{m}}$$
$$\overset{(a)}{\leq} (1 + o_{d,\mathbb{P}}(1)) \boldsymbol{f}_{>\mathsf{m}}^{\mathsf{T}} \boldsymbol{\Psi}_{\leq \mathsf{m}} \boldsymbol{\Psi}_{\leq \mathsf{m}}^{\mathsf{T}} \boldsymbol{f}_{>\mathsf{m}}/n^2$$
$$= o_{d,\mathbb{P}}(1) \cdot \|\mathsf{P}_{>\mathsf{m}} f_d\|_{L^{2+\eta}}^2 ,$$

where the first equality is by Lemma 2 and the last line follows from Lemma 7, Markov's inequality, and by the fact that $\mathsf{m}/n = o_d(1)$ by Assumption 2(c). To see that inequality $(a)$ holds note that

$$\left\| \mathbf{I_m} - \alpha e^{-(t/n)\boldsymbol{D}_{\leq \mathsf{m}}^2 \boldsymbol{B}} \right\|_{\mathrm{op}} = \left\| \boldsymbol{B}^{-1/2}(\mathbf{I_m} - \alpha e^{-(t/n)\boldsymbol{B}^{1/2}\boldsymbol{D}_{\leq \mathsf{m}}^2 \boldsymbol{B}^{1/2}})\boldsymbol{B}^{1/2} \right\|_{\mathrm{op}}$$
$$\leq \left\| \boldsymbol{B}^{-1/2} \right\|_{\mathrm{op}} \left\| \boldsymbol{B}^{1/2} \right\|_{\mathrm{op}}.$$

Since $\boldsymbol{S}_{\leq \mathsf{m}}^2 \preceq \mathbf{I}_m$,

$$\left\| (\mathbf{I_m} - \alpha e^{-(t/n)\boldsymbol{D}_{\leq \mathsf{m}}^2 \boldsymbol{B}})\boldsymbol{S}_{\leq \mathsf{m}}^2 (\mathbf{I_m} - \alpha e^{-(t/n)\boldsymbol{B}\boldsymbol{D}_{\leq \mathsf{m}}^2}) \right\|_{\mathrm{op}} \leq \left\| \boldsymbol{B}^{-1} \right\|_{\mathrm{op}} \|\boldsymbol{B}\|_{\mathrm{op}} = 1 + o_{d,\mathbb{P}}(1),$$

where the last equality is by Lemma 4. Now let us turn to $T_{233}$, which we can further split into

$$T_{233} = T_{2331} + T_{2332} \tag{28}$$

where

$$T_{2331} = \frac{1}{n^2} \boldsymbol{f}_{\ell \mathsf{m}}^{\mathsf{T}} \boldsymbol{K}_{\leq \ell} \boldsymbol{\Psi}_{\leq \ell} \boldsymbol{S}_{\leq \ell}^2 \boldsymbol{\Psi}_{\leq \ell}^{\mathsf{T}} \boldsymbol{K}_{\leq \ell} \boldsymbol{f}_{\ell \mathsf{m}},$$
$$T_{2332} = \frac{1}{n^2} \boldsymbol{f}_{\ell \mathsf{m}}^{\mathsf{T}} \boldsymbol{K}_{\leq \ell} \boldsymbol{\Psi}_{\ell \mathsf{m}} \boldsymbol{S}_{\ell \mathsf{m}}^2 \boldsymbol{\Psi}_{\ell \mathsf{m}}^{\mathsf{T}} \boldsymbol{K}_{\leq \ell} \boldsymbol{f}_{\ell \mathsf{m}}.$$

Redefining $\boldsymbol{B} = \boldsymbol{\Psi}_{\leq \ell}^{\mathsf{T}} \boldsymbol{\Psi}_{\leq \ell}/n$ and using a similar argument as for $T_{231}$, the first term can be seen as

$$T_{2331} = \frac{1}{n^2} \widehat{\boldsymbol{f}}_{\ell \mathsf{m}}^{\mathsf{T}} \boldsymbol{\Psi}_{\ell \mathsf{m}}^{\mathsf{T}} \boldsymbol{\Psi}_{\leq \ell}(\mathbf{I} - \alpha e^{-(t/n)\boldsymbol{D}_{\leq \ell}^2 \boldsymbol{B}})\boldsymbol{S}_{\leq \ell}^2 (\mathbf{I} - \alpha e^{-(t/n)\boldsymbol{B}\boldsymbol{D}_{\leq \ell}^2})\boldsymbol{\Psi}_{\leq \ell}^{\mathsf{T}} \boldsymbol{\Psi}_{\ell \mathsf{m}} \widehat{\boldsymbol{f}}_{\ell \mathsf{m}}$$
$$\leq (1 + o_{d,\mathbb{P}}(1))\left\| \widehat{\boldsymbol{f}}_{\ell \mathsf{m}} \right\|_2^2 \left\| \boldsymbol{\Psi}_{\ell \mathsf{m}}^{\mathsf{T}} \boldsymbol{\Psi}_{\leq \ell}/n \right\|_{\mathrm{op}}^2 = o_{d,\mathbb{P}}(1) \cdot \|\mathsf{P}_{\ell \mathsf{m}} f_d\|_{L^2}^2 . \tag{29}$$

Turning to the second term $T_{2332}$, let

$$\overline{\boldsymbol{A}} := \frac{1}{n} \boldsymbol{\Psi}_{\ell \mathsf{m}}^{\mathsf{T}} \left( \mathbf{I}_n - e^{-(t/n)(\boldsymbol{\Psi}_{\leq \ell} \boldsymbol{D}_{\leq \ell}^2 \boldsymbol{\Psi}_{\leq \ell}^{\mathsf{T}} + \kappa_H \mathbf{I}_n)} \right) \boldsymbol{\Psi}_{\ell \mathsf{m}},$$

then we can write this term as

$$T_{2332} = \widehat{\boldsymbol{f}}_{\ell \mathsf{m}}^{\mathsf{T}} \overline{\boldsymbol{A}} \boldsymbol{S}_{\ell \mathsf{m}}^2 \overline{\boldsymbol{A}} \widehat{\boldsymbol{f}}_{\ell \mathsf{m}}$$
$$\leq \widehat{\boldsymbol{f}}_{\ell \mathsf{m}}^{\mathsf{T}} \overline{\boldsymbol{A}}^2 \widehat{\boldsymbol{f}}_{\ell \mathsf{m}} \leq \widehat{\boldsymbol{f}}_{\ell \mathsf{m}}^{\mathsf{T}} \overline{\boldsymbol{A}} \widehat{\boldsymbol{f}}_{\ell \mathsf{m}} + o_{d,\mathbb{P}}(1) \cdot \|\mathsf{P}_{\ell \mathsf{m}} f_d\|_{L^2}^2 , \tag{30}$$

where the last inequality holds since $\overline{\boldsymbol{A}} \preceq \boldsymbol{\Psi}_{\ell \mathsf{m}}^{\mathsf{T}} \boldsymbol{\Psi}_{\ell \mathsf{m}}/n = \mathbf{I}_{\mathsf{m}-\ell} + \boldsymbol{\Delta}$ by Lemma 4 and so

$$\overline{\boldsymbol{A}}^2 = \overline{\boldsymbol{A}}^{1/2} \overline{\boldsymbol{A}} \, \overline{\boldsymbol{A}}^{1/2}$$
$$\preceq \overline{\boldsymbol{A}}^{1/2}(\mathbf{I} + \boldsymbol{\Delta})\overline{\boldsymbol{A}}^{1/2}$$
$$= \overline{\boldsymbol{A}} + \overline{\boldsymbol{A}}^{1/2} \boldsymbol{\Delta} \overline{\boldsymbol{A}}^{1/2}$$
$$= \overline{\boldsymbol{A}} + \boldsymbol{\Delta}'.$$

By the inequality $1 - e^{-x} \leq x$, in the PSD order we see that

$$\overline{\boldsymbol{A}} \preceq \frac{t}{n^2} \boldsymbol{\Psi}_{\ell \mathsf{m}}^{\mathsf{T}} \boldsymbol{\Psi}_{\leq \ell} \boldsymbol{D}_{\leq \ell}^2 \boldsymbol{\Psi}_{\leq \ell}^{\mathsf{T}} \boldsymbol{\Psi}_{\ell \mathsf{m}} + \frac{\kappa_H t}{n}\left( \frac{1}{n} \boldsymbol{\Psi}_{\ell \mathsf{m}}^{\mathsf{T}} \boldsymbol{\Psi}_{\ell \mathsf{m}} \right).$$

Therefore from Eq. (30) we have

$$T_{2332} \leq \frac{t}{n^2} \widehat{\boldsymbol{f}}_{\ell \mathsf{m}}^{\mathsf{T}} \boldsymbol{\Psi}_{\ell \mathsf{m}}^{\mathsf{T}} \boldsymbol{\Psi}_{\leq \ell} \boldsymbol{D}_{\leq \ell}^2 \boldsymbol{\Psi}_{\leq \ell}^{\mathsf{T}} \boldsymbol{\Psi}_{\ell \mathsf{m}} \widehat{\boldsymbol{f}}_{\ell \mathsf{m}} + \widehat{\boldsymbol{f}}_{\ell \mathsf{m}}^{\mathsf{T}} \frac{\kappa_H t}{n}\left( \frac{1}{n} \boldsymbol{\Psi}_{\ell \mathsf{m}}^{\mathsf{T}} \boldsymbol{\Psi}_{\ell \mathsf{m}} \right) \widehat{\boldsymbol{f}}_{\ell \mathsf{m}} + o_{d,\mathbb{P}}(1) \cdot \|\mathsf{P}_{\ell \mathsf{m}} f_d\|_{L^2}^2 .$$

For the second term on the right, by Assumption 3(b) and by Lemma 4,

$$\widehat{\boldsymbol{f}}_{\ell \mathsf{m}}^{\mathsf{T}} \frac{\kappa_H t}{n}\left( \frac{1}{n} \boldsymbol{\Psi}_{\ell \mathsf{m}}^{\mathsf{T}} \boldsymbol{\Psi}_{\ell \mathsf{m}} \right) \widehat{\boldsymbol{f}}_{\ell \mathsf{m}} = o_{d,\mathbb{P}}(1) \cdot \|\mathsf{P}_{\ell \mathsf{m}} f_d\|_{L^2}^2 .$$

For the first term by Lemma 7 and Assumptions 3(b), 3(c),

$$\frac{t}{n^2}\mathbb{E}[\widehat{\boldsymbol{f}}_{\ell\mathsf{m}}^{\mathsf{T}}\boldsymbol{\Psi}_{\ell\mathsf{m}}^{\mathsf{T}}\boldsymbol{\Psi}_{\leq\ell}\boldsymbol{D}_{\leq\ell}^2\boldsymbol{\Psi}_{\leq\ell}^{\mathsf{T}}\boldsymbol{\Psi}_{\ell\mathsf{m}}\widehat{\boldsymbol{f}}_{\ell\mathsf{m}}] \leq (t/n)C(\eta)\left\|\mathsf{P}_{\ell\mathsf{m}}f_d\right\|_{L^{2+\eta}}^2 \operatorname{Tr}\left(\boldsymbol{D}_{\leq\ell}^2\right)$$

$$= o_{d,\mathbb{P}}(1) \cdot \left\|\mathsf{P}_{\ell\mathsf{m}}f_d\right\|_{L^{2+\eta}}^2,$$

therefore by Markov's inequality

$$T_{2332} = o_{d,\mathbb{P}}(1) \cdot \left\|\mathsf{P}_{\ell\mathsf{m}}f_d\right\|_{L^{2+\eta}}^2.$$

Hence combining terms

$$T_{233} = T_{2331} + T_{2332} = o_{d,\mathbb{P}}(1) \cdot \left\|\mathsf{P}_{\ell\mathsf{m}}f_d\right\|_{L^{2+\eta}}^2.$$

By the Cauchy-Schwarz inequality

$$T_{232} = \frac{2}{n^2}\boldsymbol{f}_{\ell\mathsf{m}}^{\mathsf{T}}\boldsymbol{G}\boldsymbol{f}_{>\mathsf{m}}$$

$$\leq 2\left(\frac{1}{n^2}\boldsymbol{f}_{\ell\mathsf{m}}^{\mathsf{T}}\boldsymbol{G}\boldsymbol{f}_{\ell\mathsf{m}}\frac{1}{n^2}\boldsymbol{f}_{>\mathsf{m}}^{\mathsf{T}}\boldsymbol{G}\boldsymbol{f}_{>\mathsf{m}}\right)^{1/2}$$

$$= 2\left(T_{231} + o_{d,\mathbb{P}}(1)\cdot\left\|\mathsf{P}_{>\ell}f_d\right\|_{L^2}^2\right)^{1/2}\left(T_{233} + o_{d,\mathbb{P}}(1)\cdot\left\|\mathsf{P}_{>\ell}f_d\right\|_{L^2}^2\right)^{1/2}$$

$$\leq o_{d,\mathbb{P}}(1)\cdot\left\|\mathsf{P}_{>\ell}f_d\right\|_{L^{2+\eta}}\left\|\mathsf{P}_{>\mathsf{m}}f_d\right\|_{L^{2+\eta}}.$$

Putting everything together we see that

$$T_{23} = T_{231} + T_{232} + T_{233} + T_{234} = o_{d,\mathbb{P}}(1)\cdot\left\|f_d\right\|_{L^{2+\eta}}^2.$$

$\square$

### B.4.2   TERM $T_1$

Now we will analyse term $T_1$.

**Proposition 2** (Term $T_1$)**.**

$$T_1 = \boldsymbol{f}^{\mathsf{T}}(\mathbf{I}_n - e^{-t\boldsymbol{H}/n})\boldsymbol{H}^{-1}\boldsymbol{E} = \left\|\boldsymbol{S}_{\leq\ell}^{1/2}\widehat{\boldsymbol{f}}_{\leq\ell}\right\|_2^2 + o_{d,\mathbb{P}}(1)\cdot\left\|f_d\right\|_{L^{2+\eta}}^2.$$

*Proof of Proposition 2.* We break $T_1$ into the following terms

$$T_1 = T_{11} + T_{12} + T_{13}$$

where

$$T_{11} = \boldsymbol{f}_{\leq\ell}^{\mathsf{T}}(\mathbf{I} - e^{-t\boldsymbol{H}/n})\boldsymbol{H}^{-1}\boldsymbol{E}_{\leq\mathsf{m}}, \tag{31}$$

$$T_{12} = \boldsymbol{f}_{>\ell}^{\mathsf{T}}(\mathbf{I} - e^{-t\boldsymbol{H}/n})\boldsymbol{H}^{-1}\boldsymbol{E}_{\leq\mathsf{m}}, \tag{32}$$

$$T_{13} = \boldsymbol{f}^{\mathsf{T}}(\mathbf{I} - e^{-t\boldsymbol{H}/n})\boldsymbol{H}^{-1}\boldsymbol{E}_{>\mathsf{m}}. \tag{33}$$

Using Lemma 11 and recalling Lemma 10,

$$T_{12} \leq (T_{23})^{1/2}\left\|\widehat{\boldsymbol{f}}_{\leq\mathsf{m}}\right\|_2 = o_{d,\mathbb{P}}(1)\cdot\left\|\mathsf{P}_{\leq\mathsf{m}}f_d\right\|_{L^2}\left\|f_d\right\|_{L^{2+\eta}},$$

where $T_{23}$ is as given in Eq. (27). From the analysis of $T_{11}$ in Lemma 12 and $T_{13}$ in Lemma 13 we combine everything to get Proposition 2

$$T_1 = \left\|\boldsymbol{S}_{\leq\ell}^{1/2}\widehat{\boldsymbol{f}}_{\leq\ell}\right\|_2^2 + o_{d,\mathbb{P}}(1)\cdot\left\|f_d\right\|_{L^{2+\eta}}^2.$$

$\square$

**Lemma 11.** *For a vector* $\boldsymbol{v} \in \mathbb{R}^n$,

$$\boldsymbol{v}^{\mathsf{T}}\boldsymbol{H}^{-1}\boldsymbol{E}_{\leq m} \leq (\boldsymbol{v}^{\mathsf{T}}\boldsymbol{H}^{-1}\boldsymbol{M}\boldsymbol{H}^{-1}\boldsymbol{v})^{1/2}\left\|\widehat{\boldsymbol{f}}_{\leq\mathsf{m}}\right\|_2$$

$$= \left(\frac{1}{n^2}\boldsymbol{v}^{\mathsf{T}}\boldsymbol{\Psi}_{\leq\mathsf{m}}\boldsymbol{S}_{\leq\mathsf{m}}^2\boldsymbol{\Psi}_{\leq\mathsf{m}}^{\mathsf{T}}\boldsymbol{v} + o_{d,\mathbb{P}}(1)\cdot\frac{1}{n}\left\|\boldsymbol{v}\right\|_2\right)^{1/2}\left\|\widehat{\boldsymbol{f}}_{\leq\mathsf{m}}\right\|_2.$$

*Proof of Lemma 11.* Recall that $\boldsymbol{E}_{\leq m} = \boldsymbol{\Psi}_{\leq m}\boldsymbol{D}^2_{\leq m}\widehat{\boldsymbol{f}}_{\leq m}$. By the Cauchy-Schwarz inequality,

$$\boldsymbol{v}^{\mathsf{T}}\boldsymbol{H}^{-1}\boldsymbol{E}_{\leq m} = \boldsymbol{v}^{\mathsf{T}}\boldsymbol{H}^{-1}\boldsymbol{\Psi}_{\leq m}\boldsymbol{D}^2_{\leq m}\widehat{\boldsymbol{f}}_{\leq m},$$

$$\leq (\boldsymbol{v}^{\mathsf{T}}\boldsymbol{H}^{-1}\boldsymbol{\Psi}_{\leq m}\boldsymbol{D}^4_{\leq m}\boldsymbol{\Psi}^{\mathsf{T}}_{\leq m}\boldsymbol{H}^{-1}\boldsymbol{v})^{1/2}\left\|\widehat{\boldsymbol{f}}_{\leq m}\right\|_2,$$

$$\leq (\boldsymbol{v}^{\mathsf{T}}\boldsymbol{H}^{-1}\boldsymbol{M}\boldsymbol{H}^{-1}\boldsymbol{v})^{1/2}\left\|\widehat{\boldsymbol{f}}_{\leq m}\right\|_2$$

$$= \left(\frac{1}{n^2}\boldsymbol{v}^{\mathsf{T}}\boldsymbol{\Psi}_{\leq m}\boldsymbol{S}^2_{\leq m}\boldsymbol{\Psi}^{\mathsf{T}}_{\leq m}\boldsymbol{v} + o_{d,\mathbb{P}}(1)\cdot\frac{1}{n}\|\boldsymbol{v}\|_2\right)^{1/2}\left\|\widehat{\boldsymbol{f}}_{\leq m}\right\|_2,$$

where the last equality follows from Lemma 3. $\qquad\square$

**Lemma 12** (Term $T_{11}$ Eq. (31)).

$$T_{11} = \left\|\boldsymbol{S}^{1/2}_{\leq \ell}\widehat{\boldsymbol{f}}_{\leq \ell}\right\|^2_2 + o_{d,\mathbb{P}}(1)\cdot\|\mathsf{P}_{\leq m}f_d\|^2_{L^2}.$$

*Proof of Lemma 12.* First note that by Lemma 8 for some $\boldsymbol{\Delta}$ such that $\|\boldsymbol{\Delta}\|_{\mathrm{op}} = o_{d,\mathbb{P}}(1)$,

$$T_{11} = T_{111} + T_{112}$$

where

$$T_{111} = \boldsymbol{f}^{\mathsf{T}}_{\leq \ell}(\mathbf{I} - e^{-(t/n)(\boldsymbol{\Psi}_{\leq \ell}\boldsymbol{D}^2_{\leq \ell}\boldsymbol{\Psi}^{\mathsf{T}}_{\leq \ell}+\kappa_H\mathbf{I}_n)})\boldsymbol{H}^{-1}\boldsymbol{E}_{\leq m},$$

$$T_{112} = \boldsymbol{f}^{\mathsf{T}}_{\leq \ell}\boldsymbol{\Delta}\boldsymbol{H}^{-1}\boldsymbol{E}_{\leq m}.$$

By Lemma 11,

$$T_{112} \leq \left(\|\boldsymbol{\Delta}\|^2_{\mathrm{op}}\,(\|\boldsymbol{f}_{\leq \ell}\|^2_2/n)\left\|n\boldsymbol{H}^{-1}\boldsymbol{M}\boldsymbol{H}^{-1}\right\|_{\mathrm{op}}\right)^{1/2}\left\|\widehat{\boldsymbol{f}}_{\leq m}\right\|_2 = o_{d,\mathbb{P}}(1)\cdot\|\mathsf{P}_{\leq m}f_d\|^2_{L^2}.$$

We now consider

$$T_{111} = T_{1111} - T_{1112}$$

where

$$T_{1111} = \boldsymbol{f}^{\mathsf{T}}_{\leq \ell}\boldsymbol{H}^{-1}\boldsymbol{E}_{\leq m}$$

$$T_{1112} = \alpha\boldsymbol{f}^{\mathsf{T}}_{\leq \ell}e^{-(t/n)\boldsymbol{\Psi}_{\leq \ell}\boldsymbol{D}^2_{\leq \ell}\boldsymbol{\Psi}^{\mathsf{T}}_{\leq \ell}}\boldsymbol{H}^{-1}\boldsymbol{E}_{\leq m}.$$

Note that by Lemma 5,

$$\left\|\boldsymbol{\Psi}^{\mathsf{T}}_{\leq \ell}\boldsymbol{H}^{-1}\boldsymbol{\Psi}_{\leq m}\boldsymbol{D}^2_{\leq m} - [\boldsymbol{S}_{\leq \ell};\mathbf{0}]\right\|_{\mathrm{op}} = o_{d,\mathbb{P}}(1),$$

where $\mathbf{0}$ is a $\ell \times (m - \ell)$ matrix of zeros. Hence for the first term $T_{1111}$,

$$T_{1111} = \widehat{\boldsymbol{f}}^{\mathsf{T}}_{\leq \ell}\boldsymbol{\Psi}^{\mathsf{T}}_{\leq \ell}\boldsymbol{H}^{-1}\boldsymbol{\Psi}_{\leq m}\boldsymbol{D}^2_{\leq m}\widehat{\boldsymbol{f}}_{\leq m} = \left\|\boldsymbol{S}^{1/2}_{\leq \ell}\widehat{\boldsymbol{f}}_{\leq \ell}\right\|^2_2 + o_{d,\mathbb{P}}(1)\cdot\|\mathsf{P}_{\leq m}f_d\|^2_{L^2}.$$

Define $\boldsymbol{H}_{\leq \ell} := \boldsymbol{\Psi}_{\leq \ell}\boldsymbol{D}^2_{\leq \ell}\boldsymbol{\Psi}^{\mathsf{T}}_{\leq \ell}$. For the second term $T_{1112}$, by Lemma 11

$$T_{1112} \leq \left(S + o_{d,\mathbb{P}}(1)\cdot\|\mathsf{P}_{\leq \ell}f_d\|^2_{L^2}\right)^{1/2}\left\|\widehat{\boldsymbol{f}}_{\leq m}\right\|_2, \tag{34}$$

where

$$S = \frac{1}{n^2}\boldsymbol{f}^{\mathsf{T}}_{\leq \ell}e^{-(t/n)\boldsymbol{H}_{\leq \ell}}\boldsymbol{\Psi}_{\leq m}\boldsymbol{S}^2_{\leq m}\boldsymbol{\Psi}^{\mathsf{T}}_{\leq m}e^{-(t/n)\boldsymbol{H}_{\leq \ell}}\boldsymbol{f}_{\leq \ell}.$$

Define $\boldsymbol{A} := \frac{1}{n}\boldsymbol{\Psi}^{\mathsf{T}}_{\leq \ell}e^{-(t/n)\boldsymbol{H}_{\leq \ell}}\boldsymbol{\Psi}_{\leq \ell}$. We have

$$S \leq \frac{1}{n^2}\boldsymbol{f}^{\mathsf{T}}_{\leq \ell}e^{-(t/n)\boldsymbol{H}_{\leq \ell}}\boldsymbol{\Psi}_{\leq m}\boldsymbol{\Psi}^{\mathsf{T}}_{\leq m}e^{-(t/n)\boldsymbol{H}_{\leq \ell}}\boldsymbol{f}_{\leq \ell}$$

$$= \frac{1}{n^2}\boldsymbol{f}^{\mathsf{T}}_{\leq \ell}e^{-(t/n)\boldsymbol{H}_{\leq \ell}}\boldsymbol{\Psi}_{\leq \ell}\boldsymbol{\Psi}^{\mathsf{T}}_{\leq \ell}e^{-(t/n)\boldsymbol{H}_{\leq \ell}}\boldsymbol{f}_{\leq \ell} + \frac{1}{n^2}\boldsymbol{f}^{\mathsf{T}}_{\leq \ell}e^{-(t/n)\boldsymbol{H}_{\leq \ell}}\boldsymbol{\Psi}_{\ell m}\boldsymbol{\Psi}^{\mathsf{T}}_{\ell m}e^{-(t/n)\boldsymbol{H}_{\leq \ell}}\boldsymbol{f}_{\leq \ell}$$

$$\leq \frac{1}{n^2}\boldsymbol{f}^{\mathsf{T}}_{\leq \ell}e^{-(t/n)\boldsymbol{H}_{\leq \ell}}\boldsymbol{\Psi}_{\leq \ell}\boldsymbol{\Psi}^{\mathsf{T}}_{\leq \ell}e^{-(t/n)\boldsymbol{H}_{\leq \ell}}\boldsymbol{f}_{\leq \ell} + \left\|\widehat{\boldsymbol{f}}_{\leq \ell}\right\|^2_2\cdot\left\|\boldsymbol{\Psi}^{\mathsf{T}}_{\leq \ell}\boldsymbol{\Psi}_{\ell m}/n\right\|^2_{\mathrm{op}}$$

$$= \widehat{\boldsymbol{f}}^{\mathsf{T}}_{\leq \ell}\boldsymbol{A}^2\widehat{\boldsymbol{f}}_{\leq \ell} + o_{d,\mathbb{P}}(1)\cdot\|\mathsf{P}_{\leq \ell}f_d\|^2_{L^2}$$

$$= o_{d,\mathbb{P}}(1)\cdot\|\mathsf{P}_{\leq \ell}f_d\|^2_{L^2},$$

since as noted before in Eq. (22), $\|\boldsymbol{A}\|_{\mathrm{op}} = o_{d,\mathbb{P}}(1)$. Therefore

$$T_{11} = T_{1111} + T_{1112} + T_{112} = \left\| \boldsymbol{S}_{\leq \ell}^{1/2} \widehat{\boldsymbol{f}}_{\leq \ell} \right\|_2^2 + o_{d,\mathbb{P}}(1) \cdot \|\mathsf{P}_{\leq \mathsf{m}} f_d\|_{L^2}^2 .$$

$\square$

**Lemma 13** (Term $T_{13}$ Eq. (33)).

$$T_{13} = o_{d,\mathbb{P}}(1) \cdot \|\mathsf{P}_{>\mathsf{m}} f_d\|_{L^2} \|f_d\|_{L^2} .$$

*Proof of Lemma 13.* We have

$$
\begin{aligned}
|T_{13}| &= |\boldsymbol{f}^\mathsf{T} (\mathbf{I} - e^{-t\boldsymbol{H}/n}) \boldsymbol{H}^{-1} \boldsymbol{E}_{>\mathsf{m}}| \\
&\leq \|\boldsymbol{f}\|_2 \left\| \boldsymbol{H}^{-1} \right\|_{\mathrm{op}} \|\boldsymbol{E}_{>\mathsf{m}}\|_2
\end{aligned}
$$

Note that we have $\mathbb{E}[\|\boldsymbol{f}\|_2^2] = n \|f_d\|_{L^2}^2$. Further by Eq. (17), we have $\left\| \boldsymbol{H}^{-1} \right\|_{\mathrm{op}} \leq 2/\kappa_H$ with high probability. Finally, recalling the definition of $\boldsymbol{E}_{>\mathsf{m}}$ from Eq. (16), we have

$$\mathbb{E}[\|\boldsymbol{E}_{>\mathsf{m}}\|^2] = n \sum_{k=\mathsf{m}+1}^{\infty} \lambda_{d,k}^4 \hat{f}_k^2 \leq n \left[ \max_{k \geq \mathsf{m}+1} \lambda_{d,k}^4 \right] \|\mathsf{P}_{>\mathsf{m}} f_d\|_{L^2}^2 .$$

As a result, we have

$$
\begin{aligned}
|T_{13}| &\leq O_{d,\mathbb{P}}(1) \cdot \|\mathsf{P}_{>\mathsf{m}} f_d\|_{L^2} \|f_d\|_{L^2} \, [n^2 \max_{k \geq \mathsf{m}+1} \lambda_{d,k}^4]^{1/2}/\kappa_H \\
&= O_{d,\mathbb{P}}(1) \cdot \|\mathsf{P}_{>\mathsf{m}} f_d\|_{L^2} \|f_d\|_{L^2} \, [n \max_{k \geq \mathsf{m}+1} \lambda_{d,k}^2] / \sum_{k \geq \mathsf{m}+1} \lambda_{d,k}^2 \\
&= o_{d,\mathbb{P}}(1) \cdot \|\mathsf{P}_{>\mathsf{m}} f_d\|_{L^2} \|f_d\|_{L^2} ,
\end{aligned}
$$

where the last equality used Eq. (13) in Assumption 2(a). $\square$

### B.4.3 Terms $T_3$, $T_4$, $T_5$

To analyse the terms $T_3$, $T_4$, $T_5$ we can adapt the corresponding steps for the proof of Theorem 4 in Mei et al. (2021a). For the following analysis we recall the definition of $\varepsilon(t)$ from Eq. (24).

**Lemma 14** (Term $T_3$).

$$T_3 = \varepsilon(t)^\mathsf{T} \boldsymbol{H}^{-1} \boldsymbol{M} \boldsymbol{H}^{-1} \varepsilon(t) = o_{d,\mathbb{P}}(1) \cdot \sigma_\varepsilon^2$$

*Proof of Lemma 14.*

$$
\begin{aligned}
\frac{1}{\sigma_\varepsilon^2} \mathbb{E}_\varepsilon[T_3] &= \mathrm{Tr}\Big( (\mathbf{I}_n - e^{-t\boldsymbol{H}/n})^2 \boldsymbol{H}^{-1} \boldsymbol{M} \boldsymbol{H}^{-1} \Big) \\
&\leq \mathrm{Tr}\big( \boldsymbol{H}^{-1} \boldsymbol{M} \boldsymbol{H}^{-1} \big) \\
&\overset{(a)}{=} \mathrm{Tr}\big( \boldsymbol{\Psi}_{\leq \mathsf{m}} \boldsymbol{S}_{\leq \mathsf{m}}^2 \boldsymbol{\Psi}_{\leq \mathsf{m}}^\mathsf{T}/n^2 \big) + o_{d,\mathbb{P}}(1) \\
&\overset{(b)}{\leq} \frac{1}{n^2} \mathrm{Tr}\big( \boldsymbol{\Psi}_{\leq \mathsf{m}} \boldsymbol{\Psi}_{\leq \mathsf{m}}^\mathsf{T} \big) + o_{d,\mathbb{P}}(1) \\
&\overset{(c)}{=} \frac{1}{n^2} n\mathsf{m}(1 + o_{d,\mathbb{P}}(1)) + o_{d,\mathbb{P}}(1) = o_{d,\mathbb{P}}(1),
\end{aligned}
$$

where $(a)$ used Lemma 3, $(b)$ used $\boldsymbol{S}_{\leq \mathsf{m}} \preceq \mathbf{I}_m$, and $(c)$ used Lemma 4 and Assumption 2(c). The lemma then follows from Markov's inequality. $\square$

**Lemma 15** (Term $T_4$).

$$T_4 = \varepsilon(t)^\mathsf{T} \boldsymbol{H}^{-1} \boldsymbol{E} = o_{d,\mathbb{P}}(1) \cdot (\sigma_\varepsilon^2 + \|f_d\|_{L^2}^2).$$

*Proof of Lemma 15.*

$$\frac{1}{\sigma_\varepsilon^2}\mathbb{E}_{\boldsymbol\varepsilon}[T_4^2] = \frac{1}{\sigma_\varepsilon^2}\mathbb{E}_{\boldsymbol\varepsilon}[\boldsymbol\varepsilon^\mathsf{T}(\mathbf{I}-e^{-t\boldsymbol{H}/n})\boldsymbol{H}^{-1}\boldsymbol{E}\boldsymbol{E}^\mathsf{T}\boldsymbol{H}^{-1}(\mathbf{I}-e^{-t\boldsymbol{H}/n})\boldsymbol\varepsilon]$$

$$= \boldsymbol{E}^\mathsf{T}\boldsymbol{H}^{-1}(\mathbf{I}-e^{-t\boldsymbol{H}/n})^2\boldsymbol{H}^{-1}\boldsymbol{E}$$

$$\le \boldsymbol{E}^\mathsf{T}\boldsymbol{H}^{-2}\boldsymbol{E}$$

Notice that $\boldsymbol{M} \succeq \boldsymbol\Psi_{\le L}\boldsymbol{D}_{\le L}^4\boldsymbol\Psi_{\le L}^\mathsf{T}$ for any $L \in \mathbb{N}$, by the decomposition of Eq. (16). Therefore

$$\sup_L \left\|\boldsymbol{D}_{\le L}^2\boldsymbol\Psi_{\le L}^\mathsf{T}\boldsymbol{H}^{-2}\boldsymbol\Psi_{\le L}\boldsymbol{D}_{\le L}^2\right\|_{\mathrm{op}} = \sup_L\left\|\boldsymbol{H}^{-1}\boldsymbol\Psi_{\le L}\boldsymbol{D}_{\le L}^4\boldsymbol\Psi_{\le L}^\mathsf{T}\boldsymbol{H}^{-1}\right\|_{\mathrm{op}} \tag{35}$$

$$\le \left\|\boldsymbol{H}^{-1}\boldsymbol{M}\boldsymbol{H}^{-1}\right\|_{\mathrm{op}} = o_{d,\mathbb{P}}(1),$$

where the last inequality follows from Lemma 3. Hence,

$$\boldsymbol{E}^\mathsf{T}\boldsymbol{H}^{-2}\boldsymbol{E} = \lim_{L\to\infty}\boldsymbol{E}_{\le L}^\mathsf{T}\boldsymbol{H}^{-2}\boldsymbol{E}_{\le L}^\mathsf{T}$$

$$\overset{(a)}{=} \lim_{L\to\infty}\widehat{\boldsymbol{f}}_{\le L}^\mathsf{T}[\boldsymbol{D}_{\le L}^2\boldsymbol\Psi_{\le L}^\mathsf{T}\boldsymbol{H}^{-2}\boldsymbol\Psi_{\le L}\boldsymbol{D}_{\le L}^2]\widehat{\boldsymbol{f}}_{\le L}$$

$$\overset{(b)}{\le} \limsup_{L\to\infty}\left\|\boldsymbol{D}_{\le L}^2\boldsymbol\Psi_{\le L}^\mathsf{T}\boldsymbol{H}^{-2}\boldsymbol\Psi_{\le L}\boldsymbol{D}_{\le L}^2\right\|_{\mathrm{op}} \cdot \lim_{L\to\infty}\left\|\widehat{\boldsymbol{f}}_{\le L}\right\|_2^2$$

$$\overset{(c)}{\le} o_{d,\mathbb{P}}(1)\cdot\|f_d\|_{L^2}^2\,,$$

where $(a)$ follows from the definition of $\boldsymbol{E}_{\le L}$, $(b)$ follows from the definition of operator norm, and $(c)$ follows from Eq. (35). Therefore we get

$$T_4 = o_{d,\mathbb{P}}(1)\cdot\sigma_\varepsilon\cdot\|f_d\|_{L^2} = o_{d,\mathbb{P}}(1)\cdot(\sigma_\varepsilon^2+\|f_d\|_{L^2}^2).$$

$\square$

**Lemma 16** (Term $T_5$).

$$T_5 = \boldsymbol\varepsilon(t)^\mathsf{T}\boldsymbol{H}^{-1}\boldsymbol{M}\boldsymbol{H}^{-1}\boldsymbol{v}(t) = o_{d,\mathbb{P}}(1)\cdot(\sigma_\varepsilon^2+\|f_d\|_{L^{2+\eta}}^2).$$

*Proof of Lemma 16.* We can write term $T_5$ as

$$T_5 = T_{51}+T_{52}$$

where

$$T_{51} = \boldsymbol\varepsilon(t)^\mathsf{T}\boldsymbol{H}^{-1}\boldsymbol{M}\boldsymbol{H}^{-1}(\mathbf{I}_n-e^{-t\boldsymbol{H}/n})\boldsymbol{f}_{\le\ell},$$

$$T_{52} = \boldsymbol\varepsilon(t)^\mathsf{T}\boldsymbol{H}^{-1}\boldsymbol{M}\boldsymbol{H}^{-1}(\mathbf{I}_n-e^{-t\boldsymbol{H}/n})\boldsymbol{f}_{>\ell}.$$

Note as in Eq. (35), that by Lemma 3 and Lemma 4,

$$\left\|\boldsymbol{M}^{1/2}\boldsymbol{H}^{-2}\boldsymbol{M}^{1/2}\right\|_{\mathrm{op}} = \left\|\boldsymbol{H}^{-1}\boldsymbol{M}\boldsymbol{H}^{-1}\right\|_{\mathrm{op}} = o_{d,\mathbb{P}}(1),$$

and taking the second moment of $T_{51}$ yields

$$\frac{1}{\sigma_\varepsilon^2}\mathbb{E}_{\boldsymbol\varepsilon}[T_{51}^2] \le \frac{1}{\sigma_\varepsilon^2}\mathbb{E}_{\boldsymbol\varepsilon}[\boldsymbol\varepsilon^\mathsf{T}\boldsymbol{H}^{-1}\boldsymbol{M}\boldsymbol{H}^{-1}(\mathbf{I}_n-e^{-t\boldsymbol{H}/n})\boldsymbol{f}_{\le\ell}\boldsymbol{f}_{\le\ell}^\mathsf{T}(\mathbf{I}_n-e^{-t\boldsymbol{H}/n})\boldsymbol{H}^{-1}\boldsymbol{M}\boldsymbol{H}^{-1}\boldsymbol\varepsilon]$$

$$= \boldsymbol{f}_{\le\ell}^\mathsf{T}(\mathbf{I}_n-e^{-t\boldsymbol{H}/n})[\boldsymbol{H}^{-1}\boldsymbol{M}\boldsymbol{H}^{-1}]^2(\mathbf{I}_n-e^{-t\boldsymbol{H}/n})\boldsymbol{f}_{\le\ell}$$

$$\le \left\|\boldsymbol{M}^{1/2}\boldsymbol{H}^{-2}\boldsymbol{M}^{1/2}\right\|_{\mathrm{op}}\left\|\boldsymbol{M}^{1/2}\boldsymbol{H}^{-1}(\mathbf{I}_n-e^{-t\boldsymbol{H}/n})\boldsymbol{f}_{\le\ell}\right\|_2^2$$

$$= o_{d,\mathbb{P}}(1)\cdot T_{21}$$

$$= o_{d,\mathbb{P}}(1)\cdot\|\mathsf{P}_{\le\ell}f_d\|_{L^2}^2\,,$$

where $T_{21}$ is as given in Eq. (25). Similarly we get that

$$\frac{1}{\sigma_\varepsilon^2}\mathbb{E}_{\boldsymbol\varepsilon}[T_{52}^2] = o_{d,\mathbb{P}}(1)\cdot T_{23} = o_{d,\mathbb{P}}(1)\cdot\|f_d\|_{L^{2+\eta}}^2\,,$$

where $T_{23}$ is as given in Eq. (27). By Markov's inequality we deduce that

$$T_5 = o_{d,\mathbb{P}}(1)\cdot\sigma_\varepsilon\cdot(\|\mathsf{P}_{\le\ell}f_d\|_{L^2}+\|f_d\|_{L^{2+\eta}}) = o_{d,\mathbb{P}}(1)\cdot(\sigma_\varepsilon^2+\|f_d\|_{L^{2+\eta}}^2).$$

$\square$

Finally putting Propositions 1, 2 and Lemmas 14, 15, 16 together for terms $T_2, T_1, T_3, T_4$ and $T_5$ respectively leads to the proof of Theorem 5

*Proof of Theorem 5.*

$$R(\hat{f}_t) = \|f_d\|_{L^2}^2 - 2T_1 + T_2 + T_3 - 2T_4 + 2T_5$$

$$= \|\mathsf{P}_{>\ell} f_d\|_{L^2}^2 + \left\|\widehat{\boldsymbol{f}}_\ell\right\|_2^2 - 2\left\|\boldsymbol{S}_{\leq\ell}\widehat{\boldsymbol{f}}_{\leq\ell}\right\|^2 + \left\|\boldsymbol{S}_{\leq\ell}\widehat{\boldsymbol{f}}_{\leq\ell}\right\|^2$$

$$+ o_{d,\mathbb{P}}(1) \cdot (\|f_d\|_{L^2}^2 + \|f_d\|_{L^{2+\eta}}^2 + \sigma_\varepsilon^2)$$

$$= \left\|(\mathbf{I} - \boldsymbol{S}_{\leq\ell})\widehat{\boldsymbol{f}}_{\leq\ell}\right\|_2^2 + \|\mathsf{P}_{>\ell} f_d\|_{L^2}^2 + o_{d,\mathbb{P}}(1) \cdot (\|f_d\|_{L^2}^2 + \|f_d\|_{L^{2+\eta}}^2 + \sigma_\varepsilon^2).$$

By Assumption 2(b), $\kappa_H/n = o_d(1) \cdot \max_{j\leq\ell} \lambda_{d,j}^2$ hence

$$\left\|(\mathbf{I} - \boldsymbol{S}_{\leq\ell})\widehat{\boldsymbol{f}}_{\leq\ell}\right\|_2^2 = o_{d,\mathbb{P}}(1) \cdot \|f_d\|_{L^2}^2$$

and as a result we obtain the first part of the theorem

$$R(\hat{f}_t) = \|\mathsf{P}_{>\ell} f_d\|_{L^2}^2 + o_{d,\mathbb{P}}(1) \cdot (\|f_d\|_{L^2}^2 + \|f_d\|_{L^{2+\eta}}^2 + \sigma_\varepsilon^2). \tag{36}$$

Now observe that similar to Eq. (15) we have the following decomposition

$$\left\|\hat{f}_t - \mathsf{P}_{\leq\ell} f_d\right\|_{L^2}^2 = \|\mathsf{P}_{\leq\ell} f_d\|_{L^2}^2 - 2\boldsymbol{u}(t)^\mathsf{T} \boldsymbol{H}^{-1} \boldsymbol{E}_{\leq\ell} + \boldsymbol{u}(t)^\mathsf{T} \boldsymbol{H}^{-1} \boldsymbol{M} \boldsymbol{H}^{-1} \boldsymbol{u}(t),$$

where $\boldsymbol{u}(t)$ is given in Eq. (14). Therefore we can write

$$\left\|\hat{f}_t - \mathsf{P}_{\leq\ell} f_d\right\|_{L^2}^2 = R(\hat{f}_t) - \|\mathsf{P}_{>\ell} f_d\|_{L^2}^2 + 2\boldsymbol{u}(t)^\mathsf{T} \boldsymbol{H}^{-1} \boldsymbol{E}_{>\ell}. \tag{37}$$

We now focus on the term

$$\boldsymbol{u}(t)^\mathsf{T} \boldsymbol{H}^{-1} \boldsymbol{E}_{>\ell} = \boldsymbol{v}(t)^\mathsf{T} \boldsymbol{H}^{-1} \boldsymbol{E}_{>\ell} + \boldsymbol{\varepsilon}(t)^\mathsf{T} \boldsymbol{H}^{-1} \boldsymbol{E}_{>\ell}.$$

By choosing $L = \ell$ in the proof of Lemma 15, it follows that

$$\boldsymbol{\varepsilon}(t)^\mathsf{T} \boldsymbol{H}^{-1} \boldsymbol{E}_{\leq\ell} = o_{d,\mathbb{P}}(1) \cdot (\sigma_\varepsilon^2 + \|\mathsf{P}_{\leq\ell} f_d\|_{L^2}^2),$$

hence combining with the bound for $T_4$ in Lemma 15 yields

$$\boldsymbol{\varepsilon}(t)^\mathsf{T} \boldsymbol{H}^{-1} \boldsymbol{E}_{>\ell} = T_4 - \boldsymbol{\varepsilon}(t)^\mathsf{T} \boldsymbol{H}^{-1} \boldsymbol{E}_{\leq\ell}$$

$$= o_{d,\mathbb{P}}(1) \cdot (\sigma_\varepsilon^2 + \|f_d\|_{L^2}^2) - o_{d,\mathbb{P}}(1) \cdot (\sigma_\varepsilon^2 + \|\mathsf{P}_{\leq\ell} f_d\|_{L^2}^2)$$

$$= o_{d,\mathbb{P}}(1) \cdot (\sigma_\varepsilon^2 + \|f_d\|_{L^2}^2).$$

We will now show that

$$\boldsymbol{v}(t)^\mathsf{T} \boldsymbol{H}^{-1} \boldsymbol{E}_{>\ell} = o_{d,\mathbb{P}}(1) \cdot \|\mathsf{P}_{>\ell} f_d\|_{L^2} \|f_d\|_{L^2}. \tag{38}$$

If $\ell(d) = \mathsf{m}(d)$, then Eq. (38) follows from Lemma 13. Otherwise, consider the case $\ell(d) = \mathsf{u}(d)$. Following similar logic to the proof of Lemma 13, because $\mathbb{E}[\|\boldsymbol{f}\|_2^2] = n \|f_d\|_{L^2}^2$ and

$$\mathbb{E}[\|\boldsymbol{E}_{>\mathsf{u}}\|_2^2] = n \sum_{k=\mathsf{u}+1}^\infty \lambda_{d,k}^4 \hat{f}_k^2 \leq n\lambda_{d,\mathsf{u}+1}^4 \|\mathsf{P}_{>\mathsf{u}} f_d\|_{L^2}^2,$$

we also get Eq. (38) since

$$|\boldsymbol{v}(t)^\mathsf{T} \boldsymbol{H}^{-1} \boldsymbol{E}_{>\ell}| = |\boldsymbol{f}^\mathsf{T}(\mathbf{I} - e^{-t\boldsymbol{H}/n}\boldsymbol{H}^{-1})\boldsymbol{E}_{>\mathsf{u}}|$$

$$\leq (t/n)\|\boldsymbol{f}\|_2 \left\|(\mathbf{I} - e^{-t\boldsymbol{H}/n})(t\boldsymbol{H}/n)^{-1}\right\|_{\mathrm{op}} \|\boldsymbol{E}_{>\mathsf{u}}\|_2$$

$$\overset{(a)}{\leq} O_{d,\mathbb{P}}(1) \cdot \|\mathsf{P}_{>\mathsf{u}} f_d\|_{L^2} \|f_d\|_{L^2} t\lambda_{d,\mathsf{u}+1}^2$$

$$\overset{(b)}{=} o_{d,\mathbb{P}}(1) \cdot \|\mathsf{P}_{>\mathsf{u}} f_d\|_{L^2} \|f_d\|_{L^2},$$

where $(a)$ used the inequality $(1 - e^{-x})/x \leq 1$ and $(b)$ used Assumption 3(a). Therefore $\boldsymbol{u}(t)^\mathsf{T} \boldsymbol{H}^{-1} \boldsymbol{E}_{>\ell} = o_{d,\mathbb{P}}(1) \cdot (\sigma_\varepsilon^2 + \|f_d\|_{L^2}^2)$, hence by Eq. (36) and Eq. (37) we obtain the final part of the theorem

$$\left\|\hat{f}_t - \mathsf{P}_{\leq\ell} f_d\right\|_{L^2}^2 = o_{d,\mathbb{P}}(1) \cdot (\|f_d\|_{L^{2+\eta}}^2 + \sigma_\varepsilon^2). \tag{39}$$

$\square$

# C  DOT PRODUCT KERNELS ON $\mathbb{S}^{d-1}(\sqrt{d})$

## C.1  SETTING

We now apply our general theorems to the setting of dot product kernels on the sphere. Concretely we take $\mathcal{X}_d = \mathbb{S}^{d-1}(\sqrt{d})$ and $\nu_d = \mathrm{Unif}(\mathbb{S}^{d-1}(\sqrt{d}))$ and consider dot product kernels $H_d$ which take the form of Eq. (4). Note that by Eq. (61) any dot product kernel $h_d$ can be decomposed as

$$h_d(\langle \boldsymbol{x}_1, \boldsymbol{x}_2 \rangle / d) = \mathbb{E}_{\boldsymbol{w} \sim \mathrm{Unif}(\mathbb{S}^{d-1})}[\sigma_d(\langle \boldsymbol{w}, \boldsymbol{x}_1 \rangle)\sigma_d(\langle \boldsymbol{w}, \boldsymbol{x}_2 \rangle)]$$

for some activation function $\sigma_d$. We state mild assumptions on $\sigma_d$ and show that under these conditions we can apply the results in Section A.3.

## C.2  ASSUMPTIONS

We state our assumptions on $\sigma_d$ after some definitions. See Appendix G for additional background. Denote by $\overline{\mathsf{P}}_{\leq \ell}$ the orthogonal projection onto the subspace of $L^2(\mathbb{S}^{d-1}(\sqrt{d}))$ spanned by polynomials of degree less than or equal to $\ell$. The projectors $\overline{\mathsf{P}}_\ell$ and $\overline{\mathsf{P}}_{>\ell}$ are defined analogously. Let us emphasize that the projectors $\overline{\mathsf{P}}_{\leq \ell}$ are related but distinct from the $\mathsf{P}_{\leq \mathsf{m}}$: while $\overline{\mathsf{P}}_{\leq \ell}$ projects onto the eigenspace of polynomials of degree at most $\ell$, $\mathsf{P}_{\leq \mathsf{m}}$ projects onto the top $\mathsf{m}$-eigenfunctions.

The assumptions given on the activations are the same as Assumption 3 of Mei et al. (2021a).

**Assumption 4** (Assumptions for Dot Product Kernels at level $\mathsf{s} \in \mathbb{N}$). *Let $\{H_d\}_{d \geq 1}$ be a sequence of dot product kernels with associated activation functions $\{\sigma_d\}_{d \geq 1}$ as in Eq. (61). We assume the following hold*

    *(a) There exists an integer $k$ and constants $c_1 < 1$ and $c_0 > 0$, such that $|\sigma_d(x)| \leq c_0 \exp(c_1 x^2 / (4k))$.*

    *(b) We have*

$$\min_{k \leq \mathsf{s}} d^{\mathsf{s}-k} \left\| \overline{\mathsf{P}}_k \sigma_d(\langle \boldsymbol{e}, \cdot \rangle) \right\|_{L^2}^2 = \Omega_d(1),$$

$$\left\| \overline{\mathsf{P}}_{>2\mathsf{s}+1} \sigma_d(\langle \boldsymbol{e}, \cdot \rangle) \right\|_{L^2}^2 = \Omega_d(1),$$

    *where $\boldsymbol{e} \in \mathbb{S}^{d-1}$ is a fixed vector (it is easy to see that these quantities do not depend on $\boldsymbol{e}$).*

Consider $t(d), n(d)$ such that

$$d^{\mathsf{j}+\delta_0} \leq t \leq d^{\mathsf{j}+1-\delta_0}, \quad d^{\mathsf{s}+\delta_0} \leq n \leq d^{\mathsf{s}+1-\delta_0}$$

for some $\mathsf{j}, \mathsf{s} \in \mathbb{N}$ and $\delta_0 > 0$. We now verify that if $\{\sigma_d\}_{d \geq 1}$ satisfies Assumption 4 at level $\mathsf{s}$, then for an appropriate choice of $(\mathsf{u}(d), \mathsf{m}(d))$ the conditions in Appendix A.2 are satisfied and lead to Theorem 1. We set $\mathsf{u}(d)$ and $\mathsf{m}(d)$ to be the number of eigenvalues associated to spherical harmonics of degree less than or equal to $\mathsf{j}$ and $\mathsf{s}$ respectively

$$\mathsf{u} = \sum_{k=0}^{\mathsf{j}} B(d, k) = \Theta_d(d^{\mathsf{j}}), \quad \mathsf{m} = \sum_{k=0}^{\mathsf{s}} B(d, k) = \Theta_d(d^{\mathsf{s}}).$$

The verification of Assumption 1 (Kernel Concentration Property) and Assumption 2 (Eigenvalue Condition) at level $\{(n(d), \mathsf{m}(d)\}$ is the same as the treatment in Theorem 2 of Mei et al. (2021a). We only need to verify Assumption 3. To see part 3(a), note that $1/\lambda_{d,\mathsf{u}(d)}^2 = \Theta_d(d^{\mathsf{j}})$ and $1/\lambda_{d,\mathsf{u}(d)}^2 = \Theta_d(d^{\mathsf{j}+1})$. For part 3(b), the condition holds because $\mathsf{u}(d) < \mathsf{m}(d)$ for large $d$ if and only if $\mathsf{j} < \mathsf{s}$. Assumption 3(c) is easily seen to hold since the trace of the kernel operator $\mathrm{Tr}(\mathbb{H}_d) = \Theta_d(1)$.

# D  GROUP INVARIANT KERNELS ON $\mathbb{S}^{d-1}(\sqrt{d})$

## D.1  SETTING

We now apply our general theorems to the setting of group invariant kernels on the sphere. Concretely we take $\mathcal{X}_d = \mathbb{S}^{d-1}(\sqrt{d})$ and $\nu_d = \mathrm{Unif}(\mathbb{S}^{d-1}(\sqrt{d}))$ and consider kernels $H_d$ which take the form of Eq. (5) for some function $h$. By Eq. (61), for some activation function $\sigma_d$

$$H_d(\boldsymbol{x}_1, \boldsymbol{x}_2) = \int_{\mathcal{G}_d} \mathbb{E}_{\boldsymbol{x} \sim \mathrm{Unif}(\mathbb{S}^{d-1})}[\sigma_d(\langle \boldsymbol{x}_1, \boldsymbol{w} \rangle)\sigma_d(\langle \boldsymbol{x}_2, g \cdot \boldsymbol{w} \rangle)]\pi_d(\mathrm{d}g). \tag{40}$$

We state mild assumptions on $\sigma_d$ and show that under these conditions we can apply the results in Appendix A.3. For additional technical background refer to Appendix G.

## D.2  ASSUMPTIONS

We will assume that $\sigma_d = \sigma$ for all $d$ and make the following assumptions on $\sigma$ which are the same as Assumption 1 in Mei et al. (2021b).

**Assumption 5** (Assumption on Group Invariant Kernel at level s). *Let $\{H_d\}_{d \geq 1}$ be a sequence of invariant kernels with associated activation functions $\sigma_d = \sigma$ as in Eq. (40). We assume the following conditions hold*

*(a)  For $\mathcal{G}_d = \mathrm{Cyc}_d$, we assume $\sigma$ to be $(\mathsf{s} + 1) \vee 3$ differentiable and there exists constants $c_0 > 0$ and $c_1 < 1$ such that $|\sigma^{(k)}| \leq c_0 e^{c_1 u^2/2}$ for any $2 \leq k \leq (\mathsf{s}+1) \vee 3$.*

*For general $\mathcal{G}_d$, we assume that $\sigma$ is a (finite degree) polynomial function.*

*(b)  The Hermite coefficients $\mu_k(\sigma)$ (c.f. Appendix) verify $\mu_k \neq 0$ for any $0 \leq k \leq \mathsf{s}$.*

*(c)  We assume that $\sigma$ is not a polynomial with degrees less than or equal to $\mathsf{s}$.*

Consider $t(d)$, $n(d)$ such that

$$d^{\mathsf{j}+\delta_0} \leq t \leq d^{\mathsf{j}+1-\delta_0}, \quad d^{\mathsf{s}-\alpha+\delta_0} \leq n \leq d^{\mathsf{s}-\alpha+1-\delta_0}$$

for some $\mathsf{j}, \mathsf{s} \in \mathbb{N}$ and $\delta_0 > 0$. We now verify that if $\sigma$ satisfies Assumption 5 at level $\mathsf{s}$, then the conditions given in Appendix A.2 are satisfied for an appropriate choice of $(\mathsf{u}(d), \mathsf{m}(d))$ which leads to Theorem 2. We set $\mathsf{u}$ and $\mathsf{m}$ to be the number of eigenvalues invariant polynomials of degree less than or equal to $\mathsf{j}$ and $\mathsf{s}$ respectively

$$\mathsf{u} = \sum_{k=0}^{\mathsf{j}} D(d, k) = \Theta_d(d^{\mathsf{j}-\alpha}), \quad \mathsf{m} = \sum_{k=0}^{\mathsf{s}} D(d, k) = \Theta_d(d^{\mathsf{s}-\alpha}),$$

where $D(d, k)$ is the dimension of the subspace of invariant polynomials of degree $k$, c.f. Appendix G.6. The verification of Assumption 1 (Kernel Concentration Property) and Assumption 2 (Eigenvalue Condition) at level $\{(n(d), \mathsf{m}(d)\}$ is exactly the same as in Theorem 1 in Mei et al. (2021b).

We must verify Assumption 3. To see part 3(a), note that $1/\lambda_{d,\mathsf{u}(d)}^2 = \Theta_d(d^{\mathsf{j}})$ and $1/\lambda_{d,\mathsf{u}(d)}^2 = \Theta_d(d^{\mathsf{j}+1})$. For part 3(b), the condition holds because $\mathsf{u}(d) < \mathsf{m}(d)$ for large $d$ if and only if $\mathsf{j} < \mathsf{s}$ in which case

$$\frac{t(d)}{n(d)} \mathrm{Tr}\big(\mathbb{H}_{d,\mathsf{m}(d)}\big) \leq \frac{d^{\mathsf{j}+1-\delta_0}}{d^{\mathsf{s}-\alpha+\delta_0}}\Theta(d^{-\alpha}) = O(d^{-2\delta_0}d^{\mathsf{j}-\mathsf{s}+1}) = o_d(1).$$

Assumption 3(c) can be seen to hold from the fact that $\mathrm{Tr}\big(\mathbb{H}_{d,>\mathsf{m}(d)}\big) = \Theta(d^{\mathsf{s}-\alpha})$ and

$$\sum_{j=0}^{\mathsf{m}} \lambda_{d,j}^2 = \sum_{k=0}^{\mathsf{s}} \xi_{d,k}^2 D(d, k) = \Theta(\mathsf{s}d^{-\alpha}) = \Theta(d^{-\alpha})$$

from which it follows that for some constant $C$

$$\sum_{j=0}^{\mathsf{m}} \lambda_{d,j}^2 \leq C \sum_{j>\mathsf{m}}^{\infty} \lambda_{d,j}^2.$$

# E AUXILIARY RESULTS

## E.1 SOLUTION TO KERNEL DYNAMICS

Recall that we are interested in the following dynamics given in Eqs. (2), (3),

$$\frac{\mathrm{d}}{\mathrm{d}t} f_t^{\mathrm{or}}(\boldsymbol{x}) = \mathbb{E}[H_d(\boldsymbol{x}, \boldsymbol{z})(f_d(\boldsymbol{z}) - f_t^{\mathrm{or}}(\boldsymbol{z}))],$$

$$\frac{\mathrm{d}}{\mathrm{d}t} \hat{f}_t(\boldsymbol{x}) = \frac{1}{n} \sum_{i=1}^{n} H_d(\boldsymbol{x}, \boldsymbol{x}_i)(y_i - \hat{f}_t(\boldsymbol{x}_i))$$

with zero initialization $f_0^{\mathrm{or}} \equiv \hat{f}_0 \equiv 0$. In this section we clarify the derivation, validity, and solution of these dynamics. Let us consider the maps $R : \mathcal{H}_d \to \mathbb{R}$ and $\widehat{R}_n : \mathcal{H}_d \to \mathbb{R}$ defined in Eq. (1).

$$R(f) = \int_{\mathcal{X}_d} (f(\boldsymbol{x}) - f_d(\boldsymbol{x}))^2 \, \mathrm{d}\nu_d(\boldsymbol{x}) + \sigma_\varepsilon^2,$$

$$\widehat{R}_n(f) = \frac{1}{n} \sum_{i=1}^{n} (f(\boldsymbol{x}_i) - y_i)^2.$$

First, we recall the definition of the Fréchet derivative of a functional $V : \mathcal{H}_d \to \mathbb{R}$ at $f$. The Fréchet derivative $DV(f)$ is the linear functional such that for $g \in \mathcal{H}_d$,

$$\lim_{\|g\|_{\mathcal{H}_d} \to 0} \frac{|V(f+g) - V(f) - DV(f)(g)|}{\|g\|_{\mathcal{H}_d}} = 0.$$

The gradient $\boldsymbol{\nabla} V(f) \in \mathcal{H}_d$ is defined such that

$$\langle \boldsymbol{\nabla} V(f), g \rangle_{\mathcal{H}_d} = DV(f)(g)$$

exists uniquely by the Riesz representation theorem. The gradients of the risk functionals are

$$\boldsymbol{\nabla} R(f) = \mathbb{H}_d(f - f_d),$$

$$\boldsymbol{\nabla} \widehat{R}_n(f) = \frac{1}{n} \sum_{i=1}^{n} (f(\boldsymbol{x}_i) - y_i) H_{\boldsymbol{x}_i},$$

where $H_{\boldsymbol{x}_i}(\boldsymbol{x}) := H_d(\boldsymbol{x}_i, \boldsymbol{x})$ and $\mathbb{H}_d$ is the kernel operator as in Section A.1. A proof of this fact is given in Proposition 2.1 of Yao et al. (2007). Taking $f_t^{\mathrm{or}}(x)$ as shorthand for $f^{\mathrm{or}}(t, x)$ where $f^{\mathrm{or}}(t, \cdot) \in \mathcal{H}_d$ is the oracle model at time $t$ and similarly for $\hat{f}_t(x)$, the following gradient flows with zero initialization are well-defined for $t \geq 0$,

$$\frac{\mathrm{d}}{\mathrm{d}t} f_t^{\mathrm{or}} = -\boldsymbol{\nabla} R(f_t^{\mathrm{or}}) = -\mathbb{H}_d(f_t^{\mathrm{or}} - f_d) = \mathbb{E}_{\boldsymbol{z}}[H_d(\cdot, \boldsymbol{z})(f_d(\boldsymbol{z}) - f_t^{\mathrm{or}}(\boldsymbol{z}))] \tag{41}$$

$$\frac{\mathrm{d}}{\mathrm{d}t} \hat{f}_t = -\boldsymbol{\nabla} \widehat{R}_n(\hat{f}_t) = -\frac{1}{n} \sum_{i=1}^{n} (\hat{f}_t(\boldsymbol{x}_i) - y_i) H_{\boldsymbol{x}_i} = \frac{1}{n} \sum_{i=1}^{n} H_d(\cdot, \boldsymbol{x}_i)(y_i - \hat{f}_t(\boldsymbol{x}_i)). \tag{42}$$

The oracle model ODE Eq. (41) is simply a linear differential equation which has the following solution involving the operator exponential $\exp(\boldsymbol{A}) := \sum_{k=0}^{\infty} \boldsymbol{A}^k / k!$

$$f_t^{\mathrm{or}} = f_d + \exp(-t\mathbb{H}_d)(f_0^{\mathrm{or}} - f_d) = f_d - \exp(-t\mathbb{H}_d) f_d. \tag{43}$$

For the empirical model ODE Eq. (42) we first consider the system of scalar differential equations induced at the points $\{(\boldsymbol{x}_i, y_i)\}_{i \in [n]}$. Letting $\boldsymbol{u}(t) = (\hat{f}_t(\boldsymbol{x}_1), \dots, \hat{f}_t(\boldsymbol{x}_n))^\mathsf{T}$, $\boldsymbol{y} = (y_1, \dots, y_n)^\mathsf{T}$, and $\boldsymbol{H} = (H_d(\boldsymbol{x}_i, \boldsymbol{x}_j))_{i,j \in [n]}$ we have

$$\frac{\mathrm{d}}{\mathrm{d}t} \boldsymbol{u}(t) = -\frac{1}{n} \boldsymbol{H}(\boldsymbol{u}(t) - \boldsymbol{y}),$$

with initial condition $\boldsymbol{u}(0) = \boldsymbol{0}$. As this a linear ODE, the solution is given by

$$\boldsymbol{u}(t) = \boldsymbol{y} + e^{-t\boldsymbol{H}/n}(\boldsymbol{u}(0) - \boldsymbol{y}) = (\mathbf{I}_n - e^{-t\boldsymbol{H}/n})\boldsymbol{y}. \tag{44}$$

For $\boldsymbol{x} \in \mathbb{R}^d$, define $\boldsymbol{h}(\boldsymbol{x}) = (H_d(\boldsymbol{x}, \boldsymbol{x}_1), \ldots, H_d(\boldsymbol{x}, \boldsymbol{x}_n))^\mathsf{T} \in \mathbb{R}^n$. Let $\boldsymbol{a}(t) = \boldsymbol{H}^{-1}\boldsymbol{u}(t) \in \mathbb{R}^n$. We will show that the function $\hat{f}_t(\cdot) := \langle \boldsymbol{h}(\cdot), \boldsymbol{a}(t) \rangle \in \mathcal{H}_d$, which satisfies $(\hat{f}_t(\boldsymbol{x}_1), \ldots, \hat{f}_t(\boldsymbol{x}_n))^\mathsf{T} = \boldsymbol{u}(t)$, satisfies the following equation

$$\frac{\mathrm{d}}{\mathrm{d}t}\hat{f}_t(\boldsymbol{x}) = \frac{1}{n}\langle \boldsymbol{h}(\boldsymbol{x}), \boldsymbol{y} - \boldsymbol{u}(t) \rangle = \frac{1}{n}\langle \boldsymbol{h}(\boldsymbol{x}), e^{-t\boldsymbol{H}/n}\boldsymbol{y} \rangle,$$

which is Eq. (42) at point $\boldsymbol{x}$. Indeed, by the chain rule

$$\begin{aligned}
\frac{\mathrm{d}}{\mathrm{d}t}\hat{f}_t(\boldsymbol{x}) &= \frac{\mathrm{d}}{\mathrm{d}t}\langle \boldsymbol{h}(\boldsymbol{x}), \boldsymbol{a}(t) \rangle \\
&= \langle \boldsymbol{h}(\boldsymbol{x}), \frac{\mathrm{d}}{\mathrm{d}t}\boldsymbol{a}(t) \rangle \\
&= \langle \boldsymbol{h}(\boldsymbol{x}), \boldsymbol{H}^{-1}\frac{1}{n}\boldsymbol{H}e^{-t\boldsymbol{H}/n}\boldsymbol{y} \rangle \\
&= \frac{1}{n}\langle \boldsymbol{h}(\boldsymbol{x}), e^{-t\boldsymbol{H}/n}\boldsymbol{y} \rangle
\end{aligned}$$

which is what we wanted to show.

## E.2 Equivalence between Invariant Kernels and Data Augmentation

In this section we will show an equivalence between the (time rescaled) gradient flows for training invariant kernels and using an augmented dataset. Specifically consider a group $\mathcal{G}$ and a kernel $H$ that is $\mathcal{G}$-equivariant, that is

$$H(g \cdot \boldsymbol{x}_1, g \cdot \boldsymbol{x}_2) = H(\boldsymbol{x}_1, \boldsymbol{x}_2) \quad \forall g \in \mathcal{G}, \forall \boldsymbol{x}_1, \boldsymbol{x}_2 \in \mathcal{X}.$$

Given a $\mathcal{G}$-equivariant kernel $H$, we define a $\mathcal{G}$-invariant kernel $H_{\mathrm{inv}}$ as the group averaged kernel

$$H_{\mathrm{inv}} = \int_{\mathcal{G}} H(\boldsymbol{x}_1, g \cdot \boldsymbol{x}_2)\pi(\mathrm{d}g)$$

for the Haar measure $\pi$ on $\mathcal{G}$ (c.f. Eq. (5)). Note that any dot product kernel is $\mathcal{G}$-equivariant for $\mathcal{G}$ a subgroup the orthogonal group e.g. the cyclic group $\mathrm{Cyc}$ (c.f. Section 3.3).

Given a dataset $(\boldsymbol{X}, \boldsymbol{y}) = \{(\boldsymbol{x}_i, y_i) : i \in [n]\}$ consider the augmented dataset

$$(\boldsymbol{X}_{\mathcal{G}}, \boldsymbol{y}_{\mathcal{G}}) = \{(g \cdot \boldsymbol{x}_i, y_i) : g \in \mathcal{G}, i \in [n]\}.$$

We consider the (rescaled c.f. Remark 5) empirical dynamics Eq. (3) of the gradient flow on $(\boldsymbol{X}, \boldsymbol{y})$ using $H_{\mathrm{inv}}$ which we denote $\hat{f}_{t,\mathrm{inv}}$

$$\frac{\mathrm{d}}{\mathrm{d}t}\hat{f}_{t,\mathrm{inv}}(\boldsymbol{x}) = -m\boldsymbol{\nabla}\widehat{R}_n(\hat{f}_{t,\mathrm{inv}}) = -\frac{m}{n}\sum_{i=1}^{n}(\hat{f}_{t,\mathrm{inv}}(\boldsymbol{x}_i) - y_i)H_{\mathrm{inv}}(\boldsymbol{x}_i, \boldsymbol{x})$$

and the empirical dynamics of the gradient flow on $(\boldsymbol{X}_{\mathcal{G}}, \boldsymbol{y}_{\mathcal{G}})$ using $H$ which we denote $\hat{f}_{t,\mathrm{aug}}$

$$\frac{\mathrm{d}}{\mathrm{d}t}\hat{f}_{t,\mathrm{aug}}(\boldsymbol{x}) = -\frac{1}{n}\sum_{g \in \mathcal{G}}\sum_{i=1}^{n}(\hat{f}_{t,\mathrm{aug}}(g \cdot \boldsymbol{x}_i) - y_i)H(g \cdot \boldsymbol{x}_i, \boldsymbol{x}).$$

**Proposition 3.** *Let $\mathcal{G}$ be a finite group with $m$ elements. Given a $\mathcal{G}$-equivariant kernel $H$, if $\pi$ is the uniform measure on $\mathcal{G}$ then*

$$\hat{f}_{t,\mathrm{inv}} \equiv \hat{f}_{t,\mathrm{aug}}, \quad \forall t \geq 0.$$

*Proof.* Let $\mathcal{G} = \{g_1, \ldots, g_m\}$ where $g_1$ is the identity. Define the output vectors

$$\begin{aligned}
\boldsymbol{u}_{\mathrm{inv}}(t) &:= (\hat{f}_{t,\mathrm{inv}}(\boldsymbol{x}_1), \ldots, \hat{f}_{t,\mathrm{inv}}(\boldsymbol{x}_n))^\mathsf{T} \in \mathbb{R}^n, \\
\boldsymbol{u}_g(t) &:= (\hat{f}_{t,\mathrm{aug}}(g \cdot \boldsymbol{x}_1), \ldots, \hat{f}_{t,\mathrm{aug}}(g \cdot \boldsymbol{x}_n))^\mathsf{T} \in \mathbb{R}^n, \\
\boldsymbol{u}_{\mathrm{aug}}(t) &:= (\boldsymbol{u}_{g_1}(t), \ldots, \boldsymbol{u}_{g_m}(t))^\mathsf{T} \in \mathbb{R}^{mn}.
\end{aligned}$$

Furthermore, define the kernel matrices

$$\boldsymbol{H}_{g,g'} := [H(g \cdot \boldsymbol{x}_i, g' \cdot \boldsymbol{x}_j)]_{i,j \in [n]} \in \mathbb{R}^{n \times n} \text{ for } g, g' \in \mathcal{G},$$
$$\boldsymbol{H}_{\mathrm{aug}} := [\boldsymbol{H}_{g,g'}]_{g,g' \in \mathcal{G}} \in \mathbb{R}^{mn \times mn},$$
$$\boldsymbol{H}_{\mathrm{inv}} = [H_{\mathrm{inv}}(\boldsymbol{x}_i, \boldsymbol{x}_j)]_{i,j \in [n]}.$$

Note that by definition $\boldsymbol{H}_{\mathrm{inv}} = \frac{1}{m} \sum_{j=1}^{m} \boldsymbol{H}_{g_1,g_j}$. We will show that

$$\boldsymbol{u}_{\mathrm{aug}}(t) = (\boldsymbol{u}_{\mathrm{inv}}(t), \boldsymbol{u}_{\mathrm{inv}}(t), \ldots, \boldsymbol{u}_{\mathrm{inv}}(t)) \text{ for all } t \geq 0. \tag{45}$$

From this the result follows by Theorem 4.1 in Li et al. (2019) since $\hat{f}_{t,\mathrm{inv}}, \hat{f}_{t,\mathrm{aug}}$ are given by kernel regressions with targets $\boldsymbol{u}_{\mathrm{inv}}(t), \boldsymbol{u}_{\mathrm{aug}}(t)$ and kernels $H_{\mathrm{inv}}, H$ respectively.

By Eq. (42), we can write

$$\boldsymbol{u}_{\mathrm{inv}}(t) = (\mathbf{I} - \exp(-tm\boldsymbol{H}_{\mathrm{inv}}/n))\boldsymbol{y},$$
$$\boldsymbol{u}_{\mathrm{aug}}(t) = (\mathbf{I} - \exp(-t\boldsymbol{H}_{\mathrm{aug}}/n))\boldsymbol{y}_{\mathcal{G}},$$

where $\boldsymbol{y}_{\mathcal{G}} = (\boldsymbol{y}, \ldots, \boldsymbol{y}) \in \mathbb{R}^{mn}$. By expanding the matrix exponential series and using linearity, it suffices to show that

$$\boldsymbol{H}_{\mathrm{aug}}^k \boldsymbol{y}_{\mathcal{G}} = (m^k H_{\mathrm{inv}}^k \boldsymbol{y}, m^k H_{\mathrm{inv}}^k \boldsymbol{y}, \ldots, m^k H_{\mathrm{inv}}^k \boldsymbol{y}) \text{ for all } k \in \mathbb{N}.$$

in order to show Eq. (45) holds. We prove the above by induction on $k$. For $k = 1$, observe that

$$H_{\mathrm{aug}} \boldsymbol{y}_{\mathcal{G}} = \left( \sum_{j=1}^{m} \boldsymbol{H}_{g_i,g_j} \boldsymbol{y} \right)_{i=1}^{m}$$
$$= \left( \sum_{j=1}^{m} \boldsymbol{H}_{g_1,g_j} \boldsymbol{y} \right)_{i=1}^{m}$$
$$= (m\boldsymbol{H}_{\mathrm{inv}} \boldsymbol{y})_{i=1}^{m}$$

where the second equality follows from $\mathcal{G}$-equivariance of $H$. Assume the inductive hypothesis holds for $k$. Then

$$\boldsymbol{H}_{\mathrm{aug}}^{k+1} \boldsymbol{y}_{\mathcal{G}} = \boldsymbol{H}_{\mathrm{aug}} \boldsymbol{H}_{\mathrm{aug}}^k \boldsymbol{y}_{\mathcal{G}}$$
$$= \left( m^k \sum_{j=1}^{m} \boldsymbol{H}_{g_i,g_j} \boldsymbol{H}_{\mathrm{inv}}^k \boldsymbol{y} \right)_{i=1}^{m}$$
$$= \left( \sum_{j=1}^{m} \boldsymbol{H}_{g_1,g_j} \left( \sum_{j'=1}^{m} \boldsymbol{H}_{g_1,g_{j'}} \right)^k \boldsymbol{y} \right)_{i=1}^{m}$$
$$= \left( \left( \sum_{j=1}^{m} \boldsymbol{H}_{g_1,g_j} \right)^{k+1} \boldsymbol{y} \right)_{i=1}^{m}$$
$$= (m^{k+1} \boldsymbol{H}_{\mathrm{inv}}^{k+1} \boldsymbol{y})_{i=1}^{m}$$

where the second equality applies the induction hypothesis and the third equality uses equivariance. Thus the inductive claim is proved and the proof is complete.

$\square$

**Remark 5.** *The scaling factor $m$ in the gradient flow for $\hat{f}_{t,\mathrm{inv}}$, leads to a natural comparison with $\hat{f}_{t,\mathrm{aug}}$ as elaborated in Appendix E.3. As argued in that section, in the gradient descent discretization, it is natural to take a step-size inversely proportional to the maximum kernel eigenvalue. In the case of high-dimensional invariant kernels, note that*

$$\lambda_{\max}(\boldsymbol{H}_{\mathrm{aug}}) = m\lambda_{\max}(\boldsymbol{H}) \sim m\lambda_{\max}(\boldsymbol{H}_{\mathrm{inv}})$$

*hence the step-size for the invariant kernel flow should be $m$ times larger.*

### E.3 Discretizing Time

Comparing different "speeds" of optimization algorithms only makes sense for discrete-time algorithms. Consider the following gradient descent dynamics with step-size $\eta$, obtained as the discretization of the empirical gradient flow Eq. (42)

$$\hat{f}_{k+1} = \hat{f}_k - \eta \boldsymbol{\nabla} \widehat{R}_n(\hat{f}_k) = \hat{f}_k - \eta \frac{1}{n} \sum_{i=1}^{n} (\hat{f}_k(\boldsymbol{x}_i) - y_i) H(\cdot, \boldsymbol{x}_i), \quad k = 0, 1, \ldots \qquad (46)$$

We will argue that it is natural to take $\eta \sim n/\lambda_{\max}(\boldsymbol{H})$ where $\boldsymbol{H}$ is the kernel matrix.

Define the sampling operator $S : \mathcal{H}_d \to \mathbb{R}^n$ by $S(f) = (f(\boldsymbol{x}_i))_{i=1}^n \in \mathbb{R}^n$ and let $S^* : \mathbb{R}^n \to \mathcal{H}_d$ be its adjoint, defined by $S^*(\boldsymbol{y}) = \frac{1}{n} \sum_{i=1}^{n} y_i H_{\boldsymbol{x}_i}$ (see Yao et al. (2007) Appendix B for more details). Then we can rewrite the gradient descent equation Eq. (46) as

$$\hat{f}_{k+1} = \hat{f}_k - \eta(S^*S\hat{f}_t - S^*\boldsymbol{y}), \quad k = 0, 1, \ldots \qquad (47)$$

Let $T = S^*S$ and define $\overline{\boldsymbol{H}} := SS^* = \frac{1}{n}\boldsymbol{H}$ to be the normalized kernel matrix. Let $b := S^*\overline{\boldsymbol{H}}^{-1}\boldsymbol{y} \in \mathcal{H}_d$ and note that since $Tb = S^*\boldsymbol{y}$, we can rewrite Eq. (47) as

$$\hat{f}_{k+1} = \hat{f}_k - \eta T(\hat{f}_k - b), \quad k = 0, 1, \ldots \qquad (48)$$

Denote the eigenvalues of $\overline{\boldsymbol{H}}$ as $\lambda_1 \geq \lambda_2 \geq \ldots \geq \lambda_n > 0$. By the spectral theorem, there exists a set of orthornomal eigenvectors $\phi_1, \ldots, \phi_n \in \mathcal{H}_d$ such that $T = \sum_{i=1}^{n} \lambda_i \phi_i \phi_i^*$. Define $\alpha_i(k) = \langle \hat{f}_k - b, \phi_i \rangle \in \mathbb{R}$. By taking Eq. (48) then subtracting $b$ and taking the inner product with $\phi_i$ on both sides, we get the coordinate evolution equations for $i = 1, \ldots, n$

$$\alpha_i(k+1) = \langle (\mathrm{id} - \eta T)(\hat{f}_k - b), \phi_i \rangle$$
$$= \langle (\hat{f}_k - b), (\mathrm{id} - \eta T)\phi_i \rangle$$
$$= (1 - \eta \lambda_i)\alpha_i(k)$$

where the second equality holds since $T$ is self-adjoint and the last equality is since $\phi_i$ is an eigenvector of $T$. It is easy to see that $\alpha_i(k) = (1 - \eta \lambda_i)^k \alpha_i(0)$. Therefore we see that gradient descent Eq. (46) is guaranteed to converge if $\eta < 1/(2\lambda_1)$ and may not otherwise. Therefore it is natural to choose the step-size $\eta$ to scale asymptotically as $\eta \sim 1/\lambda_{\max}(\overline{\boldsymbol{H}})$.

For a dot product kernel $H$ and its corresponding invariant kernel $H_{\mathrm{inv}}$ the kernel matrices have operator norms of the same order

$$\lambda_{\max}(\boldsymbol{H}) \sim \lambda_{\max}(\boldsymbol{H}_{\mathrm{inv}})$$

hence no time rescaling is need to compare the corresponding optimization speeds asymptotically.

### E.4 Similarities with Empirical Phenomena

In this section we elaborate upon Remark 4 and mention some connections with empirical observations in Nakkiran et al. (2020). Although the metric in our setting is the squared loss, we can still observe three stages in classification problems when measuring the soft error. In Fig. 6a taken from Nakkiran et al. (2020) we can observe stage 1 and stage 2. Either training has not continued long enough to observe stage 3 or $n$ is large enough so that the models have converged to the approximation error of the neural network class (c.f. Remark 1). In Fig. 6b taken from Nakkiran et al. (2020), although the train errors are not plotted, by extrapolating from Fig. 6a, presumably for each $n$ stage 1 and stage 2 occur. From the dimmer curves in Fig. 6b we can see that for $n < 50000$ stage 3 occurs as well.

In Fig. 7a, we see a parallel between the use of cyclic versus dot product kernels and the use data augmentation versus not for a Resnet-18 trained on CIFAR-5m (note that using a cyclic kernel is equivalent to using a dot product kernel with data-augmentation c.f. Appendix E.2). In both our theoretical results and in the empirical results of Nakkiran et al. (2020) we observe that the ideal world optimization speed of augmented and non-augmented training are the same, but for augmented training the real world training speed is slowed down, eventually leading to better generalization for long enough training.

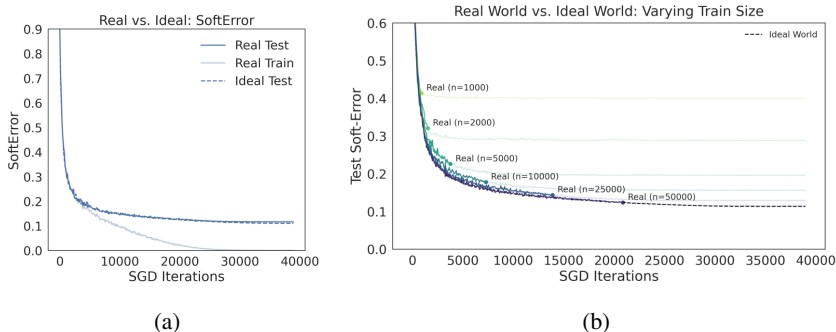

Figure 6: Soft-error curves for Resnet-18 trained on CIFAR-5m taken from ref. Nakkiran et al. (2020). Panel (6a): $n = 5 \times 10^4$ (Fig. 6 in ref). Panel (6b) Varying $n$ (Fig. 4a in ref).

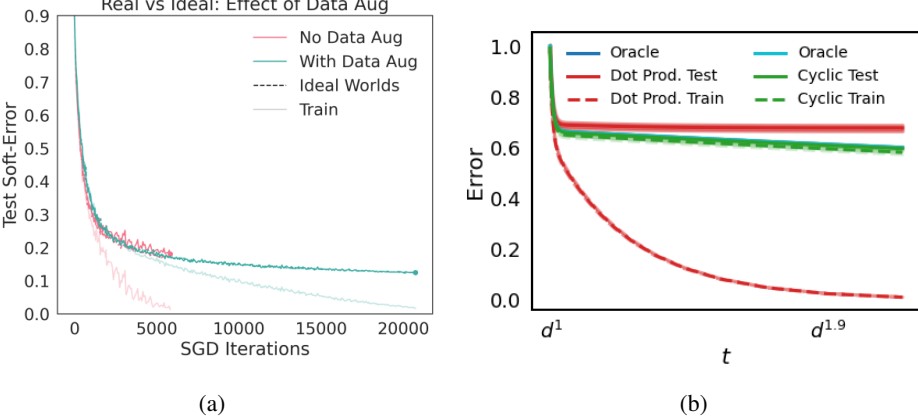

Figure 7: Panel (7a): Data-augmentation for Resnet-18 on CIFAR-5m. (Fig. 5a from Nakkiran et al. (2020)). Panel (7b): Cyclic versus dot product kernel (Fig. 5c from this work)

# F ADDITIONAL FIGURES

To see the effects of varying the dimension $d$, in Fig. 8 we replicate the log-scale plots of kernel gradient flow with dot product kernels from Fig. 4. We take $n = d^{1.5}$ and vary $d \in \{50, 100, 200, 400\}$. Each plot is averaged over 10 runs with the shaded region representing one standard deviation around the mean. We can see that as $d$ increasing the standard deviation decreases and the curves approach the theoretical high-dimensional prediction.

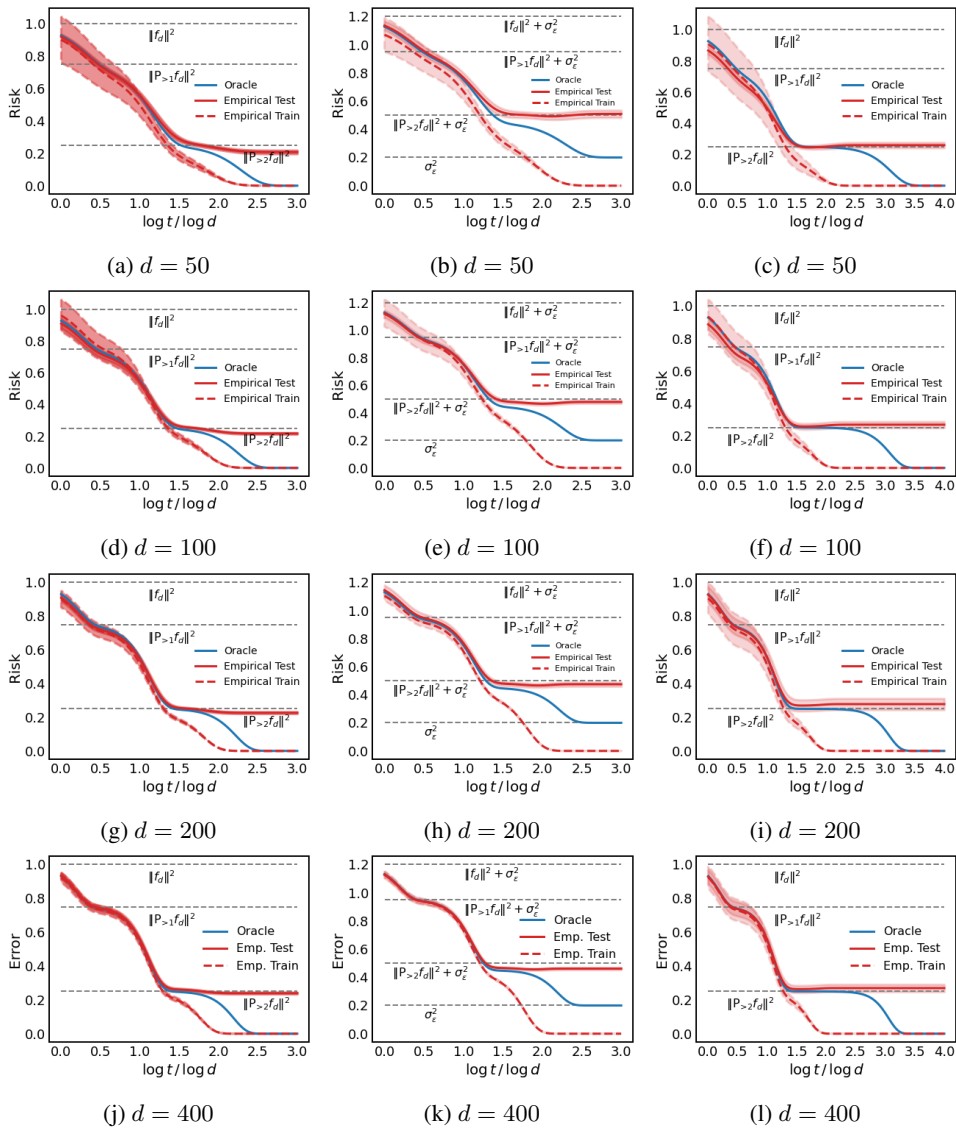

Figure 8: Each column replicates a kernel gradient flow experiment from Fig. 4 over $d \in \{50, 100, 200, 400\}$. **Column 1:** Replicates Fig. 4a **Column 2:** Replicates Fig. 4a but with $\sigma_\varepsilon^2 = 0.2$. **Column 3:** Replicates Fig. 4b. Averaged over 10 trials.

## G   TECHNICAL BACKGROUND

### G.1   NOTATIONS

For a positive integer, we denote by $[n]$ the set $\{1, 2, \ldots, n\}$. For vectors $\boldsymbol{u}, \boldsymbol{v} \in \mathbb{R}^d$, we denote $\langle \boldsymbol{u}, \boldsymbol{v} \rangle = u_1 v_1 + \ldots + u_d v_d$ their scalar product, and $\|\boldsymbol{u}\|_2 = \langle \boldsymbol{u}, \boldsymbol{u} \rangle^{1/2}$ the $\ell_2$ norm. Given a matrix $\boldsymbol{A} \in \mathbb{R}^{n \times m}$, we denote $\|\boldsymbol{A}\|_{\mathrm{op}} = \max_{\|\boldsymbol{u}\|_2 = 1} \|\boldsymbol{A}\boldsymbol{u}\|_2$ its operator norm and by $\|\boldsymbol{A}\|_F = \left( \sum_{i,j} A_{ij}^2 \right)^{1/2}$ its Frobenius norm. If $\boldsymbol{A} \in \mathbb{R}^{n \times n}$ is a square matrix, the trace of $\boldsymbol{A}$ is denoted by $\mathrm{Tr}(\boldsymbol{A}) = \sum_{i \in [n]} A_{ii}$.

We use $O_d(\,\cdot\,)$ (resp. $o_d(\,\cdot\,)$) for the standard big-O (resp. little-o) relations, where the subscript $d$ emphasizes the asymptotic variable. Furthermore, we write $f = \Omega_d(g)$ if $g(d) = O_d(f(d))$, and $f = \omega_d(g)$ if $g(d) = o_d(f(d))$. Finally, $f = \Theta_d(g)$ if we have both $f = O_d(g)$ and $f = \Omega_d(g)$.

We use $O_{d,\mathbb{P}}(\,\cdot\,)$ (resp. $o_{d,\mathbb{P}}(\,\cdot\,)$) the big-O (resp. little-o) in probability relations. Namely, for $h_1(d)$ and $h_2(d)$ two sequences of random variables, $h_1(d) = O_{d,\mathbb{P}}(h_2(d))$ if for any $\varepsilon > 0$, there exists $C_\varepsilon > 0$ and $d_\varepsilon \in \mathbb{Z}_{>0}$, such that

$$\mathbb{P}(|h_1(d)/h_2(d)| > C_\varepsilon) \le \varepsilon, \qquad \forall d \ge d_\varepsilon,$$

and respectively: $h_1(d) = o_{d,\mathbb{P}}(h_2(d))$, if $h_1(d)/h_2(d)$ converges to 0 in probability. Similarly, we will denote $h_1(d) = \Omega_{d,\mathbb{P}}(h_2(d))$ if $h_2(d) = O_{d,\mathbb{P}}(h_1(d))$, and $h_1(d) = \omega_{d,\mathbb{P}}(h_2(d))$ if $h_2(d) = o_{d,\mathbb{P}}(h_1(d))$. Finally, $h_1(d) = \Theta_{d,\mathbb{P}}(h_2(d))$ if we have both $h_1(d) = O_{d,\mathbb{P}}(h_2(d))$ and $h_1(d) = \Omega_{d,\mathbb{P}}(h_2(d))$.

### G.2   FUNCTIONAL SPACES OVER THE SPHERE

For $d \ge 3$, we let $\mathbb{S}^{d-1}(r) = \{\boldsymbol{x} \in \mathbb{R}^d : \|\boldsymbol{x}\|_2 = r\}$ denote the sphere with radius $r$ in $\mathbb{R}^d$. We will mostly work with the sphere of radius $\sqrt{d}$, $\mathbb{S}^{d-1}(\sqrt{d})$ and will denote by $\tau_d$ the uniform probability measure on $\mathbb{S}^{d-1}(\sqrt{d})$. All functions in this section are assumed to be elements of $L^2(\mathbb{S}^{d-1}(\sqrt{d}), \tau_d)$, with scalar product and norm denoted as $\langle \cdot, \cdot \rangle_{L^2}$ and $\| \cdot \|_{L^2}$:

$$\langle f, g \rangle_{L^2} \equiv \int_{\mathbb{S}^{d-1}(\sqrt{d})} f(\boldsymbol{x}) \, g(\boldsymbol{x}) \, \tau_d(\mathrm{d}\boldsymbol{x}) \,. \tag{49}$$

For $\ell \in \mathbb{Z}_{\ge 0}$, let $\tilde{V}_{d,\ell}$ be the space of homogeneous harmonic polynomials of degree $\ell$ on $\mathbb{R}^d$ (i.e. homogeneous polynomials $q(\boldsymbol{x})$ satisfying $\Delta q(\boldsymbol{x}) = 0$), and denote by $V_{d,\ell}$ the linear space of functions obtained by restricting the polynomials in $\tilde{V}_{d,\ell}$ to $\mathbb{S}^{d-1}(\sqrt{d})$. With these definitions, we have the following orthogonal decomposition

$$L^2(\mathbb{S}^{d-1}(\sqrt{d}), \tau_d) = \bigoplus_{\ell=0}^{\infty} V_{d,\ell} \,. \tag{50}$$

The dimension of each subspace is given by

$$\dim(V_{d,\ell}) = B(d,\ell) = \frac{2\ell + d - 2}{d - 2} \binom{\ell + d - 3}{\ell} \,. \tag{51}$$

For each $\ell \in \mathbb{Z}_{\ge 0}$, the spherical harmonics $\{Y_{\ell,j}^{(d)}\}_{1 \le j \le B(d,\ell)}$ form an orthonormal basis of $V_{d,\ell}$:

$$\langle Y_{ki}^{(d)}, Y_{sj}^{(d)} \rangle_{L^2} = \delta_{ij} \delta_{ks}.$$

Note that our convention is different from the more standard one, that defines the spherical harmonics as functions on $\mathbb{S}^{d-1}(1)$. It is immediate to pass from one convention to the other by a simple scaling. We will drop the superscript $d$ and write $Y_{\ell,j} = Y_{\ell,j}^{(d)}$ whenever clear from the context.

We denote by $\overline{\mathsf{P}}_k$ the orthogonal projections to $V_{d,k}$ in $L^2(\mathbb{S}^{d-1}(\sqrt{d}), \tau_d)$. This can be written in terms of spherical harmonics as

$$\overline{\mathsf{P}}_k f(\boldsymbol{x}) \equiv \sum_{l=1}^{B(d,k)} \langle f, Y_{kl} \rangle_{L^2} Y_{kl}(\boldsymbol{x}). \tag{52}$$

We also define $\overline{\mathsf{P}}_{\le \ell} \equiv \sum_{k=0}^{\ell} \overline{\mathsf{P}}_k, \overline{\mathsf{P}}_{>\ell} \equiv \mathbf{I} - \overline{\mathsf{P}}_{\le \ell} = \sum_{k=\ell+1}^{\infty} \overline{\mathsf{P}}_k$, and $\overline{\mathsf{P}}_{<\ell} \equiv \overline{\mathsf{P}}_{\le \ell-1}, \overline{\mathsf{P}}_{\ge \ell} \equiv \overline{\mathsf{P}}_{>\ell-1}$.

### G.3 GEGENBAUER POLYNOMIALS

The $\ell$-th Gegenbauer polynomial $Q_\ell^{(d)}$ is a polynomial of degree $\ell$. Consistently with our convention for spherical harmonics, we view $Q_\ell^{(d)}$ as a function $Q_\ell^{(d)} : [-d, d] \to \mathbb{R}$. The set $\{Q_\ell^{(d)}\}_{\ell \geq 0}$ forms an orthogonal basis on $L^2([-d, d], \tilde{\tau}_d^1)$, where $\tilde{\tau}_d^1$ is the distribution of $\sqrt{d}\langle x, e_1 \rangle$ when $x \sim \tau_d$, satisfying the normalization condition:

$$\langle Q_k^{(d)}(\sqrt{d}\langle e_1, \cdot \rangle), Q_j^{(d)}(\sqrt{d}\langle e_1, \cdot \rangle)\rangle_{L^2(\mathbb{S}^{d-1}(\sqrt{d}))} = \frac{1}{B(d, k)} \delta_{jk}. \tag{53}$$

In particular, these polynomials are normalized so that $Q_\ell^{(d)}(d) = 1$. As above, we will omit the superscript $(d)$ in $Q_\ell^{(d)}$ when clear from the context.

Gegenbauer polynomials are directly related to spherical harmonics as follows. Fix $v \in \mathbb{S}^{d-1}(\sqrt{d})$ and consider the subspace of $V_\ell$ formed by all functions that are invariant under rotations in $\mathbb{R}^d$ that keep $v$ unchanged. It is not hard to see that this subspace has dimension one, and coincides with the span of the function $Q_\ell^{(d)}(\langle v, \cdot \rangle)$.

We will use the following properties of Gegenbauer polynomials

1. For $x, y \in \mathbb{S}^{d-1}(\sqrt{d})$

$$\langle Q_j^{(d)}(\langle x, \cdot \rangle), Q_k^{(d)}(\langle y, \cdot \rangle)\rangle_{L^2} = \frac{1}{B(d, k)} \delta_{jk} Q_k^{(d)}(\langle x, y \rangle). \tag{54}$$

2. For $x, y \in \mathbb{S}^{d-1}(\sqrt{d})$

$$Q_k^{(d)}(\langle x, y \rangle) = \frac{1}{B(d, k)} \sum_{i=1}^{B(d,k)} Y_{ki}^{(d)}(x) Y_{ki}^{(d)}(y). \tag{55}$$

These properties imply that, up to a constant, $Q_k^{(d)}(\langle x, y \rangle)$ is a representation of the projector onto the subspace of degree-$k$ spherical harmonics

$$(\overline{\mathsf{P}}_k f)(x) = B(d, k) \int_{\mathbb{S}^{d-1}(\sqrt{d})} Q_k^{(d)}(\langle x, y \rangle) f(y) \tau_d(\mathrm{d}y). \tag{56}$$

For a function $\sigma \in L^2([-\sqrt{d}, \sqrt{d}], \tau_d^1)$ (where $\tau_d^1$ is the distribution of $\langle e_1, x \rangle$ when $x \sim \mathrm{Unif}(\mathbb{S}^{d-1}(\sqrt{d}))$), denoting its spherical harmonics coefficients $\xi_{d,k}(\sigma)$ to be

$$\xi_{d,k}(\sigma) = \int_{[-\sqrt{d}, \sqrt{d}]} \sigma(x) Q_k^{(d)}(\sqrt{d}x) \tau_d^1(\mathrm{d}x), \tag{57}$$

then we have the following equation holds in $L^2([-\sqrt{d}, \sqrt{d}], \tau_d^1)$ sense

$$\sigma(x) = \sum_{k=0}^{\infty} \xi_{d,k}(\sigma) B(d, k) Q_k^{(d)}(\sqrt{d}x).$$

For any rotationally invariant kernel $H_d(x_1, x_2) = h_d(\langle x_1, x_2 \rangle / d)$, with $h_d(\sqrt{d} \cdot) \in L^2([-\sqrt{d}, \sqrt{d}], \tau_d^1)$, we can associate a self adjoint operator $\mathcal{H}_d : L^2(\mathbb{S}^{d-1}(\sqrt{d})) \to L^2(\mathbb{S}^{d-1}(\sqrt{d}))$

$$\mathcal{H}_d f(x) \equiv \int_{\mathbb{S}^{d-1}(\sqrt{d})} h_d(\langle x, x_1 \rangle / d) f(x_1) \tau_d(\mathrm{d}x_1). \tag{58}$$

By rotational invariance, the space $V_k$ of homogeneous polynomials of degree $k$ is an eigenspace of $\mathcal{H}_d$, and we will denote the corresponding eigenvalue by $\xi_{d,k}(h_d)$. In other words $\mathcal{H}_d f(x) \equiv \sum_{k=0}^{\infty} \xi_{d,k}(h_d) \overline{\mathsf{P}}_k f$. The eigenvalues can be computed via

$$\xi_{d,k}(h_d) = \int_{[-\sqrt{d}, \sqrt{d}]} h_d(x/\sqrt{d}) Q_k^{(d)}(\sqrt{d}x) \tau_d^1(\mathrm{d}x). \tag{59}$$

For a dot product kernel $H_d(\boldsymbol{x}, \boldsymbol{y}) = h_d(\langle \boldsymbol{x}, \boldsymbol{y} \rangle / d)$ consider the Gegenbauer expansion of $h_d$ in $L^2([-\sqrt{d}, \sqrt{d}], \tau_d^1)$

$$h_d(\langle \boldsymbol{x}, \boldsymbol{y} \rangle / d) = \sum_{k=0}^{\infty} \xi_{k,d}(h_d) B(d, k) Q_k^{(d)}(\langle \boldsymbol{x}, \boldsymbol{y} \rangle). \tag{60}$$

Using Eq. (54) we can equivalently write the kernel as an expectation over random features for some activation $\sigma_d$

$$h_d(\langle \boldsymbol{x}, \boldsymbol{y} \rangle / d) = \mathbb{E}_{\boldsymbol{w} \sim \mathrm{Unif}(\mathbb{S}^{d-1})}[\sigma_d(\langle \boldsymbol{w}, \boldsymbol{x} \rangle) \sigma_d(\langle \boldsymbol{w}, \boldsymbol{y} \rangle)] \tag{61}$$

by taking

$$\sigma_d(x) = \sum_{k=0}^{\infty} \xi_{d,k}(h_d)^{1/2} B(d, k) Q_k^{(d)}(\sqrt{d}x). \tag{62}$$

Note that $\sigma_d \in L^2([-\sqrt{d}, \sqrt{d}], \tau_d^1)$ as long as $h(1) < \infty$.

### G.4 HERMITE POLYNOMIALS

The Hermite polynomials $\{\mathrm{He}_k\}_{k \geq 0}$ form an orthogonal basis of $L^2(\mathbb{R}, \gamma)$, where $\gamma(\mathrm{d}x) = e^{-x^2/2} \mathrm{d}x / \sqrt{2\pi}$ is the standard Gaussian measure, and $\mathrm{He}_k$ has degree $k$. We will follow the classical normalization (here and below, expectation is with respect to $G \sim \mathcal{N}(0, 1)$):

$$\mathbb{E}\{\mathrm{He}_j(G) \mathrm{He}_k(G)\} = k! \, \delta_{jk}. \tag{63}$$

As a consequence, for any function $g \in L^2(\mathbb{R}, \gamma)$, we have the decomposition

$$g(x) = \sum_{k=0}^{\infty} \frac{\mu_k(g)}{k!} \mathrm{He}_k(x), \qquad \mu_k(g) \equiv \mathbb{E}\{g(G) \mathrm{He}_k(G)\}. \tag{64}$$

The Hermite polynomials can be obtained as high-dimensional limits of the Gegenbauer polynomials introduced in the previous section. Indeed, the Gegenbauer polynomials (up to a $\sqrt{d}$ scaling in domain) are constructed by Gram-Schmidt orthogonalization of the monomials $\{x^k\}_{k \geq 0}$ with respect to the measure $\tilde{\tau}_d^1$, while Hermite polynomial are obtained by Gram-Schmidt orthogonalization with respect to $\gamma$. Since $\tilde{\tau}_d^1 \Rightarrow \gamma$ (here $\Rightarrow$ denotes weak convergence), it is immediate to show that, for any fixed integer $k$,

$$\lim_{d \to \infty} \mathrm{Coeff}\{Q_k^{(d)}(\sqrt{d}x) B(d, k)^{1/2}\} = \mathrm{Coeff}\left\{\frac{1}{(k!)^{1/2}} \mathrm{He}_k(x)\right\}. \tag{65}$$

Here and below, for $P$ a polynomial, $\mathrm{Coeff}\{P(x)\}$ is the vector of the coefficients of $P$. As a consequence, for any fixed integer $k$, we have

$$\mu_k(\sigma) = \lim_{d \to \infty} \xi_{d,k}(\sigma)(B(d, k)k!)^{1/2}, \tag{66}$$

where $\mu_k(\sigma)$ and $\xi_{d,k}(\sigma)$ are given in Eq. (64) and Eq. (57).

### G.5 THE INVARIANT FUNCTION CLASS AND THE SYMMETRIZATION OPERATOR

Let $\mathcal{G}_d$ be a group that is isomorphic to a subgroup of $\mathcal{O}(d)$, the orthogonal group in $d$ dimension. That means, each element of $\mathcal{G}_d$ can be identified with a matrix in $\mathcal{O}(d) \subseteq \mathbb{R}^{d \times d}$, and the group addition operation in $\mathcal{G}_d$ can be regarded as matrix multiplications in $\mathcal{O}(d)$. For any $\boldsymbol{x} \in \mathbb{S}^{d-1}(\sqrt{d})$ and $g \in \mathcal{G}_d$, we define group action $g \cdot \boldsymbol{x}$ to be the multiplication of matrix representation of $g$ with the vector $\boldsymbol{x}$. We equip $\mathcal{G}_d$ with a probability measure $\pi_d$, which is the uniform probability measure on $\mathcal{G}_d$. More specifically, the Borel sigma algebra on $\mathcal{G}_d$ is defined as the Borel sigma algebra of $\mathcal{O}(d)$ restricted on $\mathcal{G}_d$. The uniform probability measure $\pi_d$ satisfies the property that, for any Borel-measurable set $B \subseteq \mathcal{G}_d$ and any $g \in \mathcal{G}_d$, we have

$$\pi_d(B) = \pi_d(gB).$$

Let $L^2(\mathbb{S}^{d-1}(\sqrt{d}))$ be the class of $L^2$ functions on $\mathbb{S}^{d-1}(\sqrt{d})$ equipped with uniform probability measure $\mathrm{Unif}(\mathbb{S}^{d-1}(\sqrt{d}))$. We define the invariant function class to be

$$L^2(\mathbb{S}^{d-1}(\sqrt{d}), \mathcal{G}_d) = \left\{ f \in L^2(\mathbb{S}^{d-1}(\sqrt{d})) : f(\boldsymbol{x}) = f(g \cdot \boldsymbol{x}), \ \forall \boldsymbol{x} \in \mathbb{S}^{d-1}(\sqrt{d}), \ \forall g \in \mathcal{G}_d \right\}.$$

We define the symmetrization operator $\mathcal{S} : L^2(\mathbb{S}^{d-1}(\sqrt{d})) \to L^2(\mathbb{S}^{d-1}(\sqrt{d}), \mathcal{G}_d)$ to be

$$(\mathcal{S}f)(\boldsymbol{x}) = \int_{\mathcal{G}_d} f(g \cdot \boldsymbol{x}) \pi_d(\mathrm{d}g).$$

### G.6 ORTHOGONAL POLYNOMIALS ON INVARIANT FUNCTION CLASS

We define $V_{d,\leq k} \subseteq L^2(\mathbb{S}^{d-1}(\sqrt{d}))$ to be the subspace spanned by all the degree $\ell$ polynomials, $V_{d,>k} \equiv V_{d,\leq k}^{\perp} \subseteq L^2(\mathbb{S}^{d-1}(\sqrt{d}))$ to be the orthogonal complement of $V_{d,\leq k}$, and $V_{d,k} = V_{d,\leq k} \cap V_{d,\leq k-1}^{\perp}$. In words, $V_{d,k}$ contains all degree $k$ polynomials that orthogonal to all polynomials of degree at most $k-1$. We further define $V_{d,<k} = V_{d,\leq k-1}$ and $V_{d,\geq k} = V_{d,>k-1}$.

Let $\overline{\mathsf{P}}_{\leq \ell}$ to be the projection operator on $L^2(\mathbb{S}^{d-1}(\sqrt{d}), \mathrm{Unif})$ that project a function onto $V_{d,\leq \ell}$, the space spanned by all the degree $\ell$ polynomials. Then it is easy to see that $\overline{\mathsf{P}}_{\leq \ell}$ and $\mathcal{S}$ operator commute. This means, for any $f \in L^2(\mathbb{S}^{d-1}(\sqrt{d}))$, we have

$$\overline{\mathsf{P}}_{\leq \ell}[\mathcal{S}(f)] = \mathcal{S}[\overline{\mathsf{P}}_{\leq \ell}(f)].$$

Similarly, we can define $\overline{\mathsf{P}}_\ell$, $\overline{\mathsf{P}}_{<\ell}$, $\overline{\mathsf{P}}_{>\ell}$, $\overline{\mathsf{P}}_{\geq \ell}$, which commute with $\mathcal{S}$. We denote $V_{d,\ell}(\mathcal{G}_d) \equiv \mathcal{P}_\ell(\mathbb{S}^{d-1}(\sqrt{d}), \mathcal{G}_d)$ to be the space of polynomials in the images of $\overline{\mathsf{P}}_\ell \mathcal{S}$. Then we have

$$\mathcal{P}_\ell(\mathcal{A}_d, \mathcal{G}_d) = \overline{\mathsf{P}}_\ell(L^2(\mathbb{S}^{d-1}(\sqrt{d}), \mathcal{G}_d)) = \mathcal{S}[\overline{\mathsf{P}}_\ell(L^2(\mathcal{A}_d))].$$

We denote $D(d;k) = D(\mathbb{S}^{d-1}(\sqrt{d}); \mathcal{G}_d; k) \equiv \dim(\mathcal{P}_k(\mathbb{S}^{d-1}(\sqrt{d}), \mathcal{G}_d))$ to be the dimension of $\mathcal{P}_k(\mathbb{S}^{d-1}(\sqrt{d}), \mathcal{G}_d)$. We denote $\{\overline{Y}_{kl}^{(d)}\}_{l \in [D(\mathbb{S}^{d-1}(\sqrt{d});k)]}$ to be a set of orthonormal polynomial basis in $\mathcal{P}_k(\mathbb{S}^{d-1}(\sqrt{d}), \mathcal{G}_d)$. That means

$$\mathbb{E}_{\boldsymbol{x} \sim \mathrm{Unif}(\mathbb{S}^{d-1}(\sqrt{d}))}[\overline{Y}_{k_1 l_1}^{(d)}(\boldsymbol{x}) \overline{Y}_{k_2 l_2}^{(d)}(\boldsymbol{x})] = \mathbf{1}\{k_1 = k_2, l_1 = l_2\},$$

and

$$\overline{Y}_{kl}^{(d)}(\boldsymbol{x}) = \overline{Y}_{kl}^{(d)}(g \cdot \boldsymbol{x}), \ \forall \boldsymbol{x} \in \mathbb{S}^{d-1}(\sqrt{d}), \ \forall g \in \mathcal{G}_d.$$

