# OpenReview forum: "The Three Stages of Learning Dynamics in High-dimensional Kernel Methods"
_ICLR.cc/2022/Conference — ICLR 2022 Poster_

### Official Review · Reviewer_WyHh · 2021-11-01

**Correctness:** 4
**Technical Novelty And Significance:** 4
**Empirical Novelty And Significance:** 4
**Recommendation:** 8
**Confidence:** 5

**Main Review:**

This is a timely paper: the deep bootstrap paper (Nakkiran et al. 2020) reported
an interesting observation, which the authors investigate here in a more
controlled setting. While the authors don't seem to go significantly beyond the
technical tools developed by Mei et al. and Ghorbani et al., which the present
authors acknowledge, they use them to provide theoretical insight on a question
not directly investigated by Mei and Ghorbani.

This paper is also well-written: the structure is clear, previous work is
acknowledged, the schematic plots in Fig. 1+2 are helpful. I was only unsure
about how to interpret Fig. 3: is it a purely schematic plot, i.e. a drawing? Is
it obtained from simulations, or from theory? Also, the target function is
noisy, so how can even the oracle world reach zero test error in Fig. 3? Or is
the test error simply very small?

My only qualm with the presentation regards Fig. 5 and the section on SGD. If I
understand correctly, this plot only shows that the difference between gradient
flow and the discrete updates of SGD are very small, is that correct? If so, I
don't find the agreement between gradient flow predictions and discrete SGD
updates so surprising - or are there specific reasons why it would be? Instead,
it might be worth relegating these plots to the appendix and showing what
happens if you know train two-layer ReLU networks end-to-end on the same
dataset. While I understand that you focus on kernel ridge regression here,
bridging the gap to a setup with feature learning, if only numerically, would be
another interesting contribution imho. I would understand if the authors prefer
to keep the paper in the RF setting though.


**Summary Of The Paper:**

This paper studies the learning dynamics of gradient flow for kernel ridge
regression. The authors contrast to different setups: the "empirical world",
where the model is trained on a finite data set, and the "oracle world", where
the model is instead trained directly on the population loss, or in other words,
by minimising the test error. Nakkiran et al. (2020) recently reported that the
errors in both setups stay close to each throughout training for a range of deep
neural networks. By studying the relation between empirical and oracle world in
the setup of kernel ridge regression, the authors investigate this phenomenon in
a setup under precise theoretical control.

The authors perform such a study by leveraging the analysis of Ghorbani et
al. (2021) and Mei et al. (2021), who gave a detailed and precise analysis of
the implicit bias of learning in random feature models, and in particular the
learning of approximations of increasing complexity.

Contrasting the learning in the two worlds, the authors find that learning
generally, but not always, proceeds in three stages. In the case of polynomial
kernels, models in both the empirical and the oracle learn first the leading
$\ell$-components of the target function, defined as those projections of the
target function along the polynomials of small degree. After the training error
of the empirical model reaches zero, a second phase ensues where the test errors
of both models remain close, but the gap between training and test error in the
empirical world can be large. Finally, either there is enough training data to
learn the target function perfectly, or learning enters a third stage where the
oracle model learns the target function perfectly, which the empirical model
won't achieve.

The authors confirm this picture also for invariant kernels, say
translation-invariant ones, using technical tools from Mei et
al. (2021). Quantitatively, the presence of an invariance shows up in the
training speed, which is slower than for the product kernel (Remark
2). Numerical experiments for kernel least-square confirm the theory, while the
authors also report experiments for SGD for random feature regression.

**Summary Of The Review:**

I see the contribution of this paper in providing a theoretically sound
treatment of the observations presented by Nakkiran et al. (2020) in a
controlled setting. I think this paper would be a valuable contribution to ICLR,
and recommend acceptance.

*Edit 14.11.* After the first round of responses, and after reading the other reviews, I have increased my confidence score from 4 to 5.

---

> ### Author Response · Authors · 2021-11-13
> **Response to Reviewer WyHh**
>
> Thank you for your thoughtful and detailed comments! We are happy you view our paper as a valuable contribution. To address some concerns
>
> **Q1: Regarding Fig. 3**
>
> Fig. 3 is indeed purely a drawing of the theoretical results in the asymptotic limit. The drawings in Fig. 3 depict noiseless settings ($\sigma^2 = 0$), hence the oracle world can reach zero test error. However in Appendix F, column 2 of Fig. 8 shows a setting where $\sigma^2 = 0.2$, in which case the oracle world cannot reach zero error.
>
> **Q2(a): Regarding Fig. 5**
>
> The motivation for this plot was to show that our theory-based continuous-time kernel setting was still reflective of the slightly more realistic “bootstrap” experimental setting where we used mini-batch SGD with a *discrete* step-size to train a random features model with *finitely many features*. Perhaps the closeness of the plots is not so surprising since they should be the same in the limit, but we wanted to just numerically confirm that they are qualitatively similar for reasonable parameter choices.
>
> **Q2(b): Plots for two-layer networks**
>
> We agree that training two-layer networks could be an interesting experiment. However, if the network is trained in the feature-learning regime then the results will fall outside the realm of our theory and we would prefer to keep the paper in the kernel/RF setting.

---

> > ### Comment · Reviewer_WyHh · 2021-11-14
> > **Thank you for your explanations.**
> >
> > Thank you for your clarifications. I understand the purpose of Fig. 5 better now. I see your point about wanting to keep the experiments in the realm of the theory; I think this is a choice that should be up to the authors, so I am fine with this.
> >
> > I have increased the confidence in my comment since I don't seem to have missed any major point.

---

### Official Review · Reviewer_5uQz · 2021-11-01

**Correctness:** 4
**Technical Novelty And Significance:** 3
**Empirical Novelty And Significance:** Not applicable
**Recommendation:** 6
**Confidence:** 4

**Main Review:**

This is a good paper that applies the machinery developed in [1] to solve / explain certain DL phenomena. In what follows, I briefly summarize the strategies / contributions.

In the kernel gradient descent setting, the dynamics can be characterized by the gradient flow (solution) operator $G_{t, X}$, where $t$ and $X$ are the time and training dataset, resp. The $t=\infty$ (i.e. kernel regression) is solved in [1, 2]. The current paper analyzes the setting of $t<< |X|$ (stage 1), $t\sim |X|$ (stage 2) and $t>> |X|$ stage 3. The paper shows that, in the high dimensional setting, $G_{t, X}\approx P_{\min (t, |X|)}$, where $P_r$ denotes the projection operator onto certain low-frequency eigenspace (depends on $r$). The main technical difference between the current paper and [1] seems to be: in [1] $G_{t=\infty, |X|}$ is essentially a matrix inversion and in here $t<\infty$, $G_{t, |X|}$ is a matrix exponentially. This difference requires certain technical treatment and other than this, the overall strategies seem quite similar.

In sum, the strength and weakness of the paper are

(Strength) Explaining/ solving certain phenomena that were previous observed in DL setting using "Kernels".

(Weakness) Technical innovation is not very high and largely depends on the framework of [1]. As mentioned in (Strength) above, the proposed problem itself is mainly about "Kernels" rather than something unique in DL.


Minor Comments:
(1). The gradient flow dynamics in the current paper seems to be the limit of GD dynamics rather than SGD.


[1] Generalization error of random features and kernel methods: hypercontractivity and kernel matrix concentration.

[2] Learning with invariances in random features and kernel models

**Summary Of The Paper:**

The paper studies the training/learning dynamics of (inner product) kernel gradient descent in the high-dimensional setting. The main contribution is proving a three-stage learning dynamics which were observed in neural networks in existing work.
(1) Stage 1: train loss ~ test loss ~ oracle test loss (training set == whole (input, label) distribution)
(2) Stage 2: test loss ~ oracle loss; train loss ->0;
(3) Stage 3: test loss unchanged; oracle loss decays;

I think this is a nice application of the results developed in [1], showing that many seemingly surprising deep learning (DL) phenomena are indeed not unique in DL -- most of them are provable / observable in the kernel setting.

[1]Generalization error of random features and kernel methods: hypercontractivity and kernel matrix concentration.

**Summary Of The Review:**

Although this is not a breakthrough paper and the technical contributions is not very high, I believe this is another nice paper showing that a lot of DL phenomena are can be explained by Kernels. The analysis also offers some basic intuition of neural networks' learning/training dynamics.

---

> ### Author Response · Authors · 2021-11-13
> **Response to Reviewer 5uQz**
>
> Thank you for your thoughtful comments! We are glad you felt this was a nice paper. To address your minor comment (a repeat from the response to Reviewer Ud7D who raised the same concern):
>
> **limiting SGD dynamics**
>
> We are considering a limiting regime of SGD where the step-size goes to zero. Technically speaking, SGD with any batch size leads to deterministic gradient flow dynamics in this limit (see https://francisbach.com/gradient-flows/ for a proof and discussion). Intuitively, this is because the SGD noise gets multiplied by the step-size. Our numerical results in Section 4.2, for which we use SGD with a non-zero step-size and small batch size, show that this is not an unreasonable approximation (see Fig. 5).

---

> > ### Comment · Reviewer_5uQz · 2021-11-22
> > **reply**
> >
> > Thanks for the clarification regarding SGD!

---

### Official Review · Reviewer_qXRH · 2021-11-02

**Correctness:** 3
**Technical Novelty And Significance:** 1
**Empirical Novelty And Significance:** 2
**Recommendation:** 5
**Confidence:** 2

**Main Review:**

Strengths: Nice empirical results studying several cases around the three phases in the MSE curves.

Weaknesses: Although the connection to deep learning is very weak, the paper mentions deep bootstrap and some other connections with deep learning.

In my opinion, the paper would benefit from more motivation for comparing the oracle risk together with the usual train and test errors. As it stands, the paper's attraction is rather limited.

Some issues:
- Is \beta_t on page 1 a scalar or a vector? Either way, there seems to be a problem with the linear dynamics on the eq. at the bottom of the page.
- In theorem 1, I could not find where w_d(1) is introduced.

Typos:
- In Figure 1, the oracle world legend should write R(f_t^{or})
- At the top of page 2, f is not defined in the definition of K_N, I guess it is f_N. Normalization with N here does not make sense to me as N was only introduced as an index of growing width networks on page 1 (instead of precisely the width of the network)




**Summary Of The Paper:**

The authors discuss three phases of the MSE loss curve: in the first phase the training, test, and the oracle errors remain close together; in the second phase, the training error goes to zero; and in the third case, the oracle loss converges at the approximation level whereas the test loss is higher. There are two theorems on dot product kernels (including some NTKs) and invariant kernels where the functions through linear dynamics with two kernels exhibit three phases.

**Summary Of The Review:**

It is not clear why the main message of the paper around the three stages of learning is interesting for the community. The paper does study several cases including random features. Nice experiments are included supporting the main message of the paper.

---

> ### Author Response · Authors · 2021-11-13
> **Response to Reviewer qXRH**
>
> Thank you for carefully reading our paper and for your feedback! We are glad you liked the empirical random features results. We hope that we can clarify and highlight what we believe are the attractions and contributions of our paper:
>
> Our main motivation was to study a stylized “deep learning” model that allows for mathematical analysis, in order to gain theoretical insight into gradient descent training dynamics.
>
> 1) This stylized setting has non-trivial connections with deep learning since it exactly describes neural network training dynamics in the limiting regime of large width, small step-size, and high data dimension.
>
> 2) It is valuable to see that certain notable "deep" phenomena (e.g. the deep bootstrap) are in fact not specific to “deep learning” and can hold in simpler settings (c.f. Reviewers 5uQz, WyHh).
>
> 3) Furthermore, even outside of the context of deep learning, kernels and random features are interesting and useful tools. We provide relevant theory for these methods which accurately matches up with empirical evidence.
>
> 4) The motivation to compare the oracle and empirical dynamics was to a) draw a connection to the deep bootstrap phenomenon and b) gain further conceptual insight into the training dynamics (e.g. Figure 2). It is often insightful to compare complicated objects with simpler analogues (e.g. empirical world vs oracle world, finite-width networks vs NTK, etc.).
>
> 5) As noted by Reviewer Ud7D, there is an interesting connection to neural scaling laws. Namely, we show that scaling optimization time can play a role similar to scaling dataset size in certain regimes.
>
> Regarding the issues and typos: thank you for raising these, we address them in our revised paper.
>
> **Q1: $\beta_t$ on page 1**
>
> In our revised version we remove the equations on page 1 since they are not important for the rest of the paper and need more care to be stated precisely. We instead elaborate using words our main point, which is to introduce the known connection between the training dynamics of overparameterized neural networks and the neural tangent kernel.
>
> **Q2: $\omega_d$ notation in Theorem statements**
>
> We now define this asymptotic notation in the text before Theorem 1.
>
> **Q3: Typo in Fig. 1**
>
> Thanks for point this out.
>
> **Q4: Typo in definition of $K_N$**
>
> You are correct that $K_N$ should not have the normalization factor.

---

> > ### Comment · Reviewer_qXRH · 2021-11-29
> > **thanks for the response**
> >
> > I agree that it is very valuable to study objects in their infinite- limits. I would find the paper much more interesting if the authors studied the differences in the learned functions for the oracle vs. the empirical world (in the spirit of Figure 2). However, the paper compares these two functions only through the lens of the test and training errors. From this perspective, given the technical results of Ghorbani et al. (2021) and Mei et al. (2021), I found the results limited.
> >
> > I also agree that the kernel regression and random features are interesting models on their own. I would prefer if the paper was written strictly in this scope without major claims about deep learning in the introduction and elsewhere. Without further experimentation, I fail to see how any claim about deep learning fits in this paper.
> >
> > The role of the training time as a bottleneck very similar to the size of the training dataset size is indeed interesting!

---

> > > ### Author Response · Authors · 2021-12-02
> > > **Reply to Reviewer qXRH**
> > >
> > > Thank you for your response! We would like to address some of the comments made.
> > >
> > > >I would find the paper much more interesting if the authors studied the differences in the learned functions for the oracle vs. the empirical world (in the spirit of Figure 2). However, the paper compares these two functions only through the lens of the test and training errors.
> > >
> > > We would like to point out that we do actually characterize the learned functions. For example, in Theorems 1 and 2 the learned functions $f_t^{\rm{or}}$ and $\hat{f}_t$ are the low-degree polynomial components of the target function $f_d$. These two functions are different in stage 2 (on the training set) and in stage 3 (in an $L^2$ sense). This result is the direct inspiration for Fig. 2.
> > >
> > > >I would prefer if the paper was written strictly [...] without major claims about deep learning [...]. Without further experimentation, I fail to see how any claim about deep learning fits in this paper.
> > >
> > > We did make a conscious effort to not make any major claims about deep learning, stating instead that our results hold for the kernel setting which is a limiting case of deep learning (see for example the paragraph before Section 1.1, Section 5, etc.). We do believe however, that it would be scientifically incorrect to not state anything about deep learning since 1) there is a direct connection to deep learning via the NTK and 2) the analysis of training dynamics is motivated by and related to empirical phenomena in deep learning. If there any specific overclaims you had in mind, please let us know so we can address them in the revision.

---

### Official Review · Reviewer_Ud7D · 2021-11-03

**Correctness:** 4
**Technical Novelty And Significance:** 3
**Empirical Novelty And Significance:** 3
**Recommendation:** 6
**Confidence:** 4

**Main Review:**

In my opinion this is an interesting paper.
Prior works have already shown that under appropriate conditions, kernel models can learn at most an $\alpha$-degree polynomial, where $\alpha$ depends on the relation between the sample size and input dimensionality, as well as certain invariance structure.
While it is not very surprising that the training time of kernel gradient flow (up to the sample size bottleneck) plays a similar role as the sample size in kernel regression, to my knowledge this is the first work that analyzes this correspondence in the high-dimensional asymptotic limit. Such result may be interpreted as a "scaling law" for optimization, i.e., a certain sample size and training time is required to learn a target function with certain complexity. I believe the ICLR community will find this message relevant.

A few comments and questions.

1. My impression is that the theoretical analysis does not consider SGD. If this is the case, then mentioning SGD multiple times in the abstract and introduction can be misleading.

2. Related to the previous point, the population dynamics (2) requires an integration over the input distribution, which is not very realistic. If we optimize the model using one-pass SGD instead, would the number of training steps be analogous to the training time in gradient flow?

3. If we consider gradient descent on random features models (as in Section 4.2), then following the results in [Mei et al. 2021], do we expect the model width to play a similar role (in terms of limiting the complexity of the learned function) as the training set size or training time?

4. It might be a good idea to elaborate on how results in this submission differ from the classical nonparametric rates for kernel regression, in which the generalization error rate is typically specified by the source and capacity condition. What are the benefits of working with the asymptotic setting? For example, the depicted three-stage phenomenon is not really that surprising, and I don't think it is only present in high dimensions.

5. The listed assumptions in Appendix A.2 are rather opaque. Can the authors comment on whether it is straightforward to verify these conditions for more general input distributions (beyond unit sphere or hypercube), such as a Gaussian mixture?

**Summary Of The Paper:**

Studied the evolution of generalization error of the kernel gradient flow trajectory with respect to the training (empirical world) and population (ideal world) MSE loss.
The analysis builds upon [Mei et al. 2021], which relates kernel ridge regression to projection onto low-degree polynomials. The authors showed that the estimator optimizing the empirical risk achieves vanishing training error, but the test error plateaus at certain value depending on the training set size, whereas the online (population) estimator can learn increasingly complex components of the target function as training proceeds.

**Summary Of The Review:**

I believe the contributions in this paper are solid, and the take-home message is of interest to the ICLR community. I will consider adjusting my score if the authors can address some of the aforementioned concerns.

----------------------Post-rebuttal update----------------------
Thank you for the detailed reply, which addressed some of my concerns. In my opinion this submission is above the acceptance bar (I think 7 would be more appropriate for my evaluation, but that is not an option...)

---

> ### Author Response · Authors · 2021-11-13
> **Response to Reviewer Ud7D**
>
> Thank you for your thoughtful feedback! We are glad you found our paper interesting. We address your comments / questions below:
>
> **Q1: analysis does not consider SGD**
>
> We are considering a *limiting regime* of SGD where the step-size goes to zero. Technically speaking, SGD with any batch size leads to deterministic gradient flow dynamics in this limit (see https://francisbach.com/gradient-flows/ for a proof and discussion). Intuitively, this is because the SGD noise gets multiplied by the step-size. Our numerical results in Section 4.2, for which we use SGD with a non-zero step-size and small batch size, show that this is not an unreasonable approximation (see Fig. 5).
>
> **Q2: population dynamics**
>
> As noted in the previous point, one-pass SGD with vanishing step-size will result in the population gradient flow dynamics. As you remark however, analytically solving the dynamics requires knowing the input distribution and we study the analytical solution purely as a theoretical consideration.
>
> In the Section 4.2 experiments, for the Oracle World we do one-pass SGD with discrete step-size and indeed, the number of training steps corresponds to the training time $t$ of the gradient flow (see Appendix E.3 for more discussion).
>
> **Q3: random features**
>
> This is a good question! We agree that a natural conjecture would be that the number of features has a similar effect on the model complexity and leave a careful verification of this for future work.
>
> **Q4: comparison with classical non-parametrics**
>
> This is a good point and we added a short remark about this in the Related Literature section. To elaborate further here, in general the source condition requires making a strong assumption on the regularity of the target function. In our work we make much milder assumptions on the target function (i.e. bounded $L^{2+\eta}$ norm for some $\eta > 0$).
>
> Another benefit of the high dimensional setting is that we get precise limiting expressions and not potentially loose bounds which may hide big constant factors. As a result we can rigorously prove the deep bootstrap phenomenon in such a setting. Furthermore, in low-dimensions although the first and third stages will still occur, it is difficult to reliably observe the second stage since the empirical model is susceptible to overfitting.
>
> **Q5: assumptions in Appendix A.2**
>
> We are able to reduce the assumptions in Appendix A.2 to a simpler and easier to verify set of assumptions (see Appendices C.2, D.2) for uniform data on the sphere by making use of spherical harmonics decompositions. More generally however, it is not straightforward to verify these assumptions for other input distributions. However, we believe that these assumptions are mostly technical and that the three-stage phenomena should hold more broadly.

---

### Author Response · Authors · 2021-11-13
**Revision Uploaded**

We have uploaded a revision of our paper with some updates. We primarily update the writing for increased clarity. In particular, there are no changes to the results or main content. In our changes, we address some points of confusion and typos raised by the reviewers. We also restructure the proof in the Appendices and add more details to improve readability.

---

### Decision · Program_Chairs · 2022-01-20

**Decision:**

Accept (Poster)

**Comment:**

*Summary:* Study gradient flow dynamics of empirical and population square risk in kernel learning.

*Strengths:*
- Empirical results studying several cases in MSE curves.
- Explaining / solving certain phenomena in DL using kernels.

*Weaknesses:*
- More motivations would be appreciated.
- Technical innovation not so high.

*Discussion:*

Ud7D found that the main strength of this paper is the take-home message rather than innovations. They concluded 7 might be appropriate for the evaluation. This opinion was seconded by WyHh who considered 7 the most appropriate rating. 5uQz also found that 7 would be the most appropriate rating. qXRH maintained concerns about the novelty of the work and rating 5. Nonetheless, they agreed the study is valuable and would not oppose acceptance.

*Conclusion:*

Three reviewers found this paper is definitely above the acceptance threshold (suggesting rating 7) and one more reviewer found it marginally below the acceptance threshold however not opposing acceptance. I found the general impressions from the discussion well described in a comment from Ud7D, who indicates that although this is not a breakthrough paper, it is a nice paper showing that a lot of DL phenomena are can be explained by Kernels. I conclude that the paper makes a sufficiently valuable contribution and hence I am recommending accept. I suggest the authors take the reviewers’ comments carefully into account when preparing the final version of the manuscript.